



# Management matters: Testing a mitigation strategy for nitrous oxide emissions on intensively managed grassland

Kathrin Fuchs[1], Lukas Hörtnagl[1], Nina Buchmann[1], Werner Eugster[1], Valerie Snow[2], Lutz Merbold[1,3]

[1] Department of Environmental Systems Science, Institute of Agricultural Sciences, ETH Zurich, Universitätstrasse 2, CH-8092 Zurich, Switzerland
[2] AgResearch, Grasslands Research Centre, PB 11008, Palmerston North 4442, New Zealand
[3] Mazingira Centre, International Livestock Research Institute (ILRI), P.O. Box 30709, KE-00100 Nairobi, Kenya

*Correspondence to*: Kathrin Fuchs (kafuchs@ethz.ch)

**Abstract.** Replacing fertilizer nitrogen with biological nitrogen fixation (BNF) through legumes has been suggested as a strategy for nitrous oxide ($N_2O$) mitigation from intensively managed grasslands. While current literature provides evidence for an $N_2O$ emission reduction effect due to reduced fertilizer input, little is known about the effect of increased legume proportions potentially offsetting these reductions, i.e. by increased $N_2O$ emissions from plant residues and root exudates. In order to assess the overall effect of this mitigation strategy on permanent grassland, we performed an *in-situ* experiment to quantify net $N_2O$ fluxes and biomass yields in two differently managed grass-clover mixtures. We measured $N_2O$ fluxes in an unfertilized parcel with high clover proportions vs. a fertilized control parcel with low clover proportions using the eddy–covariance (EC) technique over two years. Furthermore, we related the measured $N_2O$ fluxes to management and environmental drivers. To assess the effect of the mitigation strategy, we measured biomass yields and quantified biologically fixed nitrogen using the $^{15}N$ natural abundance method.

The mitigation management effectively reduced $N_2O$ emissions by 54% and 39% in 2015 and 2016, respectively. These reductions in $N_2O$ emissions can be attributed to the absence of fertilization on the clover parcel. Differences in clover proportions during periods with no recent management showed no measurable effect on $N_2O$ emissions, indicating that decomposition of plant residues and rhizodeposition did not compensate the effect of fertilizer reduction on $N_2O$ emissions. Annual biomass yields were similar under mitigation management, resulting in a reduction of $N_2O$ emission intensities from 0.42 g $N_2O$-N $kg^{-1}$ DM (control) to 0.28 g $N_2O$-N $kg^{-1}$ DM (clover parcel) over the two years observation period. We conclude that $N_2O$ emissions from fertilized grasslands can be effectively reduced without losses in yield by increasing the clover proportion and reducing fertilization.

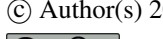



## 1 Introduction

Agricultural practices contribute 5.4 Gt $CO_2$-eq. $yr^{-1}$ (range 11–12%) to global greenhouse gas (GHG) emissions (IPCC, 2014; Tubiello et al., 2015). The technical potential to mitigate GHG emissions from agriculture ranges between 5.5 and 6.0 Gt $CO_2$-eq. $yr^{-1}$ by 2030 (Smith et al., 2008), exceeding current agricultural GHG emissions. The three major anthropogenic

GHGs comprise carbon dioxide ($CO_2$), methane ($CH_4$) and nitrous oxide ($N_2O$). The agricultural sector is responsible for 84% of global anthropogenic $N_2O$ emissions (Smith et al., 2008). $N_2O$ emissions are primarily attributed to mineral and organic fertilizer applied to soils, manure left on pastures, biomass burning, crop residues and increased mineralization of soil organic matter (SOM) caused by the cultivation of soils (IPCC, 2014; Tubiello et al., 2015). Due to the high global warming potentials of $CH_4$ and $N_2O$ (GWP, factor 34 and 298, respectively, on a per mass basis compared to $CO_2$ based on a

100-year time horizon) (IPCC, 2013), these gases are more important than the $CO_2$ fluxes from the agricultural sector. However, they remain far less understood than $CO_2$ fluxes because of interactions between multiple underlying processes that are largely unexplored. In particular, data resolving the dynamics of $N_2O$ fluxes from soils are still scarce, as advances in instruments capable of high-frequency continuous measurements and steadily deployable in the field have only become available in recent years (Eugster and Merbold, 2014).

Here we test a potential mitigation strategy for nitrous oxide emissions, namely the substitution of fertilizer with biological nitrogen fixation (BNF) via clover on intensively managed grassland. Processes producing and consuming $N_2O$ are numerous and their complex interactions and dependencies on biotic and abiotic factors that are generally known but not yet fully understood (Butterbach-Bahl et al., 2013). Nevertheless, it is known that $N_2O$ emissions particularly in grasslands strongly depend on management practices (Hörtnagl et al., 2018; Li et al., 2013; Snyder et al., 2009) and reducing $N_2O$

emissions while maintaining yields can thus contribute to climate smart agriculture (CSA) (Lipper et al., 2014). For mitigating $N_2O$ emissions from soils, a range of options (e.g. nitrification inhibitors, liming of acid soils, precision fertilizer use, legumes) are available (Bell et al., 2015; Flessa, 2012; de Klein and Eckard, 2008; Li et al., 2013; Luo et al., 2010; Paustian et al., 2016; Smith et al., 2008). The most important strategies focus on increasing the nitrogen use efficiency (NUE) of plants by adjusting the rate, type, timing and placement of organic and inorganic nitrogen fertilizers. With such

approaches, the surplus of nitrogen (N) as the substrate for microbial communities producing $N_2O$, can be reduced or avoided (Flessa, 2012; Galloway et al., 2003; Snyder et al., 2009). Reducing N surplus comes along with other environmental benefits such as reduced ammonia emissions ($NH_3$) and nitrate ($NO_3^-$) leaching, both potential sources of indirect (off-site) $N_2O$ emissions. Similar to these mitigation strategies, forage legume species of the Fabaceae family (e.g. white clover, red clover, lucerne, also called alfalfa) grown in grass-legume mixtures have the potential to reduce $N_2O$

emissions as a cost-effective mitigation strategy (Jensen et al., 2012). In legume-rich systems, large parts of the plants' nitrogen (N) demand can be provided from the atmosphere via BNF instead of using external fertilizer amendments (Ledgard et al., 2001; Suter et al., 2015). Hence, N input via BNF instead of fertilizers has the potential to avoid large N




surpluses by provisioning N in a manner synchronous to plant needs following their growth pattern (Crews and Peoples, 2005). Furthermore, BNF is down-regulated by the plant when demand is low and fixed N is located in the nodules and thus not freely available to microbiota in the soil (Lüscher et al., 2014; Nyfeler et al., 2011). Apart from the environmental benefits of a reduced N surplus when mineral fertilizer is replaced by BNF, total GHG emissions from fertilizer production

of 1.6–6.4 kg $CO_2$-eq per kg fertilizer N, could technically be avoided (Andrews et al., 2007; Brentrup and Pallière, 2008). Besides the obvious advantage of lower fertilizer amendments, grass–legume mixtures typically achieve higher yields than average grass and legume monocultures ("overyielding effect") and often also higher yields than the best performing monoculture ("transgressive overyielding"), with legume proportions of 40–70% resulting in highest yields (Finn et al., 2013; Lüscher et al., 2014; Nyfeler et al., 2009). In addition, growing selected legumes in mixtures with non-legumes could

improve resistance and resilience of forage swards against climatic extremes such as severe drought events (Hofer et al., 2017). Moreover, grass-legume mixtures are beneficial to fodder composition as they are characterized by higher protein contents than grass swards, and show well-balanced feeding values (Phelan et al., 2015). Legume-rich fodder has high crude protein (CP) contents and was shown to increase voluntary intake by 10–20% (Dewhurst et al., 2003), and to increase milk production (Dewhurst et al., 2003; Huhtanen et al., 2007).

Despite the known advantages, introducing legumes causes some challenges for farmers. For instance, maintaining a persistent optimal legume proportion of 40–60% (Lüscher et al., 2014) is not trivial (Guckert and Hay, 2001). Conservation of legumes as hay or silage can be more difficult than for grasses due to lower contents of water-soluble carbohydrates (WSC) and higher pH buffering capacities (Phelan et al., 2015). When protein-rich forage is fed without sufficient WSC, N cannot be used efficiently by livestock and N excretion from the animals increases (Phelan et al., 2015). However, the

balance between CP and WSC can be provided by carbohydrates from other plant species in mixtures (Lüscher et al., 2014). Furthermore, exceptionally high legume proportions (> 80%) and legume monocultures can lead to similar N surplus due to high levels of BNF as found in fertilized fields, and consequently to high soil nitrate concentrations (Weisser et al., 2017) which can subsequently lead to enhanced $N_2O$ emissions (Jensen et al., 2012). So far, relatively few in situ measurements at plot scale have been carried out to investigate the effect of legumes and grass-legume mixtures on $N_2O$ emissions (e.g.

studies by Klumpp et al., 2011; Virkajärvi et al., 2010; Schmeer et al., 2014; Niklaus et al., 2016; Li et al., 2011). The contribution of legumes to total field-scale $N_2O$ emissions was attributed to decomposition of N-rich plant residues and N from root exudates (Millar et al., 2004; Rochette and Janzen, 2005). Although it was shown that some Rhizobium species are able to produce $N_2O$ via rhizobial denitrification (O'Hara and Daniel, 1985; Rosen and Ljunggren, 1996), direct $N_2O$ emissions from BNF are negligible compared to $N_2O$ from denitrification rates for most investigated species and hence result

in no significant effect on field-scale $N_2O$ emissions (Garcia-Plazaola et al., 1993; Rochette and Janzen, 2005). To date, experimental studies investigating year-round $N_2O$ exchange in grassland systems are scarce (Skinner et al., 2014), and measurements of high temporal resolution in grassland relying on fertilizer input versus grassland based on BNF are missing. Thus, the aim of this study was to test the $N_2O$ mitigation strategy of substituting N fertilizer with BNF by increasing the clover proportion in grassland. Therefore, we measured $N_2O$ exchange and productivity in two adjacent



grassland parcels, one with an intensive "business as usual" management compared to a parcel where fertilizer amendments were substituted by over-sowing clover. Our specific objectives were (1) to quantify $N_2O$ emissions from both parcels, (2) to identify the drivers of $N_2O$ emissions, (3) to assess if substituting N fertilizer with BNF was an effective $N_2O$ mitigation strategy. We hypothesized considerably lower $N_2O$ emissions in the clover parcel, lower soil nutrient availability in the

clover parcel and thus no effect of legume proportions on $N_2O$ emissions, while fertilization to play the dominant role in driving $N_2O$ emissions in the control parcel. We further expected minor differences in grassland yield between the two parcels, and as a consequence, reduced $N_2O$ emission intensities in the clover parcel.

## 2 Material and methods

### 2.1 Site description

The experiment was set up at the Swiss FluxNet site Chamau (CH-Cha), located in the valley of the Reuss river on the Swiss plateau, approximately 30 km southwest of Zurich (47°12′36.8″ N 8°24′37.6″ E, 393 m a.s.l.). The site has been well investigated in terms of $CO_2$ exchange (Burri et al., 2014; Zeeman et al., 2010), as well as for $N_2O$ and $CH_4$ exchange under management that is typical for Swiss grasslands located on the Swiss Plateau (Imer et al., 2013; Merbold et al., 2014; Wolf et al., 2015). Two grassland parcels of 2.2 and 2.7 ha, are located adjacent to each other and have a similar management

history, i.e. permanent grassland since at least 2002 with a restoration year in 2012 (Merbold et al., 2014). The most abundant species are English ryegrass (*Lolium perenne*) (a mixture of early and late varieties), common meadow-grass (*Poa pratensis*), red fescue (*Festuca rubra*), timothy (*Phleum pratense*), white clover (*Trifolium repens;* small leaf varieties PEPSI, HEBE and big leaf varieties FIONA, BOMBUS), red clover (*Trifolium pratense;* variety BONUS) sown in 2012, complemented by the volunteer species dandelion (*Taraxacum officinale*) and rough meadow-grass (*Poa trivialis*). Each

parcel is usually mown four to six times per year for silage or hay production (Table 1). Each harvest is commonly followed by a fertilizer amendment, predominantly in the form of liquid slurry (average ± SD over 11 years (2003–2014) 266 ± 75 kg N ha$^{-1}$ yr$^{-1}$).

The meteorological conditions at the site are characterized by an average annual temperature of 9.1 °C and an average annual precipitation sum of 1151 mm (Sieber et al., 2011). The soil is a gleysol/cambisol, with bulk densities in 0-0.2 m depth

ranging between 0.9 and 1.3 g cm$^{-3}$ (Roth, 2006) and a soil pH of about 6.5 (Labor Ins AG, Kerzers, Switzerland, in 2014).

### 2.2 Experimental setup and management activities

The field experiment comprised a control and a clover treatment parcel (Figure 1). The control parcel was managed similarly to previous years, including the common management activities described above (harvest, fertilizer application and occasional grazing, Table 1). The eddy covariance tower, including meteorological sensors, was located at the border

between the two parcels (Figure 1). In order to test the $N_2O$ mitigation option, the treatment parcel was over-sown in March 2015 and April 2016 with clover (*Trifolium pratense* L. and *Trifolium repens* L.) to increase the clover proportion of the




sward. In contrast to the control parcel on which 296 and 181 kg N ha$^{-1}$were fertilized in 2015 and 2016, respectively (Table 1), no fertilizer was applied on the clover parcel during the experiment. To assist clover establishment and increase the clover proportion in the clover parcel, the parcel was grazed with sheep after over-sowing in mid-June and beginning of July 2015 to keep the grass species short and thus reduce competition during the clover establishment phase. The control parcel was mown once instead of being grazed during this time (beginning of July). All other harvests took place at the same day on both parcels (Table 1).

Management activities comprised the regular harvest activities (mowing) on both parcels, with subsequent slurry applications in the control parcel. Other activities were occasional grazing, plus the over-sowing of the clover parcel with *Trifolium pratense L.* and two varieties of *Trifolium repens L.*. Yields and exports of C and N were quantified by analysing biomass, sampled destructively during each harvest event (see Sect. 2.7 on vegetation samples), for C and N contents in the years 2014–2016. The fraction of N originating from BNF in the harvested biomass (2015–2016) was quantified *via* the $^{15}$N natural abundance method (Unkovich, 2008). Beyond our own observations, detailed management information for the years 2001–2016 were recorded by the farm staff in a field book. The overall amount of organic and mineral fertilizer applied to the field was documented, subsamples of the applied slurry were taken on the day of application (since 2007) and analysed in an external laboratory (LBU, Eric Schweizer AG, Thun, Switzerland). Slurry applied to the control parcel was a mixture of cattle and pig slurry after usage in a local biogas plant (for chemical composition, see Table 1). Records in the field book also included information on herbicide application, harrowing, rolling and over-sowing (for details, see Table 1).

**2.3 Greenhouse gas flux measurements**

Greenhouse gas exchange ($CO_2$, $N_2O$, $CH_4$, $H_2O$) was continuously measured at the site using the eddy covariance (EC) technique, using a mast located at the boundary between the two parcels (Fig. 1). The choice of the EC tower location resulted in the fetch lying most of the time either in one or the other parcel, taking advantage of the two prevailing wind directions. The flux measurement setup consisted of a 3-D sonic anemometer (Solent R3, Gill Instruments, Lymington, UK), an open-path infrared gas analyser for $CO_2$ and $H_2O$ concentrations (LI-7500, LiCor Biosciences, Lincoln, NE, USA) and a quantum cascade laser absorption spectrometer (QCLAS) capable to measure $N_2O$, $CH_4$ and $H_2O$ concentrations (mini-QCLAS, Aerodyne Research Inc., Billerica, MA, USA) (Merbold et al., 2014) at 10 Hz resolution. The air inlet for $N_2O$, $CH_4$ and $H_2O$ was located at a height of 2.1 m, just below the sonic anemometer head. The air was pulled through a 6 m long tube to the QCLAS located in a temperature-controlled weather proof box. Data acquisition and data storage were conducted according to the setup described in (Eugster and Plüss, 2010). From the high frequency measurements of these sensors, 10 and 30 min flux averages of the respective trace gases were calculated. The basic EC system, measuring $CO_2$ and $H_2O$ exchange, has been running since 2005 (Eugster and Zeeman, 2006; Zeeman et al., 2010) and was complemented with the field-suitable QCLAS for high frequency (10 Hz) $N_2O$ concentration measurements in 2012 (Merbold et al., 2014). Thus, more than two years of reference fluxes from both parcels under similar management regimes were collected before the beginning of the study presented here.



### 2.4 Meteorological and soil microclimate measurements

Meteorological variables measured at the Chamau site included air temperature and relative humidity (2 m height; Hydroclip S3 sensor, Rotronic AG, Switzerland), all components of the radiation balance (2 m height; CNR1, Kipp & Zonen B.V., Delft, The Netherlands), incoming and reflected photosynthetic active radiation (2 m height; PARlite sensor, Kipp and

Zonen, Delft, the Netherlands) and precipitation (1 m height; tipping bucket rain gauge model 10116, Toss GmbH, Potsdam, Germany) (Table S1). Less than two meters from the tower, basic soil microclimate measurements were carried out. These measurements included volumetric soil water content (at 0.04 and 0.15 m depth; ML2x sensors, Delta-T Devices Ltd., Cambridge, UK) and soil temperature (at 0.01, 0.02, 0.05, 0.10, and 0.15 m depth; TL107 sensors, Markasub AG, Olten, Switzerland). In addition to the sensors close to the tower, each parcel was equipped with a similar set of soil sensors in 2015

to compare potential differences in soil microclimatic conditions and subsequent effects on GHG fluxes. Soil pH (at 0.1 m depth) and soil oxygen ($O_2$) concentration (at 0.1, 0.2 m depth) were automatically measured using in-house custom-made sensors (based on ISFET pH-sensor kit, Sentron, Roden, Netherlands and EC410 Oxygen sensors, SGX Sensortech, Chelmsford, UK). In addition, soil water content (at 0.05, 0.1, 0.2, 0.5, 0.8 m soil depth; EC-5, Decagon, Pullman, WA, USA), soil temperature (at 0.05, 0.1, 0.2, 0.5, 0.8 m soil depth; T109, Campbell Scientific Inc., Logan, UT, USA), matrix

potential (at 0.1, 0.2 m soil depth; Tensiometer T8, UMS GmbH, Munich, Germany) and soil heat flux (at 0.02 m soil depth; HFP01, Hukseflux B.V., Delft, Netherlands) were recorded. Some of the soil water content sensors stopped functioning on 18th June 2015 (at 0.05, 0.1, 0.2 m) and were thus replaced on 6th August 2015 (Decagon 5TM, Pullman, WA, USA). Signals of these sensors were sampled at 10 s intervals and stored as 10 min averages on a data logger (CR1000; Campbell Scientific Inc., Logan, USA). Sensors at the tower and in its vicinity were previously connected to a CR10X model (Campbell

Scientific Inc., Logan, USA), and since March 2016 to a newer data logger (CR1000; Campbell Scientific Inc., Logan, USA).

### 2.5 Soil nutrient availability

For determining ammonium ($NH_4^+$), nitrate ($NO_3^-$) and dissolved organic carbon (DOC) concentrations in the soil, topsoil samples were taken down to 0.2 m depth. The nominally-biweekly sampling was intensified to daily intervals for seven

consecutive days following slurry application (see also Wolf et al., 2015). Five samples per parcel were taken along a transect within the footprint of the EC measurements. Extraction of $NH_4^+$, $NO_3^-$ and DOC was achieved by shaking 15 g of fresh soil with 50 mL 0.5 M $K_2SO_4$ for 1 h and subsequent filtering (Whatman no. 42 ashless filter paper, 150 mm diameter, GE Healthcare AG, Glattbrugg, Switzerland) into centrifuge tubes (50 mL tubes, PP, Greiner Bio-One GmbH, St. Gallen, Switzerland). From the extract, a subsample was acidified for the measurement of DOC by combustion in a total organic C

and N analyser (multi N/C TOC analyser 2100S, Analytik Jena AG, Jena, Germany). $NH_4^+$ and $NO_3^-$ were analysed colorimetrically (Vis v-1200, VWR International, Radnor, PA, USA). Thereafter, the remaining soil samples were dried for one week at 105 °C and weighed before and after drying in order to determine the gravimetric soil water content.




### 2.6 Vegetation sampling and determination of biological nitrogen fixation

Vegetation samples were taken from each parcel at each harvest date by destructive sampling using harvest frames (0.1 m$^2$; n = 10 for each parcel per date randomly sampled within the EC footprint, clipped at mowing height of 0.05 m, Table S1). Vegetation was separated into legumes and non-legumes (grasses and forbs) to assess the legume proportion in the dry

biomass. The only legume species found on site were the sown clover species *Trifolium pratense* L. and *Trifolium repens* L.. Vegetation samples were dried at 70 °C for one week and weighed before and after drying to estimate the water content. Milling of dry biomass samples was done separately for legumes and non-legumes, and a subsample of 5 mg was weighed into tin capsules for further analyses (n = 5 for each parcel per date). C and N concentrations, as well as $\delta^{13}$C and $\delta^{15}$N values were analysed with a Flash EA 1112 Series elemental analyser (Thermo Italy, former CE Instruments, Rhodano, Italy)

coupled to an isotope ratio mass spectrometer (DeltaplusXP, Finnigan MAT, Bremen, Germany). Estimates of BNF were based on the $\delta^{15}$N measurement. The percentage of shoot N derived from BNF (%N$_{dfa}$, nitrogen derived from atmosphere) in legume biomass was calculated with the $^{15}$N natural abundance method, (Boddey et al., 2000; Unkovich, 2008), following Eq (1):

$$\%N_{dfa} = \frac{(\delta^{15}N_{ref} - \delta^{15}N_{legume})}{(\delta^{15}N_{ref} - B)} \times 100, \tag{1}$$

where %N$_{dfa}$ is the percentage of legume shoot N derived from atmosphere, $\delta^{15}$N$_{ref}$ is the $\delta^{15}$N value of a non-fixing reference plant (i.e. grass species) growing in the proximity of the legume and $\delta^{15}$N$_{legume}$ is the $\delta^{15}$N value of the legume shoot. The B value is the $\delta^{15}$N signature of the legume species growing without N available from soil. B was estimated as the weighted mean of B values of *Trifolium repens* L. reported in the literature (–1.48 × ⅔) and *Trifolium pratense* L. (-0.94 × ⅓) (B values from Unkovich, 2008, Appendix 4). Weights were chosen according to the sown legume species composition

of ⅔ white clover and ⅓ red clover. The %N$_{dfa}$ in legume shoots was calculated for each legume biomass sample taken. The non-legumes cut within the same harvest frame as the legumes were used as reference delivering the $\delta^{15}$N$_{ref}$ value (Carlsson and Huss-Danell, 2014). For annual values, harvests and their components, uncertainty estimates were calculated with the gauss uncertainty propagation (Table 2). Vegetation development was tracked via leaf area index (LAI) measurements (LAI-2000, LiCor Biosciences, Lincoln, NE, USA) carried out on both parcels biweekly as well as before and after mowing or

grazing activities. Vegetation height and plant development as well as grazing activities within the footprint were further monitored via standard webcams (IN-5907HD, INSTAR Deutschland GmbH, Huenstetten, Germany).

### 2.7 Eddy covariance flux post-processing

Net ecosystem fluxes of CO$_2$, N$_2$O and CH$_4$ were quantified by the eddy covariance (EC) method as the covariance between turbulent fluctuations calculated by Reynolds averaging of 10-min blocks of data of vertical wind speeds and trace gas molar

densities (CO$_2$) or mixing ratios (N$_2$O, CH$_4$). Molar densities of CO$_2$ were corrected for water vapour transfer effects (Webb et al., 1980). Frequency response corrections applied to raw fluxes accounted for high-pass (Moncrieff et al., 2004) and low-





pass filtering ($CO_2$: (Horst, 1997); $N_2O$ and $CH_4$: (Fratini et al., 2012). $N_2O$ and $CH_4$ fluxes were additionally corrected for spectral losses due to instrument separation (Horst and Lenschow, 2009). All fluxes were calculated using the EddyPro software (v6.1.0, LI-COR Inc., Lincoln, NE, USA).

Before flux calculations, the statistical quality of the raw time series was checked (Vickers and Mahrt, 1997). Raw high-

frequency data used in flux calculations were rejected (1) if raw measurements were outside a physically plausible range (vertical wind speed: $\pm 5$ m s$^{-1}$; $CO_2$: 200 to 900 ppm, $N_2O$: above 250 ppb, $CH_4$: above 1700 ppb), (2) if spikes, defined as data points outside pre-defined sigma ($\sigma$) plausibility ranges (vertical wind speed: $\pm 5\sigma$, $CO_2$: $\pm 3.5\sigma$, $N_2O$ and $CH_4$: $\pm 8\sigma$), accounted for more than 1% of the respective raw time series, or (3) if more than 10% of available raw data were statistically different from the overall trend in a specific 10-min period. Raw $CO_2$ measurements were only used for flux calculations if

the window dirtiness signal from the open-path infrared gas analyser did not exceed 80% on average per 10-min data block. Half-hourly fluxes were rejected, (1) if fluxes were outside pre-defined ranges ($CO_2$: $\pm 50$ umol m$^{-2}$ s$^{-1}$; $N_2O$: between –50 and 100 nmol m$^{-2}$ s$^{-1}$; $CH_4$: between –400 and 800 nmol m$^{-2}$ s$^{-1}$), (2) if the steady state test (Foken and Wichura, 1996) was outside $\pm 30\%$, or (3) if the test on developed turbulent conditions was outside $\pm 30\%$ (Foken et al., 2004; Foken and Wichura, 1996). The analytical flux footprint model by Kljun et al. (2015) was used for footprint calculations.

The boundary between the two parcels is oriented approximately in East-West direction (75° degrees from north, Fig. 1). Each 10-min flux average was attributed to a parcel only if a minimum of 80% of the flux footprint was in the direction of the respective parcel (i.e. footprint weights from the direction of the respective parcel divided by the total of all flux footprint weights > 80%). Similar methods with EC fluxes from one setup being attributed to certain land use categories according to the respective footprint area were successfully used before (e.g. (Biermann et al., 2014; Gourlez de la Motte et al., 2018;

Neftel et al., 2008; Rogiers et al., 2005; Sintermann et al., 2011). After quality control, data coverage for $N_2O$ exchange for both years was 62% of the entire period (details in Table 3). A similar share of quality-controlled $N_2O$ fluxes was obtained from the control (48%) and the clover parcel (52%). Our aim was to analyse flux data originating from either one or the other parcel and avoid mixed GHG fluxes due to wind direction changes during the flux-averaging interval. As the standard 30-min averaging interval often resulted in mixed flux signals, we reduced the averaging period to 10 min, which resulted in

a clearer representation of the temporal dynamics of GHG fluxes from each individual parcel. On grassland systems in flat terrain (as the Chamau site), eddies with a time scale of 1–5 minutes are dominating, and thus fluxes based on a 10-min averaging interval adequately represent the atmospheric exchange of GHGs (Lenschow et al., 1994). Our comparison of flux data (full time series) based on 10 and 30 minutes averaging intervals showed that the average of 10-min $N_2O$ fluxes was only 2.3% lower than the 30-min $N_2O$ fluxes. Daily averages were calculated based on all data points per parcel that fulfilled

quality criteria 0 (best quality fluxes) or 1 (fluxes suitable for general analysis such as annual budgets) (Mauder and Foken, 2004).



### 2.8 Comparison of N₂O fluxes between parcels

We applied non-parametric bootstrapping in order to estimate the mean annual N₂O fluxes from both parcels and their respective confidence intervals. From all available 10-min fluxes, we took 1000 bootstrapping samples of each day per parcel. Averaging over time results in the bootstrapping estimate of the average annual flux, while the 0.025 and 0.975

percentiles of the bootstrapping distribution reveal the 95% confidence intervals for the mean flux per parcel.

Relative flux differences between parcels were defined as the difference of daily averages between clover and control parcels with respect to the average flux from the control, calculated based on all days for which data from both parcels were available following Eq. (2):

$$\Delta F/F = \frac{\overline{F_{Clover} - F_{Control}}}{F_{Control}} \qquad (2)$$

$F_{Clover}$ and $F_{Control}$ are daily average fluxes from the clover and the control parcels, respectively. Before being able to identify differences in N₂O exchange during the experimental periods, two years of flux data (2013 and 2014) were used to quantify how much the fluxes and the productivity from the two parcels deviated under exactly the same (2013) and similar (2014) management practice.

### 2.9 Management and rain event specific N₂O exchange

Three management types and one natural event type were analysed in more detail. These included organic fertilizer application, harvesting (mowing), sheep grazing, and rain events following dry weeks. When fertilization took place less than seven days after harvest, days after fertilization were classified as fertilization and thus not associated with the harvest event. If days after harvest overlapped with days before fertilization, these days were excluded from the fertilization class. In this case, the data displayed and analysed only refer to days after harvest but not to days before fertilization in order to avoid

misleading references. A rain event was defined with > 4 mm precipitation following a dry period with <1.5 mm collected during the 7 days preceding the rain event. When a fertilization event took place at the same time as the rain event (9th August 2015 and 16th July 2016), the event was classified as fertilization event but not as rain event. Grazing overlapped with a rain event on 15th June 2015 and 1st July 2015, thus these days were excluded from the rain event analysis. A pre-analysis was conducted for all these events, comparing N₂O emissions during seven days before the event to seven days after

the start of the event (incl. starting date). Grazing showed no significant differences between emissions before and during grazing, nor did rain events. These categories were therefore not considered in the generalized additive model (GAM, see Sect. 2.11).

### 2.10 Statistical analysis

In order to assess the influence of management and environmental drivers of N₂O fluxes, we used semi-parametric

generalized additive modelling (Wood, 2006). We expected non-linear effects of some predictor variables on N₂O emissions,





such as soil water content and oxygen concentration. The GAM model is adequate for including these non-linear effects because it prescribes no parametric relationship between predictors and response variable. Instead, the model fits smoothing splines (piecewise defined polynomials) to the relationship between each predictor and the response variable, allowing highly flexible curves if needed (i.e. if improving the goodness of fit), but resulting in the smoothest possible relationship

(i.e. linear relationship) if suitable. The response variable was predicted by the sum of all these smooth functions ("additive"). The degree of smoothing for each additive function was determined using generalized cross-validation (GCV). The response variable was the log-transformed $N_2O$ flux in order to better meet the assumptions of normally distributed residuals. The additive model with a log-transformed response corresponds to a model with multiplicative effects in the original scale. Thus, the predictors' effects influence $N_2O$ fluxes multiplicatively. The influence of management (i.e.

fertilization and harvest) and environmental driver variables (e.g. soil meteorological variables, soil chemical variables) on $N_2O$ emissions was investigated based on daily averages of measured 10-min flux data and corresponding environmental variables. For introducing management influence in the regression analysis, dates were labelled according to three *a priori* selected management categories only: post-fertilization (F), post-harvest (H) and no management (0) in combination with the treatment clover (Clo) or control (Ctr). Thus, five management categories existed (Ctr-F, Ctr-H, Ctr-0, Clo-H, Clo-0). The

control parcel without recent management activity (Ctr-0) served as the reference level in comparison to all other management categories. As grazing intensity is low at the site, and grazing did not show any influence on $N_2O$ exchange, we did not include grazing in the GAM analysis. The full set of predictors included soil temperature, soil water content, oxygen concentration, $NH_4^-$, $NO_3^+$ and DOC concentration for substrate availability, net ecosystem exchange (NEE) of $CO_2$ as a proxy for plant activity, and the categorical variable for management activity.

All predictors were included as non-linear terms in the first step, and the basic GAM was fitted using generalized cross-validation as the criterion for the parameter choice resulting in the best fit. This method resulted in several terms being included in the GAM as linear predictors (empirical degrees of freedom, edf = 1). These were finally treated as linear terms in order to obtain their effect sizes. For linear predictors such as soil temperatures, effect sizes can be interpreted as in linear regression models. Soil water content and oxygen concentration showed a non-linear influence on log-$N_2O$ emissions

(reverse U-shape), as estimated by the GAM to require more degrees of freedom (edf > 1). These were kept as (nonlinear) smooth terms in the GAM. Stepwise backward elimination was applied for model selection, whereby the number of predictors was reduced until the local minimum value of the Akaike Information Criterion (AIC) was found. Residual analysis showed that the final model residuals were in line with the assumptions of a Gaussian distributed, homoscedastic error term with a mean of zero.

Due to focusing the analysis on in situ measured data only, models that included the soil sampling variables are limited to the observational days on which manually sampled data were available (full model and optimized model). To check consistency of these results (i.e. effect sizes) with results from a wider range of observations (year-round continuous measurements) we built a model ("simple model") based on only the major driver variables soil temperatures, SWC and management as predictors, with the advantage of including more observations due to the wide coverage of these variables. Negative $N_2O$



fluxes were analysed separately, but no significant effects of the same set of predictors on $N_2O$ uptake were found. For auto-correlated time series (i.e. soil microclimatic variables) the t-test on the differences was corrected for autocorrelation by calculating the effective sample sizes according to (Wilks, 2011:147) and using these in the tests, resulting in adjusted standard errors and p values (se.adj; p.adj). All statistical analyses were performed with the open source software R (R Core Team, 2016), using the "mgcv" package (Wood, 2011) for generalized additive modelling.

## 3 Results

### 3.1 General environmental conditions

Mean annual temperatures in 2015 and 2016 were 10.3 °C and 9.7 °C, respectively (Fig. 2a). Thereby 2015 was 0.2 °C warmer and 2016 was 0.4 °C colder than the previous five years which averaged 10.1 °C. Daily photosynthetically active radiation (PAR) followed the typical seasonal pattern (Fig. 2b). Annual precipitation was 1029 mm in 2015 and 1202 mm in 2016, which is 7% lower and 9% higher, respectively (Fig. 2c), than the 5-year mean annual precipitation (1101 mm). While both years were characterized by a typical wet beginning of the growing season (MAM with 376 mm in 2015 and 379 mm in 2016), similar to the five years prior to our period of analysis, the peak growing season (JJA) in 2015 was considerably drier (260 mm precipitation) than in 2016 (396 mm, Fig. 2c). Growing season, defined by $T_{air}$ exceeding 5 °C for at least five subsequent days, started on 17th March 2015 and 30th January 2016. Starting dates of net $CO_2$ uptake for at least ten subsequent days, an alternative indicator for start of the growing season, were 27th February 2015 and 8th March 2016, similar to previous years.

### 3.2 Soil microclimate

An important precondition for the $N_2O$ mitigation experiment is to check for approximately equal soil microclimatic conditions in both parcels, i.e. to exclude the possibility that soil microclimatic variables did act as confounders in the experiment. Soil temperatures were similar in the control (mean 14.5 °C) and the clover parcel (13.6 °C) with measured differences being smaller than the sensor accuracy of ± 1°C. While air temperature fell below 0 °C, soil temperature at 0.1 m depth never fell below 0 °C during the course of the experiment (Fig. 3a). This was also the case for the two reference years 2013 and 2014. Volumetric soil water content (at 0.1 m depth) in the control (33 ± 4%) and the clover parcel (31 ± 5%). The difference between treatments was within the sensor accuracy of ± 3% (Fig. 3b). Oxygen concentration (at 0.1 m depth) ranged between 15 and 21% during three quarters of the measurement period and decreased consistently to 0% during spring in both years (Fig. 3c). Moreover, temporal patterns seen in $O_2$ concentration were not significantly different in both parcels (measured difference 0.3 ± 0.2% se.adj; p.adj = 0.075). Oxygen concentration during summer (JJA) 2015 was higher compared to 2016 (t= 2.64; p.adj = 0.03), as a consequence of less rainfall compared to summer 2016 (Fig. 2c). Soil oxygen concentration was inversely related to soil water content.





### 3.3 Soil mineral N and DOC concentration

Ammonium ($NH_4^+$) concentration in the soil peaked on each day of slurry application in the control parcel and declined during the following few days (Fig. 4a). $NH_4^+$-N concentration measured in the topsoil ranged between 0.4 and 19.2 mg $NH_4^+$-N kg$^{-1}$ dry soil in the control parcel during the two years of observations. Significantly lower $NH_4^+$-N concentration

was measured in the clover parcel (0.6–11.1 mg $NH_4^+$-N kg$^{-1}$ dry soil; paired Wilcoxon-test, $p < 0.01$). While $NH_4^+$-N concentration peaked after fertilization events in the control parcel, no distinct patterns were observed in the clover parcel where no fertilizer was applied. Soil nitrate ($NO_3^-$) concentration ranged between 1.7 and 27.7 mg $NO_3^-$-N kg$^{-1}$ dry soil in the control parcel (Fig. 4b). Similar to the observations found for $NH_4^+$-N, significantly lower soil nitrate levels (0.6–18.9 mg $NO_3^-$-N kg$^{-1}$ dry soil) were found in the clover parcel (paired Wilcoxon-test, $p < 0.01$). $NO_3^-$-N concentration significantly

increased over the course of the season in the control parcel (Mann-Kendall-test, 2015 tau = 0.50, $p < 0.001$; 2016 tau = 0.40, $p < 0.001$). Such trend was not observed in the clover parcel in 2015, while it was significant in 2016 (Mann-Kendall-test, 2015: tau = 0.15, $p > 0.05$; 2016: tau = 0.35, $p < 0.01$) (Fig. 4b). Dissolved organic carbon (DOC) measured regularly from soil samples resulted in a range of 42–234 mg C kg$^{-1}$ dry soil in the control parcel (Fig. 4c). Again, significantly lower values were measured for DOC in the clover parcel (0.6–160 mg C kg$^{-1}$ dry soil) (paired Wilcoxon-test, $p < 0.01$) compared

to the control. As observed for $NO_3^-$-N, DOC concentration significantly increased with the growing season in the control parcel in both years and in the clover parcel in 2016 (Mann-Kendall-test, control parcel 2015: tau = 0.25, $p < 0.01$, 2016: tau = 0.23, $p < 0.05$; clover parcel 2015: tau = 0.14, $p > 0.5$, 2016: tau = 0.26, $p < 0.05$) (Fig. 4bc). Overall, soil mineral N and DOC concentrations were lower in the clover parcel.

### 3.4 Sward productivity and vegetation composition

Total annual yields (mean ± SE) of the control parcel were 12.8 ± 0.6 t dry matter (DM) ha$^{-1}$ in 2015 and 11.9 ± 0.4 t DM ha$^{-1}$ in 2016, while yields of the clover parcel were 10.4 ± 0.7 t DM ha$^{-1}$ and 11.0 ± 0.5 t DM ha$^{-1}$ in 2015 and 2016, respectively (Table 2). Previous years' yields of both parcels were 9.3 ± 3.2 t DM ha$^{-1}$ yr$^{-1}$ in the control and 6.6 ± 2.3 t ha$^{-1}$ yr$^{-1}$ in the later clover parcel, based on data of all years with complete records between 2007 and 2013 (mean difference between parcels 2007–2013 of –2.7 t ha$^{-1}$ yr$^{-1}$; experiment difference 2015/16 –2.4 and –0.9 t ha$^{-1}$). Thus, yield differences between

the two parcels in 2015 and 2016 were in the range of yield differences observed during previous years, with yields being 19% (2015) and 9% (2016) lower at the clover parcel compared to the control parcel (Fig. 5a).

Average clover proportion in harvested biomass in 2015 was 14.5% in the control parcel and 21.4% in the clover parcel. The difference in clover proportion between the two parcels was more visible in 2016, with 4.1% clover proportion in the control parcel and 44.2% in the clover parcel. When analysing individual sampling dates, differences in clover proportion between

the control and clover parcel were highly variable in 2015, with substantially higher values for the clover parcel in the months April and June and slightly lower clover proportion in August when compared to the control parcel. In 2016, clover proportions increased and stabilized in the clover parcel, while they decreased in the control parcel with progress of the



growing season (Fig. 5c). Leaf area index (LAI) ranged between 0.4 and 5.9, with a maximum at the first harvest each year (Fig. 5d). Average C concentrations in the biomass of all harvests were similar across parcels and plant functional types (legumes, non-legumes, Fig. 5e). Average N concentrations in the biomass were always higher in legumes ($3.3 \pm 0.2\%$) compared to non-legumes ($2.1 \pm 0.2\%$) (Fig. 5f). C/N ratios (data not shown) of total annual yields were slightly higher in

the control ($19.2 \pm 1.7$ and $19.8 \pm 2.8$) than in the clover parcel ($17.1 \pm 1.0$ and $16.7 \pm 2.1$) for both years, respectively. Vegetation height reflected the vegetation dynamics and reached similar maxima on the control parcel (41 cm and 59 cm) and the clover parcel (44 and 60 cm) in 2015 and 2016, respectively (Fig. 5g). C in annual yields at the control parcel was $5.8 \pm 0.2$ t ha$^{-1}$ in 2015 and $4.7 \pm 0.3$ t ha$^{-1}$ in 2016, while the C in biomass at the clover parcel was lower ($5.1 \pm 0.3$ t ha$^{-1}$ yr$^{-1}$) in 2015 and similar ($4.8 \pm 0.2$ t ha$^{-1}$ yr$^{-1}$) in 2016 (Table 2). N in annual yields was higher for the control ($301 \pm 10$ kg ha$^{-1}$

yr$^{-1}$) than for the clover parcel ($264 \pm 13$ kg ha$^{-1}$ yr$^{-1}$) in 2015 (Table 2). In contrast, N exported was lower in the control parcel in the second year (control: $238 \pm 13$ kg ha$^{-1}$ yr$^{-1}$; clover: $262 \pm 8$ kg ha$^{-1}$ yr$^{-1}$) even though total biomass yields were higher in the control (Table 2). Biological nitrogen fixation *via* rhizobia associated with clover (N derived from the atmosphere – N$_{dfa}$) yielded $55.6 \pm 5.3$ kg N ha$^{-1}$ yr$^{-1}$ and $14.2 \pm 1.7$ kg N ha$^{-1}$ yr$^{-1}$ in the control parcel and $71.6 \pm 5.0$ kg N ha$^{-1}$ yr$^{-1}$ and $130 \pm 8.0$ kg N ha$^{-1}$ yr$^{-1}$ in the clover parcel during the first and the second year of the experiment, respectively

(Table 2, Fig. 5h).

### 3.5 Differences in N$_2$O exchange between control and clover parcel

Average N$_2$O fluxes (with 95% confidence interval CI from the bootstrapping given in parentheses) in the control parcel in 2015 were 4.1 kg N$_2$O-N ha$^{-1}$ yr$^{-1}$ (CI 3.8–4.2 kg N$_2$O-N ha$^{-1}$ yr$^{-1}$) and 1.9 kg N$_2$O-N ha$^{-1}$ yr$^{-1}$ (CI 1.8–2.0 kg N$_2$O-N ha$^{-1}$ yr$^{-1}$) in the clover parcel. In 2016, average N$_2$O fluxes were higher for both parcels (6.3 kg N$_2$O-N ha$^{-1}$ yr$^{-1}$, CI 6.0–6.5 kg N$_2$O-N

ha$^{-1}$ yr$^{-1}$ in the control and 3.8 kg N$_2$O-N ha$^{-1}$ yr$^{-1}$, CI 3.7–3.9 kg N$_2$O-N ha$^{-1}$ yr$^{-1}$ in the clover parcel) (Fig. 6a). Annual N$_2$O fluxes in the clover parcel were 54% (51–57% as 95% confidence intervals) and 39% (36–42%) lower than at the control parcel in 2015 and 2016, respectively (Fig. 6b). During the reference year 2013 (before the experiment), average N$_2$O fluxes in the later control parcel were 4.7 kg N$_2$O-N ha$^{-1}$ yr$^{-1}$ (4.6–4.8 kg N$_2$O-N ha$^{-1}$ yr$^{-1}$) and in the later clover parcel 4.8 kg N$_2$O-N ha$^{-1}$ yr$^{-1}$ (4.6–4.9 kg N$_2$O-N ha$^{-1}$ yr1) and did thus not differ significantly. N$_2$O emission intensities (yield-scaled N$_2$O

emissions) during the experiment were 0.31g N$_2$O-N kg$^{-1}$ DM in the control parcel and thus higher than the 0.18 g N$_2$O-N kg$^{-1}$ DM observed in the clover parcel in 2015. A similar pattern was observed in 2016, with N$_2$O emission intensities of 0.53 g N$_2$O-N kg$^{-1}$ DM versus 0.37 g N$_2$O-N kg$^{-1}$ DM in 2016 for control and clover parcel, respectively.

### 3.6 Effects of management activities on N$_2$O exchange

We observed increased N$_2$O fluxes after fertilisation in the control parcel, with maximum daily N$_2$O fluxes reaching 17.4 mg

N$_2$O-N m$^{-2}$ d$^{-1}$ on 25$^{th}$ August 2015 (Fig. S1a), a day of slurry amendment. The effect of fertilizer amendment on N$_2$O fluxes depended on the environmental conditions during and after the fertilisation event. While several events (e.g. 10$^{th}$ June 2015, 25$^{th}$ August 2015, 16$^{th}$ July 2016 and 17$^{th}$ August 2016, Fig. S1a) were followed by increased N$_2$O emissions, other events





(e.g. 1$^{st}$ June 2016) did not show such an effect (Fig. S1a, inter-quartile range displayed in Fig. 7a). N$_2$O fluxes decreased to background levels within a few (3–7) days after fertilizer application. Harvest had a moderate influence on N$_2$O emissions on both parcels (Fig. 7c). Maximum daily N$_2$O fluxes after harvest were 7.0 mg N$_2$O-N m$^{-2}$ d$^{-1}$ on 5$^{th}$ July 2016 (Fig. S1a). Average N$_2$O fluxes on both parcels were significantly higher the weeks after harvest (average of both parcels: 2.0 mg N$_2$O-

N m$^{-2}$ d$^{-1}$) compared to average fluxes during the pre-harvest weeks (1.4 mg N$_2$O-N m$^{-2}$ d$^{-1}$) (Fig. 7b). Neither grazing nor rain events significantly affected N$_2$O exchange (Fig. 7cd).

### 3.7 Influence of potential drivers on N$_2$O exchange

Nitrous oxide emissions significantly increased after fertilizer application (Ctr-F compared to Ctr-0, p < 0.05) when compared to N$_2$O fluxes during periods without management on the same (control) parcel (Fig. 8a, Table 4). The effect size

showed a 2.5-fold increase in N$_2$O emissions during the seven days following slurry amendment compared to no management (Table 4). It is important to state that the management effect exists in addition to the effect explained by the other measured driver variables, such as soil moisture, soil temperature, NH$_4^+$-N, NO$_3^-$-N and DOC concentration in the soil. After mowing no significant increase in N$_2$O emissions was found for the optimized model in either of the parcels (Table 4b). In contrast a difference in N$_2$O emissions after harvest was observed for the simple model on the control parcel (Table

4c). If the difference in sward composition itself affected N$_2$O emissions (e.g. via plant residues or rhizodeposition), we expected a significant effect of the clover treatment compared to the control during times without management (Ctr-0 which was the reference compared to Clo-0, Table 4). Due to the absence of such an effect, we deduce that the increased clover proportions at the clover parcel did not affect N$_2$O emissions.

Still, soil microclimate affected N$_2$O emissions in both parcels. Soil temperature significantly influenced N$_2$O emissions (p <

0.05), indicating a 7% (± 2%) increase in N$_2$O per °C temperature increase (p < 0.05, Table 4, Fig. 8b). Soil temperature had the highest explanatory power (r$^2$ = 0.17) for the prediction of log-transformed N$_2$O flux as a single explanatory variable (data not shown). Besides soil temperature, volumetric soil water content showed a significant non-linear effect on N$_2$O emissions (p < 0.05, Fig. 8c). The humpback-shaped functional relationship between volumetric soil water content and log-transformed N$_2$O emissions (Fig. 8c) shows an increase until 34% and a decrease above 36% volumetric soil water content.

Similarly, oxygen concentration significantly affected N$_2$O emissions (p < 0.05, Fig. 8d). Oxygen concentration was non-linearly related to N$_2$O emissions, showing lowest N$_2$O emissions (10$^{-4}$ µmol m$^{-2}$ s$^{-1}$) at 0% oxygen concentration. N$_2$O emissions increased until a maximum was reached at 17–19% oxygen concentration, and then decreased with further increasing oxygen concentration to atmospheric concentrations of 20.9% (Fig. 8d). Net ecosystem exchange of CO$_2$, which was used here as a proxy for plant activity, affected N$_2$O emissions (p < 0.05, Fig. 8e) with a 4% (± 2%) decrease of N$_2$O

emissions per µmol m$^{-2}$ s$^{-1}$ net carbon dioxide uptake. Inclusion of NH$_4^+$-N concentration improved the prediction of N$_2$O emissions (Table 4, Fig. 8f), leading to an emission increase of 5% (± 3%) per µmol m$^{-2}$ s$^{-1}$. Note that large NH$_4^+$-N concentrations only occurred after fertilization, thus the NH$_4^+$-N effect was mainly influenced by these dates, while it did not play a role for the other management categories. In contrast, NO$_3^-$-N concentration did not improve the prediction of N$_2$O




emissions (Table 4, Fig. 8g). Also, DOC concentrations showed no effect on $N_2O$ emissions (Table 4, Fig. 8h). The slopes of the relationship between drivers and predicted $N_2O$ emission are flatter than expected from visual inspection of the observed values (Fig. 8), as the predictions here depict the dependency of $N_2O$ emissions on the respective driver alone (based on averages of all other drivers), in contrast to observations, which depict combinations of effects of several drivers. The effects

of soil temperature, soil water content and management in the full and the optimized model (Tables 4a and 4b) were consistent with the simple model (Table 4c) that included only these three variables and therefore more observations (n=891 versus n=93). Including additional variables ($O_2$, $NH_4^+$-N, NEE of $CO_2$) besides soil temperature and soil water content increased the explained variance in $N_2O$ emissions from 26.3% in the simple model (Table 4c) to 54.5% in the optimized model (Table 4b).

## 10   4 Discussion

We quantified ecosystem $N_2O$ exchange at a fertilized control parcel ("business as usual") and an unfertilized clover parcel where we increased the clover proportion ("mitigation management"). The mitigation management was composed of two major changes compared to the "business as usual" practice; (1) omitted fertilization and (2) over-sowing clover, leading to an increased clover proportion in the experimental sward. Our analysis showed that the difference in $N_2O$ emissions between

both parcels can be attributed to the absence of fertilization on the clover parcel. Increased clover proportion could still have increased $N_2O$ emissions in the clover parcel due to N-rich clover residues and N from root exudates (Rochette and Janzen, 2005), and thereby offset the effect of reduced fertilization. However, we measured similar $N_2O$ fluxes originating from the two parcels of different clover proportion during periods without management, indicating that differences in clover proportion alone (i.e. excluding recent management effects) resulted in unchanged $N_2O$ emissions (i.e. plant residues and

root exudates affected $N_2O$ emissions similarly on the clover and the control parcel). We quantified the effects of environmental drivers on $N_2O$ emissions and identified soil temperature, soil oxygen concentration, soil water content and NEE of $CO_2$ as main environmental drivers of $N_2O$ emissions. The assessment of the mitigation strategy revealed reductions in $N_2O$ emissions, an increase in BNF and stable yields under mitigation management. In sum, our results indicate that $N_2O$ can be effectively reduced through the replacement of fertilizer N with N from BNF.

### 25   4.1 $N_2O$ emissions in the fertilized grassland parcel

$N_2O$ emissions in the control parcel summed up to 4.1 and 6.3 kg $N_2O$-N ha$^{-1}$ yr$^{-1}$ for the two years, respectively, corresponding to 1.3 and 3.5% of the applied fertilizer N. Annual $N_2O$ emissions are of the same order of magnitude as the values reported from the site in previous years (2010 and 2011) by (Imer et al., 2013), who estimated 2.2–7.4 kg $N_2O$-N ha$^{-1}$ yr$^{-1}$ based on manual $N_2O$ measurements using static GHG chambers. Similar $N_2O$ emissions of 4.5 kg $N_2O$-N ha$^{-1}$ yr$^{-1}$ (0.3–

18.2 kg $N_2O$-N ha$^{-1}$ yr$^{-1}$) from other fertilized grassland sites were reported by Jensen et al. (2012) in a synthesis paper covering 19 site-years. Fertilized grassland sites in Central Europe, and particularly grasslands located at higher altitudes



than our site, showed typically lower $N_2O$ emissions (0.38–2.28 kg $N_2O$-N $ha^{-1}$ $yr^{-1}$), which can be explained by lower fertilizer inputs compared to our site (Hörtnagl et al. 2018). In sum, our year-round measurements of $N_2O$ emissions are higher than multi-site averages due to its fertilizer regime and site conditions, but within plausible ranges compared to other sites.

### 4.2 $N_2O$ emissions in the unfertilized clover parcel

$N_2O$ emissions in the clover parcel during our two-year observation period summed up to 1.9 and 3.8 kg $N_2O$-N $ha^{-1}$ $yr^{-1}$ in 2015 and 2016, respectively. These values were clearly lower than the values observed from the control parcel. However, $N_2O$ emissions in the clover parcel were high compared to other unfertilized grass–clover mixtures with zero or low fertilizer inputs (< 50 kg N) for which average emissions of 0.54 kg $N_2O$-N $ha^{-1}$ $yr^{-1}$ (0.10–1.30 kg $N_2O$-N $ha^{-1}$ $yr^{-1}$) were reported by Jensen et al. (2012) based on site-years. Further non-fertilized grass-clover mixtures showed annual $N_2O$ emissions of up to 2.5 kg $N_2O$-N $ha^{-1}$ $yr^{-1}$ (Li et al. 2011, Table 5). Thus, our measurements exceeded the typical range of values in the second year by 50%. Regular N amendments at the Chamau site in the past might have led to immobilization of N via microbes and subsequent enrichment of the soil organic N (SON) pool (Conant et al., 2005; Ledgard et al., 1998). This in turn is known to lead to higher background $N_2O$ emissions in relation to $N_2O$ emissions observed from sites under long-term extensive management. In addition, high total N deposition ($NH_3$-N, $NO_3$-N, $HNO_3$-N, $NO_2$-N) on intensively managed Swiss grasslands (15–40 kg N $ha^{-1}$ $yr^{-1}$, Seitler et al., 2016) might foster background $N_2O$ emissions due to increased $NH_3$-N and $NO_3$-N availability (Butterbach-Bahl et al., 2013). Furthermore, a possible explanation for the relatively high $N_2O$ emissions from our clover parcel in 2016 were the meteorological conditions which were wetter during summer and therefore more favourable for $N_2O$ production than during 2015. High background $N_2O$ emissions in the clover parcel in 2016 were reflected by similarly high background $N_2O$ emissions in the control parcel, indicating that these were mainly driven by external factors (favourable meteorological conditions, sufficient N substrate availability) and not by the sward composition itself.

### 4.3 Effects of management and environmental drivers on $N_2O$ emissions

Our aim was to identify the main drivers of $N_2O$ emissions and therefore we investigated the effects of management (fertilization, harvest, grazing, over-sowing leading to increased clover proportion) and environmental variables on $N_2O$ emissions. Fertilization of the control parcel had the largest effect on $N_2O$ emissions. Increased N availability due to fertilization is widely known as a main driver of $N_2O$ emissions, which makes it a key factor for mitigating $N_2O$ emissions (Bouwman et al., 2002; Smith et al., 1997). Nevertheless, effects of fertilization on $N_2O$ emissions vary widely across sites and years (Flechard et al., 2007; Hörtnagl et al., 2018), indicating that fertilization alone is insufficient for explaining $N_2O$ emissions and highlighting the need to take additional drivers into account. We further observed increased $N_2O$ emissions following harvest events on the control parcel, which may be explained as a consequence of increased rhizodeposition (Bolan et al., 2004; Butenschoen et al., 2008). Subsequently, greater availability of labile C compounds can lead to increased



microbial activity, accompanied with increased production of $N_2O$ (Rudaz et al., 1999). Higher $N_2O$ fluxes following cutting were similarly observed on a pasture in Central France (Klumpp et al., 2011). Grazing had only a minor influence on the overall $N_2O$ budget of the Chamau site and data analysis showed that $N_2O$ fluxes did not significantly respond to the presence of animals (Fig. 7c). We attribute this observation to low stocking densities and short duration of grazing (Table 1).

Other studies with higher stocking densities have shown that more intensive grazing led to increased $N_2O$ emissions (van Groenigen et al., 2005; Oenema et al., 1997). These were attributed to C and N from animal excreta and to soil compaction by treading and trampling animals, creating anaerobic soil conditions (Flechard et al., 2007; Lampe et al., 2006; Oenema et al., 1997).

An important finding from this study is that increased clover proportion, and subsequently increased BNF, did not increase

$N_2O$ emissions, as shown by comparing $N_2O$ emissions between both parcels during periods without management (Table 5c, Clo-0). In other words, substrate from decomposition of plant residues and from root exudates may affect $N_2O$ emissions, but this effect was similar on both parcels, independent of the higher clover proportion and BNF in the clover parcel. This is in contrast to a study on a boreal grass-clover mixture in which significant $N_2O$ emissions were observed in spring, largely exceeding the fertilized grassland control (Virkajärvi et al., 2010). These higher emissions were explained by increased

substrate available to microbial communities producing $N_2O$ in the surface layer after spring thaw (Wagner-Riddle et al., 2008). Nitrous oxide emissions from BNF itself (rhizobial denitrification) have been shown to be possible (O'Hara and Daniel, 1985). Nevertheless, due to its small magnitude the contribution to field-scale $N_2O$ emissions is negligible (Rochette and Janzen, 2005). Previous results from a laboratory incubation by Carter and Ambus (2006), who investigated $N_2O$ emissions from unfertilized soils for up to 36 weeks, showed that recently fixed $N_2$ in a white clover-ryegrass mixture

contributed as little as $2.1 \pm 0.5\%$ to total $N_2O$ emissions. In agreement with our result, an experiment without seasonal frozen soils at an Irish permanent ryegrass/clover mixture, annual $N_2O$ emissions between unfertilized ryegrass ($2.38 \pm 0.12$ kg $N_2O$-N ha$^{-1}$ yr$^{-1}$) were not significantly different from an unfertilized grass–clover sward ($2.45 \pm 0.85$ kg $N_2O$-N ha$^{-1}$ yr$^{-1}$) with clover proportions of 20–25%, hence providing evidence that $N_2O$ emission due to BNF itself and clover residual decomposition were negligible (Li et al., 2011). Our findings are in line with these observations and add the insight that

clover proportions of up to 44%, as found in our study, will not result in increased $N_2O$ emissions.

The effects of temperature and soil water content on $N_2O$ emissions as found in our study are in line with established knowledge (Butterbach-Bahl et al., 2013; Flechard et al., 2007). Furthermore, directly measured soil oxygen concentrations, which have hardly been used in field-scale studies before, improved the prediction of $N_2O$ emissions (Table 4). Our data showed that larger plant C uptake (negative NEE) of $CO_2$ as proxy for plant activity was associated with reduced $N_2O$

emissions, which supports the hypothesis that plant roots are in competition for available N with microbes and often reduce the N availability to microbes (Merbold et al., 2014). Thus, we observed lower $N_2O$ emissions at higher levels of photosynthesis. Our analysis showed that inclusion of $NH_4^+$-N concentration in the statistical analysis improved the prediction of $N_2O$ emissions, while $NO_3^-$-N and DOC were of less importance for the prediction of $N_2O$ emissions. Comparable results for the influence of $NH_4^+$ and $NO_3^-$ were found at an Irish grassland (Rafique et al., 2012). In summary,



fertilization was the dominant predictor of $N_2O$ emissions, while soil temperature, soil water content, soil oxygen concentration and NEE of $CO_2$ were significant environmental drivers. The magnitude of the fertilization effect of 2.5-fold $N_2O$ emissions on average during the week after fertilization (at 43 kg N amendment per event on average) was comparable to the effect of a 14 °C soil temperature increment if further environmental variables remained constant. Concluding from all management effects, the decrease in annual $N_2O$ emissions under the mitigation strategy was primarily caused by the absence of fertilization, while a potential effect of the increase in clover proportion and increased BNF offsetting these emission reductions was absent.

### 4.4 Effect of the mitigation strategy on productivity and biological nitrogen fixation

An important precondition for the acceptance of any climate change mitigation strategy is that yields need to be maintained at similar levels as under conventional management. Differences in biomass yields between the control and clover parcels were only minor (19% and 9% lower in the clover parcel in 2015 and 2016, respectively), and comparable to the observed differences between the two parcels prior to the mitigation experiment (Table S2). Maintaining high yields without fertilization can be explained by the increased BNF in the clover parcel and positive interactions between clover and grass ("overyielding effect") (Lüscher et al., 2014; Nyfeler et al., 2009). Additionally, high SON content due to previous year's fertilizer amendments are expected to contribute to the persistently high production levels (Table 2). Similar productivity levels of an unfertilized grass-clover mixture (three cuts, 9% less DM) compared to an adjacent intensive grass-clover mixture (230 kg N fertilizer, 4-5 cuts) were also found at a site 50 km from the Chamau field site in the past (Ammann et al., 2009). Furthermore, our findings are consistent with findings from the more comprehensive study by Nyfeler et al. (2009), who found large overyielding effects in comparable Swiss grassland systems, i.e. grass-clover yields at 50 kg N ha$^{-1}$ yr$^{-1}$ and 50 to 70% clover were as productive as grass monocultures fertilized with 450 kg N ha$^{-1}$ yr$^{-1}$. The overyielding effect has been reported across a wide range of climates and soil types (Finn et al., 2013; Kirwan et al., 2007), indicating that our result of maintained productivity levels under the mitigation strategy is likely to be reproducible across a wider range of site conditions.

Biologically fixed nitrogen found in shoot biomass was slightly higher in the clover parcel (72 kg N ha$^{-1}$ yr$^{-1}$) compared to the control parcel (55 kg N ha$^{-1}$ yr$^{-1}$) in 2015 due to only small differences in clover proportion between both parcels. During the second year, the over-sowing was more effective and biologically fixed nitrogen found in shoot biomass in the clover parcel summed up to 130 kg N ha$^{-1}$ yr$^{-1}$ while only 14 kg N ha$^{-1}$ yr$^{-1}$ were measured in the control parcel. Previous studies reported similar amounts of biologically fixed nitrogen for mown and grazed pasture systems (Ledgard and Steele, 1992; Nyfeler et al., 2011), with maxima being as high as 323 kg N ha$^{-1}$ yr$^{-1}$ as observed in a comparable grass-clover mixture (Nyfeler et al., 2011). This indicates that biologically fixed nitrogen at the Chamau could reach higher amounts than observed during our experiment. Clover proportions at our site varied seasonally, with a minimum in spring and maximum in summer in both parcels. Such seasonal cycles in clover proportions occur due to drier conditions observed for instance in summer (JJA). Drier conditions result in competitive advantages of the clover compared to grasses, as $N_2$ fixation is less




sensitive to dry conditions than uptake of mineral N (Hofer et al., 2017; Lüscher et al., 2005). Furthermore, inter-annual variability of clover proportions can be an additional management challenge for farmers whose aim is to keep a persistent sward composition (Lüscher et al., 2014).

Lower SON content (3490 kg N ha$^{-1}$ yr$^{-1}$) in a grass-clover mixture compared to a 200 kg ha$^{-1}$ yr$^{-1}$ fertilized grassland (4350
kg N ha$^{-1}$ yr$^{-1}$) was observed after 13 years of management comparable to our experiment (Ledgard et al., 1998). It is well-known that N exports exceeding inputs lead to a decreasing SON pool. Potential losses in SON were shown to be closely linked to losses in soil organic C (SOC) (Ammann et al., 2009; Conant et al., 2005) and can therefore compromise the soil's $CO_2$ sink strength. Thus, detailed investigations on the effect of the clover treatment on SON, SOC content and $CO_2$ exchange are recommended to comprehensively evaluate the mitigation strategy in the long term.

**4.5 Effect of the mitigation strategy on $N_2O$ emissions and emission intensities**

We found that the mitigation strategy effectively reduced both $N_2O$ emissions by 54% (51–57%) and 39% (36–42%) in 2015 and 2016 as well as $N_2O$ emission intensities by 41% and 30% in 2015 and 2016, respectively. Past studies carried out in temperate grasslands consistently found reductions in $N_2O$ emissions when replacing fertilizer with N input via BNF through legumes (Table 5). The magnitude of relative $N_2O$ emission reductions ranged from 34% (Šimek et al., 2004) to 100%
(Ammann et al., 2009), with absolute $N_2O$ emission reductions of 0.8 kg N ha$^{-1}$ yr$^{-1}$ (Šimek et al., 2004) to 11.1 kg N ha$^{-1}$ yr$^{-1}$ (Schmeer et al., 2014). The variability across studies can be attributed to differences in meteorological and soil conditions as well as variations in the experimental setup (i.e. fertilizer rates applied, realized legume proportions, grass and legume species, Table 5). Nevertheless, contrasting effects to our observations were observed under boreal climate conditions in eastern Finland, with much higher $N_2O$ emissions (92% higher) from an unfertilized grass-clover mixture compared to $N_2O$
emissions in a fertilized grass sward (220 kg N ha$^{-1}$ yr$^{-1}$) due to large springtime emissions (Virkajärvi et al., 2010) indicating that the mitigation strategy is likely to be inappropriate for sites with seasonally frozen soils. Similarly, the mitigation strategy may have adverse effects in cropland, in contrast to our observations in permanent grassland, as legume cover crops were shown to increase $N_2O$ emissions following their incorporation into the soil (Basche et al., 2014; Lugato et al., 2018). Due to this effect, temporary grasslands may not reproduce the findings from permanent grassland.
In summary, the implementation of the mitigation option tested here was found to be effective at permanent grassland in the temperate zone, and is cheap and simple as it requires few management activities, which would favour farmers willingness for implementation (Vellinga et al., 2011).

**Acknowledgements**

This study was carried out within the framework of FACCE-JPI, in the Models4Pastures project with the Swiss contribution
funded by the Swiss National Science Foundation (SNSF, contract number: 40FA40_154245), which contributes to the
Global      Research      Alliance      for      Agricultural      Greenhouse      Gases      –      Integrative      Research      Group




(http://globalresearchalliance.org/research/integrative). We gratefully acknowledge the Models4Pastures project team for enriching discussions on the measured data. We greatly thank for the support provided by the head of the former ETH research station Chamau Hans-Rudolf Wettstein, and the staff on site: Ivo Widmer, Tina Stiefel and Meinrad Stalder. We are particularly grateful to Britta Jahn-Humphrey for conducting the DOC analysis and to Annika Ackermann and Roland A.

Werner for the C, N and [15]N isotope analyses. We are very thankful to Florian Käslin and Patrick Koller for technical assistance in the experimental work and oxygen sensor development. Further thanks go to our colleagues Charlotte Decock and Elisabeth Verhoeven from the Sustainable Agroecosystems research group at ETH for laboratory introductions and valuable advice at the beginning of the experiment. Valuable practical support in biomass and soil sampling as well as processing during the experiment was given by the interns Astrid Riemer and Manjunatha Chandregowda and the student

assistant Reto Zihlmann.

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



**List of Figures**

Figure 1. (a) Experimental setup and measured variables at the experimental research site Chamau (CH-Cha). The clover parcel (north) is managed to increase nitrogen inputs from the atmosphere via increased biological nitrogen fixation (BNF). This was achieved by over-sowing with clover in March 2015 and April 2016. In contrast, the control parcel under

conventional management (south) obtains most N in form of organic fertilizer (i.e. slurry) and only small N inputs via BNF. (b) Footprint climatology of the year 2016 with footprint contour lines of 10% to 90% in 10% steps using the Kljun et al. (2004) footprint model.

Figure 2. Meteorological conditions during 2015 and 2016. (a) Average daily air temperature (2 m), (b) average daily

photosynthetically active radiation (2 m). The grey bars indicate the sub-daily variability (quartiles based on 10 min values). (c) Daily precipitation sums during 2015 and 2016 (1 m).

Figure 3. Soil meteorological conditions during 2015 and 2016. (a) Average daily soil temperature (0.1 m depth), (b) average daily soil water content (0.1 m depth), (c) average daily soil oxygen concentration (0.1 m depth) at the control (left, red) and

clover parcel (right, blue). The bars indicate the sub-daily variability (ranges of 10 min values).

Figure 4. (a) Ammonium-N concentration, (b) nitrate-N concentration, (c) dissolved organic carbon concentration per unit of dry soil at the control (left, red) and clover parcel (right, blue) during 2015 and 2016. Black arrows indicate slurry applications, which only took place in the control parcel. Numbers above the arrows indicate the amount of kg N per ha

added to the parcel.

Figure 5. (a) Yields and intake by grazing at the control (left, red) and clover parcel (right, blue), (b) total aboveground biomass. Circles represent the total biomass (legumes and non-legumes), filled triangles are displaying the remaining biomass after harvest (stubble), which was measured once (sampling date 21st April 2015) and assumed to be approximately

similar during subsequent harvests. (c) Clover proportion in dry biomass, (d) leaf area index (LAI), (e) C content, and (f) N content in biomass. Diamonds represent the legumes and triangles non-legumes. (g) Vegetation heights derived from webcam images, (h) amounts of total N removal at harvest (semi-transparent), including total amount of N derived from the atmosphere in the removed biomass (saturated).

Figure 6. (a) Annual $N_2O$ exchange at control (red) and clover parcels (blue) for the reference years 2013–2014 and the experimental years 2015–2016. (b) Relative differences between $N_2O$ exchange in the control and clover parcels for the reference years (grey) and the experimental years (white). Boxes indicate the inter-quartile range based on nonparametric bootstrapping; bold black lines within boxes indicate the medians.

Figure 7. $N_2O$ fluxes (bold lines: average; color bands: inter-quartile range of daily means across all events in 2015 and 2016) in the control and the clover parcels from one week before to two weeks after management events: after (a) organic fertilizer application, (b) harvests, (c) grazing events, and (d) rain events. The black dashed line indicates the start of an event.

Figure 8. Influence of management and environmental variables on $N_2O$ emissions as predicted by the generalized additive model (GAM). Significant effects were found for (a) the factor management, (b) soil temperature (TS, 0.1 m depth), (c) soil water content (SWC, 0.1 m depth), (d) oxygen concentration ($O_2$, 0.1 m depth), (e) carbon dioxide ($CO_2$) flux and, while not significant (f) ammonium-N concentration ($NH_4$-N, 0–0.2 m depth) still improved the model (lowered the AIC). No significant influence was found for (g) nitrate-N concentration ($NO_3$-N, 0–0.2 m depth) and (h) dissolved organic carbon concentration (DOC, 0–0.2 m depth). Measurements are displayed as squares for "no management", upward triangles for harvests at the control (red) and clover (blue) parcels, and downward triangles (red) for fertilization (control). Predictions are displayed if lowering AIC as solid lines for the category "no management", as dashed lines for harvests, and as dot-dashed line for fertilization based on average values for all other drivers, respectively.

Figure S1: (a) Daily averaged $N_2O$ fluxes, (b) daily averaged $CO_2$ fluxes, and (c) daily averaged $CH_4$ fluxes at the control (left, red) and the experiment parcels (right, blue) in 2015 and 2016. Shaded areas indicate within-day variability (standard deviations of 10 min fluxes). Black downward arrows and dotted lines indicate fertilization events, upward arrows and solid lines indicate mowing, and dashed lines indicate the beginning of grazing events.

**List of Tables**

Table 1. Management activities carried out at the control and clover parcels during the experimental years 2015 and 2016 according to the field book entries of the farmer. For organic fertilizer amendments, the results of laboratory analyses (slurry composition) are given.

Table 2. Characteristics of the exported biomass from the control and clover parcels in 2015 and 2016 for legumes, non-legumes and total biomass (legumes and non-legumes). Numbers in brackets give the respective standard errors. The legume proportion is based on the annual biomass exported. C and N content and $\delta^{15}N$ values refer to mean values across all



samples. BFN refers to N derived from the atmosphere in harvested clover biomass. Means sharing the same superscript (per row) are not significantly different from each other (Tukey's HSD, $p < 0.05$); No significance tests were applied for percentages and ratios.

Table 3. Data availability of the GHG flux measurements over the two years experimental period (a) before quality assessment and quality control (QAQC) (flagged 0, 1 and 2; after Foken et al., 2004) and (b) after QAQC (acceptable quality flagged 0 and 1; after Foken et al., 2004). The reference for 100% is a year without data gaps.

Table 4. Results of generalized additive models (GAM) (a) including all variables (full model), (b) reduced after stepwise
backward elimination, dismissing DOC and nitrate (optimized model); (c) simplified including only management, soil temperature (TS) and volumetric soil water content (SWC). The control parcel without recent management (Ctr-0) was used as the reference level for the categorical variable management, thus the constant represents predictions for Ctr-0 and the effect sizes of all other management categories depict differences compared to Ctr-0. The effect sizes are displayed with their standard errors and p values for all linear terms. For the non-linear terms soil water content and oxygen concentration, the
respective empirical degrees of freedom (edf) and p values are shown. The effect sizes are direct model outputs, while the values used in the text were back-transformed to increase comprehensibility.

Table 5. Summary of studies investigating $N_2O$ emissions simultaneously in permanent grasslands of at least two different clover proportions. We included studies with > 200 days temporal coverage and at least biweekly sampling of $N_2O$
emissions, or if discontinuously sampled included a sensible strategy used by the authors in order to estimate annual fluxes.

Table S1. Details on the measurement setup including variables measured, sensor specifications, sensor locations, and measurement frequencies for continuous measurements such as (a) eddy covariance and (b) soil and meteorological sensors, as well as method, sampling frequency and locations for sampled data such as (c) vegetation and (d) soil samples.

Table S2. Annual yields 2007-2014 based on the field book entries of the farmer.



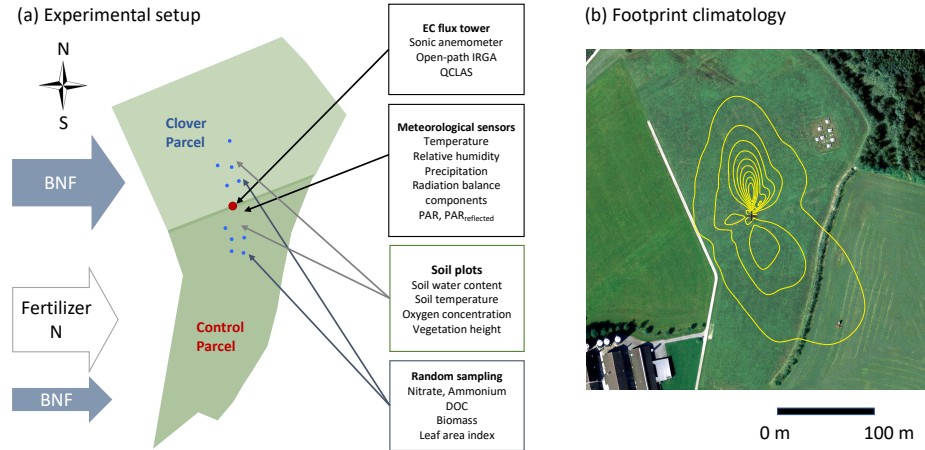

**Figure 1.** (a) Experimental setup and measured variables at the experimental research site Chamau (CH-Cha). The clover parcel (north) is managed to increase nitrogen inputs from the atmosphere via increased biological nitrogen fixation (BNF). This was achieved by over-sowing with clover in March 2015 and April 2016. In contrast, the control parcel under conventional management (south) obtains most N
5   in form of organic fertilizer (i.e. slurry) and only small N inputs via BNF. (b) Footprint climatology of the year 2016 with footprint contour lines of 10% to 90% in 10% steps using the Kljun et al. (2004) footprint model.




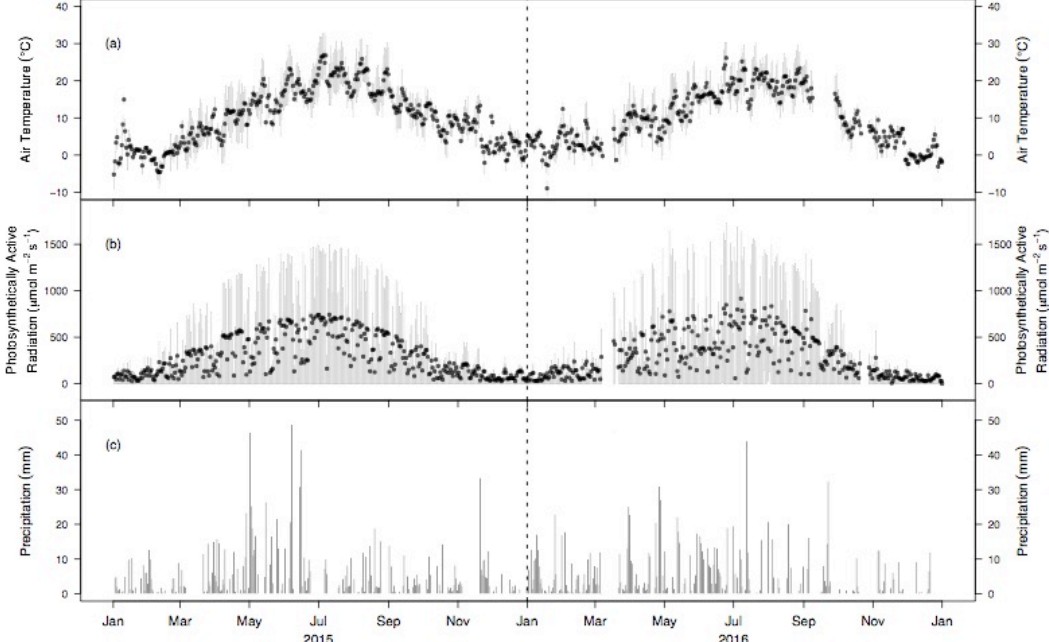

**Figure 2.** (a) Experimental setup and measured variables at the experimental research site Chamau (CH-Cha). The clover parcel (north) is managed to increase nitrogen inputs from the atmosphere via increased biological nitrogen fixation (BNF). This was achieved by over-
sowing with clover in March 2015 and April 2016. In contrast, the control parcel under conventional management (south) obtains most N in form of organic fertilizer (i.e. slurry) and only small N inputs via BNF. (b) Footprint climatology of the year 2016 with footprint contour lines of 10% to 90% in 10% steps using the Kljun et al. (2004) footprint model.





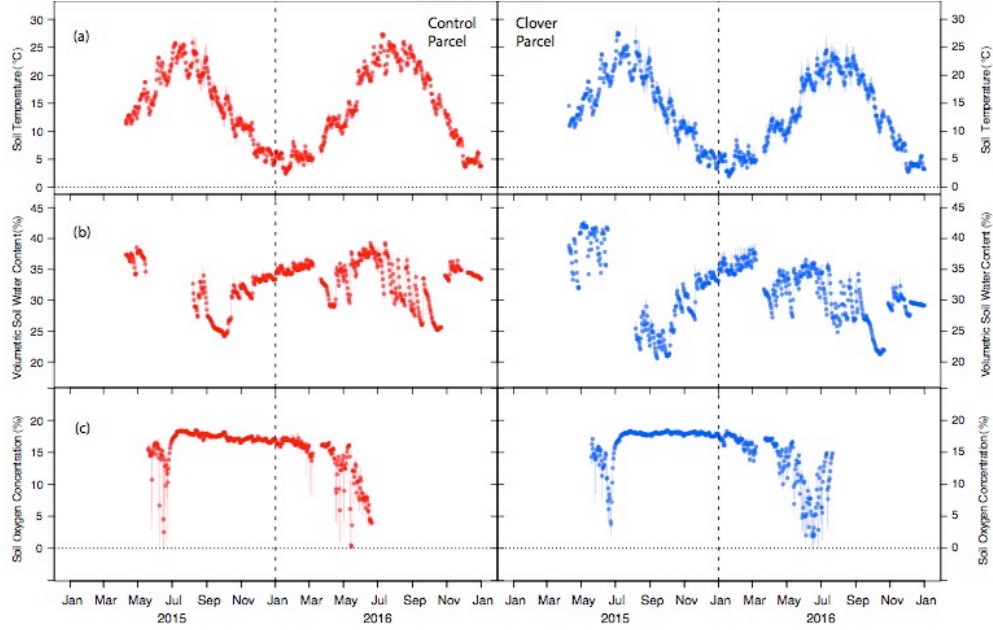

**Figure 3.** Soil meteorological conditions during 2015 and 2016. (a) Average daily soil temperature (0.1 m depth), (b) average daily soil water content (0.1 m depth), (c) average daily soil oxygen concentration (0.1 m depth) at the control (left, red) and clover parcel (right, blue). The bars indicate the sub-daily variability (ranges of 10 min values).



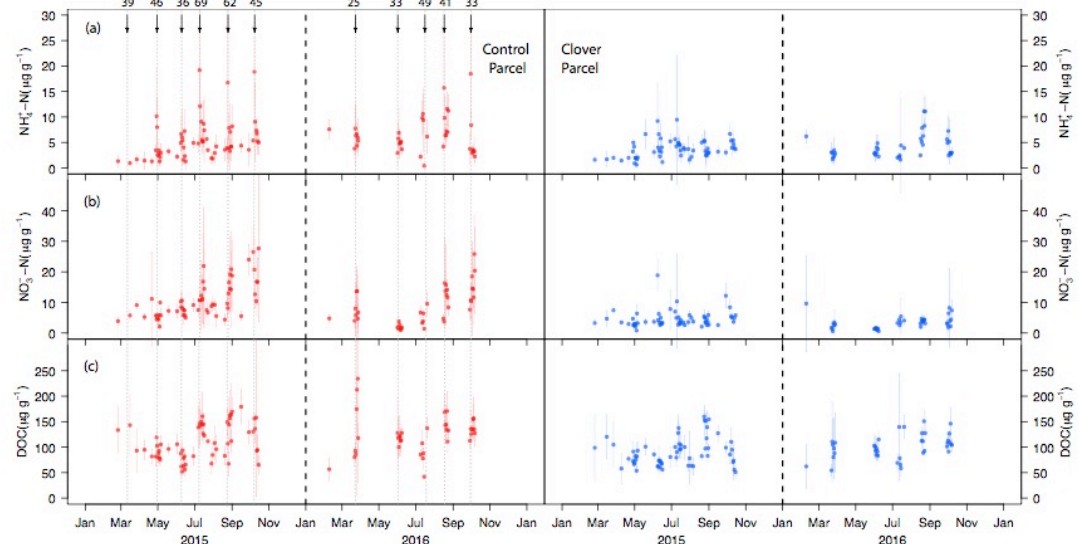

**Figure 4.** (a) Ammonium-N concentration, (b) nitrate-N concentration, (c) dissolved organic carbon concentration per unit of dry soil at the control (left, red) and clover parcel (right, blue) during 2015 and 2016. Black arrows indicate slurry applications, which only took place in the control parcel. Numbers above the arrows indicate the amount of kg N per ha added to the parcel.





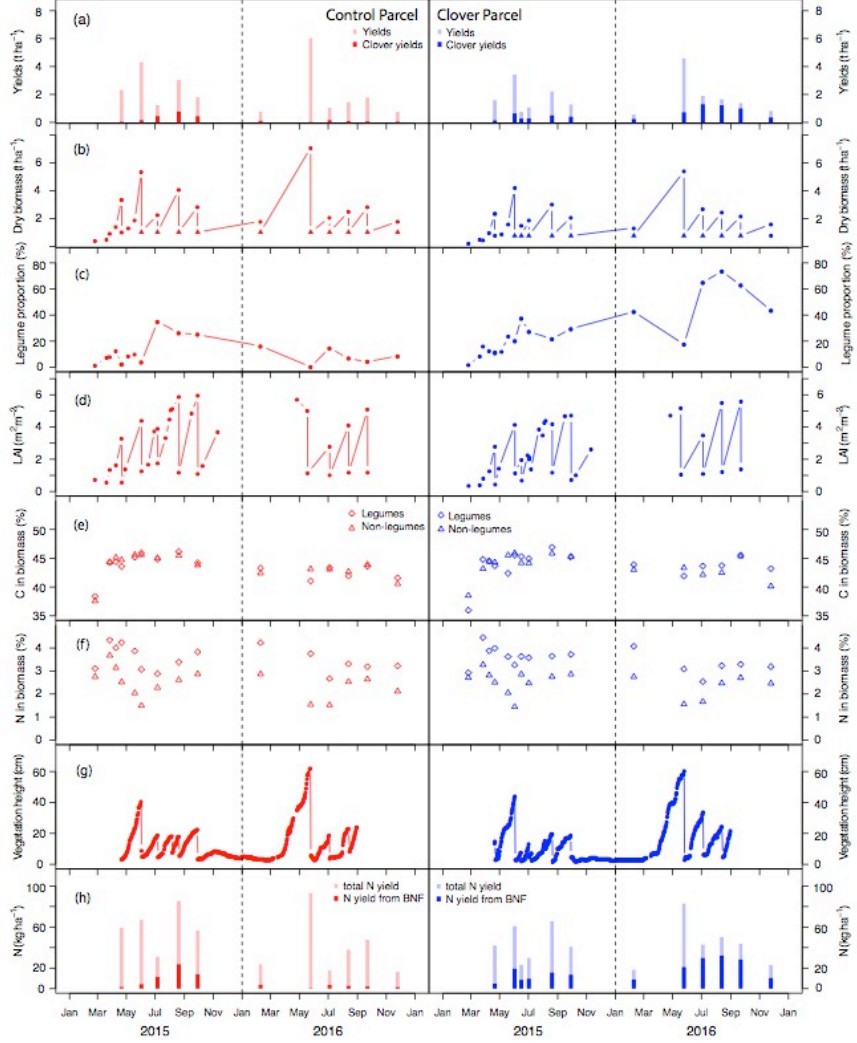

**Figure 5.** (a) Yields and intake by grazing at the control (left, red) and clover parcel (right, blue), (b) total aboveground biomass. Circles represent the total biomass (legumes and non-legumes), filled triangles are displaying the remaining biomass after harvest (stubble), which was measured once (sampling date 21st April 2015) and assumed to be approximately similar during subsequent harvests. (c) Clover proportion in dry biomass, (d) leaf area index (LAI), (e) C content, and (f) N content in biomass. Diamonds represent the legumes and triangles non-legumes. (g) Vegetation heights derived from webcam images, (h) amounts of total N removal at harvest (semi-transparent), including total amount of N derived from the atmosphere in the removed biomass (saturated).



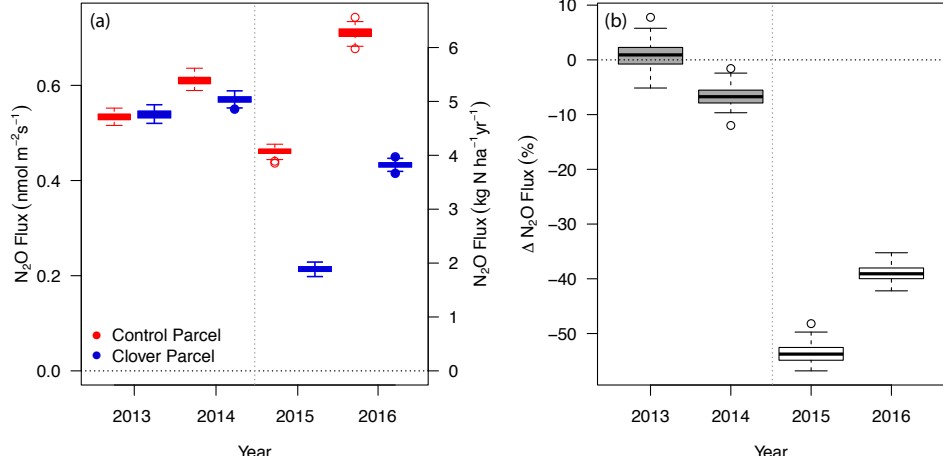

**Figure 6.** (a) Annual N$_2$O exchange at control (red) and clover parcels (blue) for the reference years 2013–2014 and the experimental years 2015–2016. (b) Relative differences between N$_2$O exchange in the control and clover parcels for the reference years (grey) and the experimental years (white). Boxes indicate the inter-quartile range based on nonparametric bootstrapping; bold black lines within boxes indicate the medians.



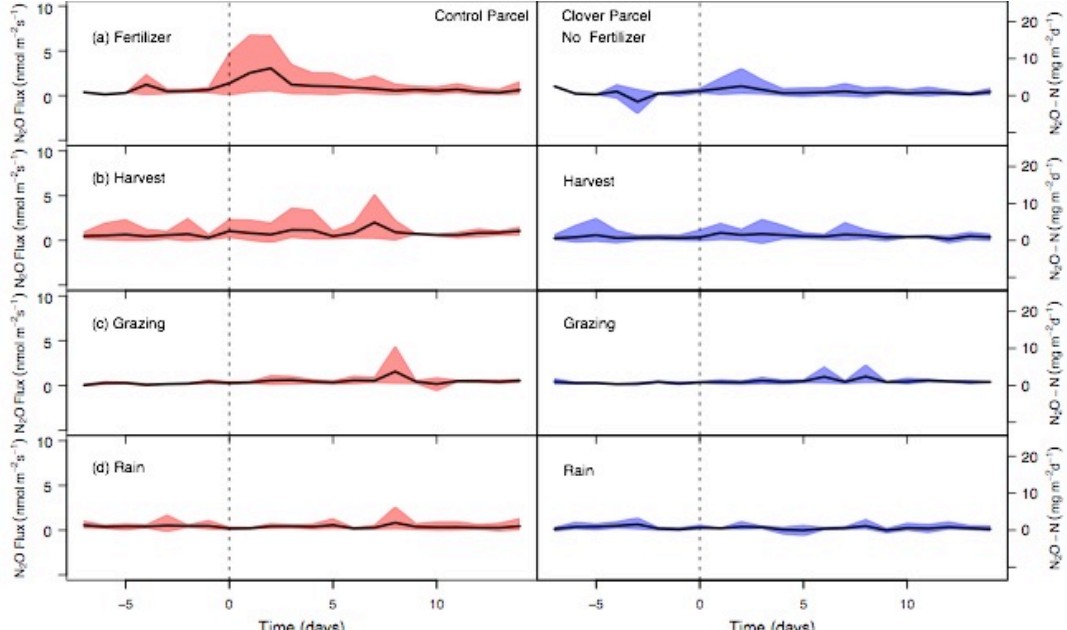

**Figure 7.** N$_2$O fluxes (bold lines: average; color bands: inter-quartile range of daily means across all events in 2015 and 2016) in the control and the clover parcels from one week before to two weeks after management events: after (a) organic fertilizer application, (b) harvests, (c) grazing events, and (d) rain events. The black dashed line indicates the start of an event.



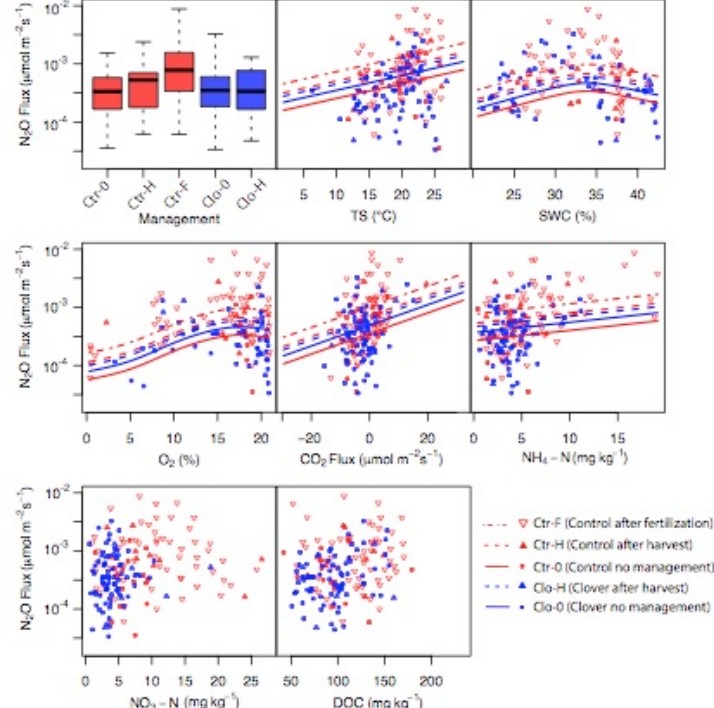

**Figure 8.** Influence of management and environmental variables on $N_2O$ emissions as predicted by the generalized additive model (GAM). Significant effects were found for (a) the factor management, (b) soil temperature (TS, 0.1 m depth), (c) soil water content (SWC, 0.1 m depth), (d) oxygen concentration ($O_2$, 0.1 m depth), (e) carbon dioxide ($CO_2$) flux and, while not significant (f) ammonium-N concentration ($NH_4$-N, 0–0.2 m depth) still improved the model (lowered the AIC). No significant influence was found for (g) nitrate-N concentration ($NO_3$-N, 0–0.2 m depth) and (h) dissolved organic carbon concentration (DOC, 0–0.2 m depth). Measurements are displayed as squares for "no management", upward triangles for harvests at the control (red) and clover (blue) parcels, and downward triangles (red) for fertilization (control). Predictions are displayed if lowering AIC as solid lines for the category "no management", as dashed lines for harvests, and as dot-dashed line for fertilization based on average values for all other drivers, respectively.




**Table 1.** Management activities carried out at the control and clover parcels during the experimental years 2015 and 2016 according to the field book entries of the farmer. For organic fertilizer amendments, the results of laboratory analyses (slurry composition) are given.

*Two varieties of Trifolium repens L., variety HEBE, FIONA, and one variety of Trifolium pratense L. TEDI; 20 kg seeds ha$^{-1}$; ⅓ of each sort, identical mixture and amounts in both years;
10 aquired from UFA Samen, fenaco Genossenschaft, Winterthur, Switzerland.

| Year | Parcel | Start | End | Management | Specification | Amount (Unit ha$^{-1}$) | Unit | Dry matter (%) | Organic matter (%) | Organic C (g kg$^{-1}$ DM) | pH | total N (g kg$^{-1}$ DM) | $NH_4$-N (g kg$^{-1}$ DM) | $NO_3$-N (g kg$^{-1}$ DM) | C/N | P (g kg$^{-1}$ DM) | $P_2O_5$-P | K | $K_2O$/K (g kg$^{-1}$ DM) | Ca | Mg | S | Total DM (kg ha$^{-1}$ DM) | Total N (kg ha$^{-1}$) |
|---|---|---|---|---|---|---|---|---|---|---|---|---|---|---|---|---|---|---|---|---|---|---|---|---|
| 2015 | Clover | 2015-03-13 | 2015-03-13 | Oversowing, rolling | Seed mixture OH HEBE, FIONA\TEDI* | 20.0 | kg | | | | | | | | | | | | | | | | | |
| | | 2015-04-21 | 2015-04-22 | Mowing, swathing, bringing in silage | Grass silage | 11.1 | dt FS | | | | | | | | | | | | | | | | | |
| | | 2015-06-02 | 2015-06-03 | Mowing, swathing, put hay on wagon | Hay | 44.4 | dt FS | | | | | | | | | | | | | | | | | |
| | | 2015-06-15 | 2015-06-19 | Grazing | Sheep | 28.1 | | | | | | | | | | | | | | | | | | |
| | | 2015-05-06 | 2015-06-30 | Drainage grubber | | | | | | | | | | | | | | | | | | | |
| | | 2015-07-01 | 2015-07-06 | Grazing | Sheep | 31.1 | | | | | | | | | | | | | | | | | | |
| | | 2015-08-30 | 2015-08-21 | Mowing, swathing, spinning, bringing in silage | Grass silage | 25.0 | dt FS | | | | | | | | | | | | | | | | | |
| | | 2015-09-28 | 2015-09-28 | Mowing, swathing, put hay on wagon | Hay | 22.2 | dt FS | | | | | | | | | | | | | | | | | |
| | Control | 2015-03-11 | 2015-03-11 | Organic fertilizer, trail hose | Slurry | 21.5 | m³ | 2.2 | 67.6 | 392.1 | 7.8 | 82.3 | 49.3 | 0.3 | 4.8 | 11.0 | 25.1 | 55.7 | 67.2 | 33.4 | 6.6 | 5.5 | 474.0 | 39.0 |
| | | 2015-04-21 | 2015-04-22 | Mowing, swathing, bringing in silage | Grass silage | 23.9 | dt FS | | | | | | | | | | | | | | | | | |
| | | 2015-04-29 | 2015-04-29 | Organic fertilizer | Slurry | 28.6 | m³ | 2.6 | 71.5 | 414.5 | 7.7 | 61.2 | 37.1 | <0.001 | 6.8 | 9.5 | 21.7 | 64.6 | 77.8 | 27.8 | 7.2 | 5.3 | 744.5 | 45.6 |
| | | 2015-06-02 | 2015-06-03 | Mowing, swathing, put hay on wagon | Hay | 50.0 | dt FS | | | | | | | | | | | | | | | | | |
| | | 2015-06-09 | 2015-06-09 | Organic fertilizer, trail hose | Slurry | 30.5 | m³ | 1.7 | 66.6 | 386.2 | 7.5 | 69.4 | 47.8 | <0.001 | 5.6 | 11.2 | 25.7 | 85.1 | 102.5 | 29.3 | 9.0 | 5.4 | 517.7 | 35.9 |
| | | 2015-06-30 | 2015-06-30 | Drainage grubber | | | | | | | | | | | | | | | | | | | |
| | | 2015-07-06 | 2015-07-06 | Mowing, swathing, spinning, bringing in silage | Grass silage | 34.1 | dt FS | | | | | | | | | | | | | | | | | |
| | | 2015-07-09 | 2015-07-09 | Organic fertilizer, trail hose | Slurry | 35.5 | m³ | 2.6 | 65.8 | 381.2 | 8.0 | 74.5 | 47.7 | <0.001 | 5.1 | 15.1 | 34.6 | 72.8 | 87.7 | 41.8 | 9.1 | 6.6 | 921.8 | 68.7 |
| | | 2015-08-20 | 2015-08-21 | Mowing, swathing, spinning, bringing in silage | Grass silage | 37.5 | dt FS | | | | | | | | | | | | | | | | | |
| | | 2015-08-25 | 2015-08-25 | Organic fertilizer, trail hose | Slurry | 27.3 | m³ | 2.6 | 65.8 | 381.5 | 8.0 | 87.9 | 55.8 | 5.7 | 4.3 | 15.6 | 35.8 | 67.2 | 81.0 | 45.8 | 8.1 | 6.3 | 709.1 | 62.3 |
| | | 2015-09-28 | 2015-09-28 | Mowing, swathing, put hay on wagon | Hay | 40.9 | dt FS | | | | | | | | | | | | | | | | | |
| | | 2015-10-08 | 2015-10-08 | Organic fertilizer, trail hose | Slurry | 28.2 | m³ | 2.0 | 63.8 | 370.1 | 8.1 | 79.0 | 45.7 | 3.5 | 4.7 | 14.5 | 33.2 | 78.2 | 94.3 | 42.8 | 8.0 | 6.5 | 563.6 | 44.5 |
| 2016 | Clover | 2016-01-26 | 2016-02-10 | Oversowing, rolling | Seed mixture OH HEBE, FIONA\TEDI* | 5.5 | kg | | | | | | | | | | | | | | | | | |
| | | 2016-04-06 | 2016-04-06 | Grazing | Hay | 20.0 | dt FS | | | | | | | | | | | | | | | | | |
| | | 2016-06-25 | 2016-05-27 | Mowing, swathing, spinning, bringing in hay | Hay | 66.7 | dt FS | | | | | | | | | | | | | | | | | |
| | | 2016-07-04 | 2016-07-04 | Mowing, swathing, spinning, bringing in silage | Grass silage | 22.2 | dt FS | | | | | | | | | | | | | | | | | |
| | | 2016-08-13 | 2016-08-14 | Mowing, swathing, spinning, bringing in silage | Grass silage | 15.3 | dt FS | | | | | | | | | | | | | | | | | |
| | | 2016-09-22 | 2016-09-24 | Mowing, swathing, spinning, silage bales | Grass silage | 20.0 | dt FS | | | | | | | | | | | | | | | | | |
| | | 2016-11-22 | 2016-11-30 | Grazing | Sheep | 5.5 | | | | | | | | | | | | | | | | | | |
| | Control | 2016-01-26 | 2016-02-10 | Grazing | Sheep | 5.5 | kg | | | | | | | | | | | | | | | | | |
| | | 2016-03-23 | 2016-03-23 | Organic fertilizer, trail hose | Slurry | 21.5 | m³ | 1.6 | 66.9 | 387.6 | 8.0 | 72.8 | 43.4 | 1.0 | 5.3 | 10.7 | 24.5 | 84.5 | 101.8 | 29.5 | 7.1 | 5.7 | 343.7 | 25.0 |
| | | 2016-05-25 | 2016-05-27 | Mowing, swathing, spinning, bringing in hay | Hay | 81.8 | dt FS | | | | | | | | | | | | | | | | | |
| | | 2016-06-01 | 2016-06-01 | Organic fertilizer, trail hose | Slurry | 25.5 | m³ | 1.6 | 65.3 | 378.8 | 8.0 | 80.6 | 45.9 | 1.5 | 4.7 | 12.4 | 28.4 | 79.9 | 96.3 | 36.0 | 8.1 | 6.9 | 407.3 | 32.8 |
| | | 2016-07-04 | 2016-07-04 | Mowing, swathing, spinning, bringing in silage | Grass silage | 23.9 | dt FS | | | | | | | | | | | | | | | | | |
| | | 2016-07-16 | 2016-07-16 | Organic fertilizer, trail hose | Slurry | 24.5 | m³ | 2.8 | 68.8 | 398.9 | 8.2 | 71.2 | 49.8 | <0.001 | 5.6 | 12.0 | 27.4 | 66.5 | 80.1 | 35.3 | 8.0 | 6.2 | 687.3 | 48.9 |
| | | 2016-08-13 | 2016-08-14 | Mowing, swathing, spinning, bringing in silage | Grass silage | 23.9 | dt FS | | | | | | | | | | | | | | | | | |
| | | 2016-08-17 | 2016-08-17 | Organic fertilizer, trail hose | Slurry | 23.2 | m³ | 1.6 | 67.0 | 388.4 | 8.0 | 110.0 | 60.3 | <0.001 | 3.5 | 13.7 | 31.4 | 72.8 | 87.7 | 42.2 | 9.2 | 6.7 | 370.9 | 40.8 |
| | | 2016-09-22 | 2016-09-23 | Mowing, swathing, spinning, silage bales | Grass silage | 24.5 | dt FS | | | | | | | | | | | | | | | | | |
| | | 2016-09-30 | 2016-09-30 | Organic fertilizer, trail hose | Slurry | 26.8 | m³ | 1.2 | 66.5 | 385.3 | 8.0 | 103.0 | 55.0 | <0.001 | 3.7 | 13.8 | 31.7 | 80.3 | 96.7 | 39.3 | 9.5 | 6.8 | 321.8 | 33.1 |
| | | 2016-11-22 | 2016-11-30 | Grazing | Sheep | 5.5 | | | | | | | | | | | | | | | | | | |





**Table 2.** Characteristics of the exported biomass from the control and clover parcels in 2015 and 2016 for legumes, non-legumes and total biomass (legumes and non-legumes). Numbers in brackets give the respective standard errors. The legume proportion is based on the annual biomass exported. C and N content and $\delta^{15}N$ values refer to mean values across all samples. BFN refers to N derived from the atmosphere in harvested clover biomass. Means sharing the same superscript (per row) are not significantly different from each other (Tukey's HSD, p < 0.05); No significance tests were applied for percentages and ratios.

| Variable (Unit) | | 2015 | | 2016 | |
| --- | --- | --- | --- | --- | --- |
| | | Control | Clover | Control | Clover |
| Biomass export (DM t ha⁻¹) | Total | 12.8 (± 0.5)[a] | 10.4 (± 0.7)[b] | 11.9 (± 0.4)[ab] | 11.0 (± 0.5)[ab] |
| Biomass export (DM kg ha⁻¹) | Legumes | 1860 (± 176)[a] | 2240 (± 141)[b] | 503 (± 80)[ab] | 4840 (± 355)[ab] |
| | Non-Legumes | 11000 (± 541)[a] | 8170 (± 666)[b] | 11400 (± 462)[a] | 6150 (± 493)[b] |
| Legume proportion (%) | Total | 15 (± 12) | 21 (± 8) | 4 (± 5) | 44 (± 20) |
| C content (%) | Legumes | 45.3 (± 1.1) | 45.6 (± 0.3) | 42.9 (± 0.9) | 43.8 (± 0.6) |
| | Non-Legumes | 45.1 (± 1.4) | 45.2 (± 0.4) | 43.0 (± 1.0) | 43.0 (± 1.0) |
| N content (%) | Legumes | 3.36 (± 0.24) | 3.56 (± 0.14) | 3.30 (± 0.14) | 3.08 (± 0.18) |
| | Non-Legumes | 2.18 (± 0.12) | 2.25 (± 0.16) | 1.94 (± 0.19) | 1.85 (± 0.17) |
| $\delta^{15}N$ (‰) | Legumes | -0.47 (± 0.54) | -0.72 (± 0.21) | -0.37 (± 0.55) | -0.76 (± 0.24) |
| | Non-Legumes | 4.77 (± 0.83) | 4.48 (± 0.42) | 5.10 (± 0.94) | 3.45 (± 0.55) |
| C (kg ha⁻¹) | Total | 5780 (± 222)[a] | 4720 (± 289)[b] | 5120 (± 221)[ab] | 4760 (± 228)[b] |
| | Legumes | 843 (± 78)[a] | 1020 (± 70)[a] | 216 (± 24) | 2120 (± 123) |
| | Non-Legumes | 4940 (± 235)[a] | 3700 (± 295) | 4900 (± 220)[a] | 2640 (± 275) |
| N (kg ha⁻¹) | Total | 301 (± 10)[a] | 264 (± 13)[b] | 238 (± 13)[ab] | 262 (± 8)[b] |
| | Legumes | 63 (± 6)[a] | 80 (± 5)[a] | 17 (± 2) | 149 (± 9) |
| | Non-Legumes | 238 (± 9)[a] | 184 (± 13)[a] | 221 (± 11)[a] | 113 (± 9)[a] |
| BFN (kg ha⁻¹) | Legumes | 55 (± 5)[a] | 72 (± 5)[a] | 14 (± 2) | 130 (± 8) |





**Table 3.** Data availability of the GHG flux measurements over the two years experimental period (a) before quality assessment and quality control (QAQC) (flagged 0, 1 and 2; after Foken et al., 2004) and (b) after QAQC (acceptable quality flagged 0 and 1; after Foken et al., 2004). The reference for 100% is a year without data gaps.

| (a) | | Acquired measurement hours before QAQC (h) | | | Data coverage before QAQC (%) | | |
|---|---|---|---|---|---|---|---|
| | | $CO_2$ Flux | $N_2O$ Flux | $CH_4$ Flux | $CO_2$ Flux | $N_2O$ Flux | $CH_4$ Flux |
| | Both Parcels | 6958 | 7969 | 7964 | 79 | 91 | 91 |
| 2015 | Control Parcel | 4089 | 4826 | 4823 | 47 | 55 | 55 |
| | Clover Parcel | 2869 | 3143 | 3141 | 33 | 36 | 36 |
| | Both Parcels | 7456 | 7734 | 7734 | 85 | 88 | 88 |
| 2016 | Control Parcel | 3911 | 4485 | 4485 | 45 | 51 | 51 |
| | Clover Parcel | 2302 | 2518 | 2518 | 26 | 29 | 29 |
| (b) | | Acquired measurement hours after QAQC (h) | | | Data coverage after QAQC (%) | | |
| | | $CO_2$ Flux | $N_2O$ Flux | $CH_4$ Flux | $CO_2$ Flux | $N_2O$ Flux | $CH_4$ Flux |
| | Both Parcels | 4930 | 5984 | 5223 | 56 | 68 | 60 |
| 2015 | Control Parcel | 1418 | 2120 | 1837 | 16 | 24 | 21 |
| | Clover Parcel | 2298 | 2395 | 2091 | 26 | 27 | 24 |
| | Both Parcels | 3787 | 5040 | 4250 | 43 | 58 | 49 |
| 2016 | Control Parcel | 1081 | 1895 | 1581 | 12 | 22 | 18 |
| | Clover Parcel | 1548 | 1921 | 1615 | 18 | 22 | 18 |





**Table 4.** Results of generalized additive models (GAM) (a) including all variables (full model), (b) reduced after stepwise backward elimination, dismissing DOC and nitrate (optimized model); (c) simplified including only management, soil temperature (TS) and volumetric soil water content (SWC). The control parcel without recent management (Ctr-0) was used as the reference level for the categorical variable management, thus the constant represents predictions for Ctr-0 and the effect sizes of all other management categories depict differences compared to Ctr-0. The effect sizes are displayed with their standard errors and p values for all linear terms. For the non-linear terms soil water content and oxygen concentration, the respective empirical degrees of freedom (edf) and p values are shown. The effect sizes are direct model outputs, while the values used in the text were back-transformed to increase comprehensibility.

| Dependent variable: log $N_2O$ Flux | (a) full model | | (b) optimized model | | (c) simple model | |
|---|---|---|---|---|---|---|
| Covariates | effect size (± se) | p-value | effect size (± se) | p-value | effect size (± se) | p-value |
| Parametric coefficients: | | | | | | |
| Control after mowing | 0.30 (± 0.24) | 0.223 | 0.13 (± 0.22) | 0.567 | 0.17 (± 0.07) | 0.012* |
| Control after fertilization | 0.46 (± 0.19) | 0.016* | 0.40 (± 0.17) | 0.025* | 0.31 (± 0.06) | <0.0001*** |
| Clover no management | 0.14 (± 0.18) | 0.432 | 0.11 (± 0.18) | 0.529 | -0.02 (± 0.03) | 0.567 |
| Clover after mowing | 0.24 (± 0.22) | 0.269 | 0.20 (± 0.22) | 0.359 | 0.10 (± 0.07) | 0.129 |
| TS (°C) | 0.03 (± 0.01) | 0.023* | 0.03 (± 0.01) | 0.004** | 0.03 (± 0.002) | <0.0001*** |
| $CO_2$ Flux ($\mu$ mol m$^{-2}$ s$^{-1}$) | 0.02 (± 0.01) | 0.018* | 0.02 (± 0.01) | 0.025* | | |
| $NH_4$-N ($\mu$ g g$^{-1}$) | 0.02 (± 0.01) | 0.167 | 0.02 (± 0.01) | 0.074 | | |
| $NO_3$-N ($\mu$ g g$^{-1}$) | -0.01 (± 0.01) | 0.231 | | | | |
| DOC ($\mu$ g g$^{-1}$) | 0.002 (± 0.001) | 0.303 | | | | |
| Constant | -4.22 (± 0.25) | <0.0001*** | -4.17 (± 0.23) | <0.0001*** | -3.97 (± 0.04) | <0.0001*** |
| Approximate significance of smooth terms: | | | | | | |
| | edf | p-value | edf | p-value | edf | p-value |
| SWC (%) | 2.33 | 0.119 | 1.87 | 0.048* | 1.98 | <0.0001 *** |
| $O_2$ (%) | 2.81 | 0.0001*** | 2.72 | 0.0003*** | | |
| Observations | 90 | | 93 | | 891 | |
| Adjusted r$^2$ | 53.5% | | 54.5% | | 26.3% | |
| Explained deviance | 60.9% | | 60.2% | | 26.9% | |
| GCV score | 0.1183 | | 0.1152 | | 0.1761 | |

*p<0.05 **p<0.01 ***p<0.001



**Table 5**. Summary of studies investigating N₂O emissions simultaneously in permanent grasslands of at least two different clover proportions. We included studies with > 200 days temporal coverage and at least biweekly sampling of N₂O emissions, or if discontinuously sampled included a sensible strategy used by the authors in order to estimate annual fluxes.

| Source | Treatment | $N_{fert}$ (kg N ha⁻¹ yr⁻¹) | Clover % | N₂O (kg N₂O-N ha⁻¹ yr⁻¹) |
|---|---|---|---|---|
| Ammann et al. 2009 | low clover | 230 | 21 | 1.60 |
| Ammann et al. 2009 | high clover | 0 | 32 | -0.10 |
| Jensen et al. 2012 | fertilized pasture | NA | 0 | 4.49 |
| Jensen et al. 2012 | unfertilized grass | 0 | 0 | 1.20 |
| Jensen et al. 2012 | grass-clover | 0 | NA | 0.54 |
| Jensen et al. 2012 | pure clover | 0 | 100 | 0.79 |
| Klumpp et al. 2012 | low clover | 157 | 19 | 1.72 |
| Klumpp et al. 2012 | high clover | 157 | 35 | 1.52 |
| Li et al. 2011 | rhyegrass grazed | 226 | 0 | 7.82 |
| Li et al. 2011 | fertilized rhyegrass-white clover grazed | 58 | 20-25 | 6.35 |
| Li et al. 2011 | unfertilized rhyegrass-white clover grazed | 0 | 20-25 | 6.54 |
| Li et al. 2011 | rhyegrass-background | 0 | 0 | 2.38 |
| Li et al. 2011 | grass-clover background | 0 | 20-25 | 2.45 |
| Ruz-Jerez et al. 1994 | low clover | 400 | NA | 5.20 |
| Ruz-Jerez et al. 1994 | high clover | 0 | NA | 1.30 |
| Schmeer et al. 2014 | uncompacted grass | 360 | 15 | 8.74 |
| Schmeer et al. 2014 | compacted grass | 360 | 15 | 13.31 |
| Schmeer et al. 2014 | uncompacted lucerne-grass | 0 | 70 | 2.46 |
| Schmeer et al. 2014 | compacted lucerne-grass | 0 | 70 | 2.22 |
| Simek et al. 2004 | no clover | 210 | 0 | 2.28 |
| Simek et al. 2004 | high clover | 20 | 60 | 1.50 |
| Simek et al. 2004 | pure clover | 20 | 100 | 1.50 |
| This study 2015 | low clover | 296 | 15 | 3.93 |
| This study 2016 | low clover | 181 | 4 | 6.27 |
| This study 2015 | high clover | 0 | 21 | 1.89 |
| This study 2016 | high clover | 0 | 44 | 4.07 |
| Virkajärvi et al. 2010 | no clover | 220 | 0 | 3.65 |
| Virkajärvi et al. 2010 | high clover | 0 | 75 | 7.00 |