# Peer review of "Management matters: Testing a mitigation strategy for nitrous oxide emissions using legumes on intensively managed grassland"

_Biogeosciences, 2018_

## Referee Comment (RC1) · Anonymous Referee #1 · 1 Jun 2018

General Comments The manuscript by Fuchs et al. is well written and easily to follow. The authors report on a 2-year field study of eddy covariance N2O flux measurements on two side-by-side grasslands, one managed 'business as usual' (i.e. with frequent additions of organic fertilizer in the form of slurry) and the other with increased proportion of clover and no slurry application (i.e. nitrogen provided by biological fixation instead of organic fertilizer). The authors find that absence of fertilization in the field with increased clover resulted in significant reduction in N2O emissions.

I agree with the authors' justification of the lack of studies on year-round N2O fluxes for grassland systems. The flux measurement methodology used is sound (but see

questions below) and the authors have collected a very complete set of supporting measurements to help in the interpretation of fluxes. The study contributes a solid dataset that could be valuable for future modelling efforts. I also liked the use of the GAM models to attribute the N2O flux to various covariates.

My main difficulty with the manuscript is the larger context of practicality of the management practice studied. The authors use the term 'fertilizer nitrogen' and 'fertilization' throughout but as far as I understand their experimental setup refers to substituting livestock-derived organic nitrogen with biologically fixed organic nitrogen. Substituting an external input such as synthetic fertilizer with N from biological fixation makes a lot of sense but presumably the dairy/pig farms have slurry that contains N and other nutrients to be recycled back to the soil. The authors used slurry that had been digested in a biogas plant so what will be the fate of this material if not returned to soil? Hence, the proposed mitigation management does not fit within a larger nutrient balance framework: grasslands are producing animal feed but the manure is not returned to fields and instead more nitrogen is added to fields through biological fixation. Perhaps there is some local context that would justify the proposed mitigation practice. If not, I am having a bit of difficulty in identifying the value of the research findings. Overall, I would like to see the dataset published but I think the argument for the value of the findings has to be much better articulated. Perhaps placing their measurements within an N budget framework would help making the manuscript a unique contribution.

Specific comments: Title: Use of legumes is a central theme of the manuscript but does not appear in the title. 'Management matters' is catchy but also well-known and a bit vague since it does not specifically identify which management.

Abstract L. 11: I think the suggested mitigation strategy is for replacing synthetic fertilizer with BNF not animal-derived N with BNF. Here and throughout need to identify if referring to organic or synthetic fertilizer. Check the consistent use of BNF vs. BFN throughout. L. 14: Could a broader objective statement be used here (i.e. quantify is a step in trying to answer a more meaningful objective); L. 17: 'To assess the effect of

the mitigation strategy' on what? L. 18: No results of the 15N method are presented in abstract leaving the reader wondering why it is mentioned here. L. 21: Here and throughout, the authors use 'no management' which is not a very clear term. I think the authors mean during background or baseline emission periods (i.e. outside of emission events associated with management). Either the term is defined early on and used or different wording should be used. L. 24-25: Did the overall N input also decrease in the clover treatment? An overall decrease in N input is different than the argument of 'replacing fertilizer N with BNF' used at the start of the abstract. It is not surprising that N2O emissions are lower if N inputs decreased.

Page 2 L. 13-14: Instruments for EC measurements have been available for a least a decade. L. 17: 'abiotic factors are generally known' L. 18-19: N2O emissions strongly depend on management practices in most managed systems so I suggest to remove 'particularly grasslands'. L. 31: Here the authors refer to 'external fertilizer amendments' but slurry would not be considered an external amendment.

Page 3: L. 3-5: This statement seems a bit misleading given that this study is not dealing with synthetic fertilizer. L. 33: What is the reason for substituting the N fertilizer from organic origin that is already available on the farm?

Page 4: L. 3: Could the second objective be made more specific? 'to identify the drivers of N2O emissions' is a bit broad. L. 21: Is this 'dairy' liquid slurry? Why 'predominantly'? Table 1 refers to 'organic fertilizer': was there another source beside slurry? Was the aim to replace all the ∼260 kg N/ha with BNF? Table 2 shows a maximum of 130 kg N/ha from BNF. What is the recommended N input to sustain the N removed? L. 24-25: Where there any differences in texture, C content, etc. in the two fields monitored?

Page 5: L. 3: Was the control plot also grazed? If not, what are the implications of the different grazing regimes? L. 16: better to say 'digested cattle and pig slurry obtained from a local biogas plant'. If this digestate is not applied to land what is its fate?

Page 6: L. 7-8: Any replicates or just one sensor per depth? L. 9: How many locations of the 2 ha plots were monitored to determine impacts on microclimate?

Page 8: L. 3: How well does EddyPro work for processing N2O data? How were the lag times determined given that low N2O signal often means the cross correlation method for determining lag times will be off. The same applies to the corrections - if cospectra are being used for any of the corrections they may not be correct for periods with low N2O or CH4 signals. L. 16: Are there any biases associated with S or N flow being associated with different kinds of weather (rain occurrence, higher T, etc.)? Please expand on this potential impact. L. 24: Were directions always steady over this time period or did you use high frequency data to select the 10 min periods with steady wind direction?

Page 10: L. 34: Could more information be given on 'management'? Which 'Management' aspects? These are not quantitative variables or continuous so how was this handled?

Page 12: L. 22: Odd wording: 'later clover parcel'; please edit. Was there a trend for lower yields for the 'clover' parcel before the experiment started? Is this related to differences in soil? L. 24: Was difference consistently <0 for all years? What are the implications for previous C input to soil and N2O fluxes during measurement period?

Page 14: L. 15-16: This interpretation assumes that there was no interaction between factors, i.e. all effects during non-events, called 'times without management' by authors, are due to sward composition while all effects during events was due to slurry application. Please discuss potential interaction effects.

Page 15: L. 23-24: 'In sum, our results indicate that N2O can be effectively reduced through the replacement of fertilizer N with N from BNF.' But was all slurry N replaced by BNF? Did the system have an N surplus before that could have been addressed by just matching the Slurry N input to crop needs? IF that is the case then the most achievable mitigation practice would be to adjust the slurry rates to the plant demand

instead of using BNF.

Page 17: L. 31: 'Thus, we observed lower N2O emissions at higher levels of photo-synthesis'. Higher levels of photosynthesis also correspond to periods of higher water loss as transpiration and lower water content as seen in your dataset. How can you separate these two confounding effects?

Page 19: L. 24: This brings up an important point. 'Permanent grasslands' are temporarily restored as was done with the study area in 2012. Does this restoration involve ploughing or a similar tillage at this site? If so, then authors should discuss the impact of incorporating a larger proportion of legumes into soil on the following N2O emissions.

L. 25-26: But what would farmers do with the slurry?

Technical corrections: Page 4: L. 5-6: Word missing here: 'while fertilization to play the dominant role in driving N2O emissions in the control parcel'. Page 5: L. 7: Mowing is only one of the activities used in harvesting (presumably biomass is also removed). I think it is less confusing if 'harvest' is used throughout. L. 8-9: Some of this information is already given on L 31 in previous page. L. 11: Please clarify the study years (2015-2016, correct). Are the data for 2014 presented? L. 17: Check table as 'herbicide' not mentioned. L. 20: I think you mean with instruments mounted on a mast... L. 21: Instead of 'lying' use 'placed'. L. 26: Instead of 'The air inlet for N2O, CH4 and H2O' use 'The air inlet for the absorption spectrometer'.

Page 6: L. 1: Should refer to Fig. 1 in this section. L. 13-15: Is this in addition to the 'soil sensors for microclimate measurements' mentioned in L. 9-10? Are matric potential and soil heat flux data presented? L. 26: Do you mean the 'average footprint'? Page 7: L. 20: Perhaps give info on clover proportion before (start of methods). The assumption is that proportion of establishment is the same as sowing composition. Did you check the final stand composition? L. 23: Use 'Gauss'. L. 29-30: CO2 molar density refers to the Licor measurements? Since CO2 was also measured with the QCLAS it may be helpful to inform the reader in EC section as to which data will be

presented for CO2. Page 8: L. 6: 'below' instead of 'above' for N2O and CH4 values. L.14: fig 1 refers to Kljun et al 2004. L. 15: "Parcel" does not seem to be the correct term here. Do you mean 'field' or 'plot'. Why is footprint so different for South vs North? Page 9: L. 13: What was the 'similar management practice' in 2014? Please explain earlier in text. L.15: I suggest 'Three management types and one natural event type' be changed to 'Three management-derived events and one natural event' Page 11: L. 3: Check citation. What does 'these' refer to? L. 23: Sentence not complete. Page 12: L. 20: Table has SE as 0.5 not 0.6. Page 13: L. 8-9: I am not sure what is being compared here (within years or across years): biomass at the clover parcel was lower (5.1 ± 0.3 t ha-1 yr-1) in 2015 and similar (4.8 ± 0.2 t ha-1 yr-1) in 2016. L. 10-11: But values are indicated as 'ab' so not different? Page 14: L. 8: This statement is confusing: 'periods without management'. L. 10-11: 'a 2.5-fold increase in N2O emissions during the seven days following slurry amendment compared to no management (Table 4).'. It is hard to find this information in table; please help reader by giving values...

L. 11: 'It is important to state that the management effect exists in addition to the effect explained' is not clear, please edit.

Page 16: L. 21: I do not consider soil conditions as not 'external' factors. . .

Page 17: L. 11: 'Clo-0' is not used in table, please use consistent terminology.

Page 19: L. 24: This brings up an important point. 'Permanent grasslands' are temporarily restored as was done with the study area in 2012. Does this restoration involve ploughing or a similar tillage at this site? If so, then authors should discuss the impact of incorporating a larger proportion of legumes into soil on the following N2O emissions.

L. 25-26: But what would farmers do with the slurry?

Figure 1: Add explanation of 'blue dots' into figure caption. Why are there many more contour lines in the footprint for the N plot vs. S plot?

Figure 2: Please indicate if symbols are for daily values.

Figure 6b: not referred to in text.

Figure 8: letter a, b, ….missing from graphs. X-axis scale for CO2 flux seems to be wider than dataset.

Table 2: Was means test applied to treatments within a year or across two years? The latter (I think) but some of the discussion in text seems to consider ab different than b or a… Letters are missing for last row and 3rd last row year 2016 data.

Table 4: Use 'harvest' instead of 'mowing'.

---

## Referee Comment (RC2) · Anonymous Referee #2 · 1 Jun 2018

This is an important paper, comparing N2O emissions from adjacent fields with different proportions of clover content. The paper is well written, and suitable for publication in BG. My main comments are as follows: 1) In the 'mitigation' treatment more clover was added, but as it took time to establish, the differences between clover % was rather small in year 2015 (15% & 21%), whereas in year 2016 the differences were large (4% and 44%). Similar differences were observed for the BNF rates (Table 2). These differences and their implications on the yield and N2O fluxes is not adequately addressed in this paper. 2) N2O was measured using eddy covariance, from 2 adjacent fields. The overall data coverage of both fields was similar (Table 3), but the authors need to demonstrate that the temporal coverage of measurements was similar for both

fields. One would not want situations where the airflow is always from field 1 at dawn, for example.

Further suggestions of edits can be found below Abstract: needs to contain the grass clover proportions for the 2 fields

Introduction: 'Apart from the environmental benefits of a reduced N surplus when mineral fertilizer is replaced by BNF, total GHG emissions from fertilizer production of 1.6–6.4 kg CO2-eq per kg fertilizer N, could technically be avoided (Andrews et al., 2007; Brentrup 5 and Pallière, 2008).' R: In which country of climate zone can such GHG reduction rates be achieved?

Methods: ''The site has been well investigated in terms of CO2 exchange (Burri et al., 2014; Zeeman et al., 2010), as well as for N2O and CH4 exchange under management that is typical for Swiss grasslands located on the Swiss Plateau (Imer et al., 2013; Merbold et al., 2014; Wolf et al., 2015).' R: Add that CO2 exchange is measured by EC and N2O/CH4 by , presumably, static chambers.

R: Given that you have reduced the EC averaging time to 10 min from the usual 30 min, I assume that you must have had a relatively equal spread between coverage of both the two plots. You need to demonstrate this, for example by including a graph of N2O versus time with different colour dots for the two treatments.

R: Why did you fertilise with 296 kg N/ha/2015 and 181 in 2016?

Section 2.6: 'and a subsample of 5 mg was weighed into tin capsules for further analyses (n = 5 for each parcel per date).' R: You need to add: '. . .for further analysis of total C and N and . . .. . ..'

R: Figure 1: include the prevailing wind direction, or say what it is in the legend

R: Legend to Figure 2 needs to be tidied up.

R: Figure 5: Why do you join the dots for graphs b-d, but not for graphs e-f? Looks like

there is some inconsistency here.

R: Section 3.5: 'During the reference year 2013' Add the reference to these 2013 data

R: Figure 8 'the factor management' delete 'the factor', or place it before '(a)'

Discussion 1st Parag. 'major changes compared to the "business as usual" practice; (1) omitted fertilization and (2) over-sowing clover, leading to an increased clover proportion in the experimental sward' R: Add the % of clover to remind the reader ' to an increased clover proportion of x %'

R: Last sentence and elsewhere: change 'in sum' to 'in summary'.

Section 4.1: 'than our site, showed typically lower N2O emissions (0.38–2.28 kg N2O-N ha-1 yr-1), which can be explained by lower fertilizer inputs compared to our site (Hörtnagl et al. 2018). 'In sum, our year-round measurements of N2O emissions are higher than multi-site averages due to its fertilizer regime and site conditions, but within plausible ranges compared to other sites. R: Discuss the differences in fertiliser rate and the differences in site composition between the Hortnagl study and yours in greater details, so that the reader also understands why your N2O fluxes are larger. Provide more information on the differences between your site and the Hortnagl sites. And, to improve the English change 'In sum, our year-round measurements of N2O' to 'In summary, our one-year measurements of N2O'

Section 4.2: 'N2O emissions in the clover parcel during our two-year observation period summed up to 1.9 and 3.8 kg N2O-N ha-1 yr-1 in 2015 and 2016, respectively. These values were clearly lower than the values observed from the control parcel.' R: You need to discuss these observations and others in this section with the fact that the differences in clover proportions between the two fields in 2015 were rather small compared to 2016 (Table 2).

'Jensen et al. (2012) based on site-years.' R: based on how many site years?

'In addition, high total N deposition (NH3-N, NO3-N, HNO3-N, NO2-N) on intensively

managed Swiss grasslands (15–40 kg N ha-1 yr-1, Seitler et al., 2016)' R: Can you be more specific regarding the N dep rate in your study area. The range you quote is very large.

Section 4.3: 1st paragraph: you should qualify phrases such as those shown below 'N2O emissions vary widely across sites' R: add the ranges of emissions, and presumably the studies for reference are from grasslands?) 'Higher N2O fluxes following cutting were similarly observed on a pasture in Central France (Klumpp et al., 2011).' R: what is the difference relative to your study? You have done this much better in the 2nd paragraph.

'In agreement with our result, an experiment without seasonal frozen soils at an Irish permanent ryegrass/clover mixture, annual N2O emissions between unfertilized ryegrass' R: 'change to ' In agreement with our result, measurements from permanent grasslands in Ireland, where winter freeze-thaw cycles are very rare, a comparison of a ryegrass/clover mixture, with . . . . . .'

'The magnitude of the fertilization effect of 2.5-fold N2O emissions on average during the week after fertilization (at 43 kg N amendment per event on average) was comparable to the effect of a 14 °C soil temperature increment if further environmental variables remained constant.' R: This sentence requires an introduction and significant explanation. It is a bit out of place here.

Section 4.4 'Additionally, high SON content due to previous year's fertilizer amendments are expected to contribute to the persistently high production levels' R: I suppose you mean 'years' and not year's'

'the over-sowing was more effective and biologically fixed nitrogen found in shoot biomass in the clover parcel summed up to 130 kg N ha-1 yr-1 while only 14 kg N ha-1 yr-1 were measured in the control parcel' R: What is the reason for the legume proportion in the control to decrease between 2015 and 2016?

'This indicates that biologically fixed nitrogen at the Chamau could reach higher amounts than observed during our experiment.' R: Can you really deduce this statement from the New Zealand study, where the climate, soil types and perhaps even the grass and clover species used may be rather different?

Section 4.5 'due to large springtime emissions (Virkajärvi et al., 2010) indicating that the mitigation strategy is likely to be inappropriate for sites with seasonally frozen soils.' R: Your Swiss soils also experience winter freeze-thaw cycles, but your data suggest that this mitigation strategy works in Switzerland. Please address this discrepancy.

'Due to this effect, temporary grasslands may not reproduce the findings from permanent grassland.' R: You need to provide evidence for this statement. Temporary grasslands are maintained for several years, so are rather different to croplands.
* * *

---

## Referee Comment (RC3) · Anonymous Referee #3 · 5 Jun 2018

GENERAL QUESTIONS AND COMMENTS

The paper is well structured and written, it presents a comparison between two differently managed grasslands, to evaluate the impact of the addition of clover as a mitigation strategy for N2O emissions. In my opinion, it is suitable for publication in Biogeosciences. See below for some minor comments to the paper.

The focus of the paper is the evaluation of the integration of legumes as a mitigation strategy, but that does not appear explicitly in the title: I think it should be part of it.

To what extent do you think soil properties are needed (frequency of sampling, etc) to interpret flux data? Why daily or bi-weekly? Why many 20cm samples? Would you

have any specific suggestions for long term measurements? This is not necessary, of course, but I think it would add value to the discussion to know what you found out to be the most useful variable for your parameterisation, beside the interpretation of your results, of course.

Could you also report the GWP of the N2O measurements to express the mitigation induced by the grassland composition change?

The eddy covariance tower in this experiment is placed at the edge between two differently treated fields. Could you explain a bit better the way you tackled the advection issues between the two treatments/crops? Two years is an impressive duration for such dataset, and I would guess all conditions (time of the day, stability, etc.) have been met in such long time for both fields, but it would be good if it were expressed more clearly through the results. Also, for what concerns the special events, do you have a suitable coverage for both fields in terms of footprint?

BY SECTIONS:

MATERIAL AND METHODS

P5 L1> Replace "fertilised" with "added".

Section 2.2:

What is the reason to use two different fertilisation rates in the 2 years? Is it for simulating the business as usual behaviour of the farmer, or did you increase the amount to enhance the effects of the contrast? I see that the clover abundance difference between the two years is quite relevant: is it solely due to the additional grazing? Was the field over sown at the same rates? In connection to the conclusion that up to 44% of clover addition does not lead to further N2O emissions, it could be useful to suggest how to achieve such abundances. Perhaps you could expand on this. Section 2.7: could you specify and motivate what method you used to calculate the time lag for the different GHG species?

**RESULTS**

P12 L6-7. From figure 4, it almost looks like this is not the case in the last 2 events of 2016, "41,33". The value of NH4+ after these events seems to be almost double compared to the period before. Could you comment on this, especially addressing the N deposition issue? P13,L2: it could be helpful to quantify this similarity, e.g. providing a ratio of C content in biomass between the 2 different treatments directly in the text.

**DISCUSSION**

P17,L3-4. "Grazing had only a minor influence on the overall N2O budget of the Chamau site and data analysis showed that N2O fluxes did not significantly respond to the presence of animals (Fig.7c)". I think that a quantification would be better here than referring to the plot only, i.e. the relative % in contribution on the total N2O budget, for example.

**TABLES AND FIGURES**

Figures 1 and 6 seem to have the appropriate resolution, the others tend to be a bit blurry: would it be possible to increase the image resolution?

Figure 2. The caption under the image needs correction. "(b) Footprint climatology of the year 2016 with footprint contour lines of 10% to 90% in 10% steps using the Kljun et al. (2004) footprint model" belongs to previous figure; explain panels a, b, and c.

Figure 4. Albeit the treatments with slurry were not applied on the clover field, I think it would be useful to introduce the days of treatment also on the North field charts (perhaps the same dotted lines, no arrows). If no slurry was directly applied, the amount of N in the air during the fertilisation events has certainly changed, and potentially increased the amount of BNF on the clover field.

[Figure]

---

## Author Comment (AC1) · 28 Jun 2018

We want to thank all three anonymous reviewers for the multiple helpful comments. We think that these clearly helped to improve the manuscript. Please find the detailed response in the attached document.

Please also note the supplement to this comment: https://www.biogeosciences-discuss.net/bg-2018-192/bg-2018-192-AC1-supplement.pdf

---

## Author Comment (AC3) · 28 Jun 2018

Authors' Response to **Referee #1- Referee #2 and Referee #3**

Contact Author: Kathrin Fuchs: kafuchs@ethz.ch

Reviewer comments are written in blue

Authors' comments are written in black letters.

**Anonymous Referee #1**

**Referee #1** General Comments

R: The manuscript by Fuchs et al. is well written and easily to follow. The authors report on a 2-year field study of eddy covariance N2O flux measurements on two side-by-side grasslands, one managed 'business as usual' (i.e. with frequent additions of organic fertilizer in the form of slurry) and the other with increased proportion of clover and no slurry application (i.e. nitrogen provided by biological fixation instead of organic fertilizer). The authors find that absence of fertilization in the field with increased clover resulted in significant reduction in N2O emissions. I agree with the authors' justification of the lack of studies on year-round N2O fluxes for grassland systems. The flux measurement methodology used is sound (but see questions below) and the authors have collected a very complete set of supporting measurements to help in the interpretation of fluxes. The study contributes a solid dataset that could be valuable for future modelling efforts. I also liked the use of the GAM models to attribute the N2O flux to various covariates.

My main difficulty with the manuscript is the larger context of practicality of the management practice studied. The authors use the term 'fertilizer nitrogen' and 'fertilization' throughout but as far as I understand their experimental setup refers to substituting livestock-derived organic nitrogen with biologically fixed organic nitrogen. Substituting an external input such as synthetic fertilizer with N from biological fixation makes a lot of sense but presumably the dairy/pig farms have slurry that contains N and other nutrients to be recycled back to the soil. The authors used slurry that had been digested in a biogas plant so what will be the fate of this material if not returned to soil? Hence, the proposed mitigation management does not fit within a larger nutrient balance framework: grasslands are producing animal feed but the manure is not returned to fields and instead more nitrogen is added to fields through biological fixation. Perhaps there is some local context that would justify the proposed mitigation practice. If not, I am having a bit of difficulty in identifying the value of the research findings. Overall, I would like to see the dataset published but I think the argument for the value of the findings has to be much better articulated. Perhaps placing their measurements within an N budget framework would help making the manuscript a unique contribution.

AC: Thank you for this relevant comment. Indeed, we investigated a system which was conventionally managed by livestock-derived organic fertilizer application, but we see this in the wider perspective and argue that the slurry should replace mineral fertilizer at other sites (e.g. nearby crop sites). While we do argue for direct replacement of mineral N with legumes on fields currently fertilized with mineral N, it is clear that introducing legumes is considered much easier in grasslands than in other crops or would introduce unwanted challenges for the farmer (i.e. to harvest mixed crop cultures is more challenging compared to grassland). We strongly agree that the slurry digested in a biogas plant should be used, may it be for fertilization as it contains valuable N while industrial N fixation requires high energy input, or another purpose. Implications on the farm level etc. should be

investigated but this was not the target of our study. We think that the results of our experiment provide useful insights on $N_2O$ fluxes from both, clover and control parcel. It needs to be noted that it was not our intention to artificially change the farmers management to mineral fertilization but instead remain as closely as possible to the conventional management as defined by the farmers practice. The wider context of where to apply this strategy and under which conditions it is worthwhile implementing can be investigated in a potential follow-up project. Similarly, we think that the full N budget of both parcels would be worthwhile an investigation in the upcoming years. During our experiment in 2015/16 our focus was on the processes leading to (increased or decreased) $N_2O$ fluxes and to measure a large dataset of driver variables of $N_2O$ fluxes, while the N budget considerations would have required additional data acquisition (i.e. $NH_3$ emissions, dry/wet deposition, $N_2$ emissions) and was beyond the scope of this project.

We added a sentence in the discussion to make clear that we did not cover these aspects, but think that they are important: (Page 15 L17)

The assessment of the mitigation strategy revealed reductions in $N_2O$ emissions, an increase in BFN and stable yields under mitigation management. Long-term effects of the mitigation strategy on the N budget of the site, as well as implications on the farm level, (e.g. the feasibility to use the slurry to replace mineral fertilizer elsewhere, fodder composition) should be investigated in future studies.

**R:** Title: Use of legumes is a central theme of the manuscript but does not appear in the title. 'Management matters' is catchy but also well-known and a bit vague since it does not specifically identify which management.

**AC:** We strongly agree that stating the specific strategy in the title is useful and change the title to: "Management matters: Testing a mitigation strategy for nitrous oxide emissions using legumes on intensively managed grassland"

**R:** Abstract L. 11: I think the suggested mitigation strategy is for replacing synthetic fertilizer with BNF not animal-derived N with BNF.

**AC:** This seems to be a misunderstanding. Indeed, our aim was to use BNF to reduce the application of animal derived N in form of feces to this grassland and potentially at a wider scale to reduce mineral N application in other fields. Please see also our previous commentary.

**R:** Here and throughout need to identify if referring to organic or synthetic fertilizer

**AC:** We agree with the reviewers comment that the type of fertilizer needs to be introduced in the abstract. The introductory sentence is still valid from a more general viewpoint. The findings on the clover parcel provide a reference to any fertilized field. We would have expected similar or even larger differences when we would have changed the control to field that is characterized by mineral fertilizer amendments. However, this would have implied to artificially change the existing management practices at the control. As a consequence, we added "organically" fertilized in the abstract (L15).

**R:** Check the consistent use of BNF vs. BFN throughout.

**AC:** According to your suggestion we introduced biologically fixed N (BFN) on Page 1 (L9) and used it in a consistent manner throughout the revised manuscript.

**R:** L. 14: Could a broader objective statement be used here (i.e. quantify is a step in trying to answer a more meaningful objective);

**AC:** We agree, that the objective statement should be broad and meaningful and rephrased in order to make our objective clearer: "In order to assess the overall effect of this mitigation strategy on permanent grassland, we performed an in-situ experiment and quantified net $N_2O$ fluxes and biomass yields in two differently managed grass-clover mixtures."

**R:** L. 17: 'To assess the effect of the mitigation strategy' on what?

**AC:** ... on biomass yields and $N_2O$ emissions, we found it repetitive to state this twice in the sentence. See also our previous comment.

**R:** L. 18: No results of the 15N method are presented in abstract leaving the reader wondering why it is mentioned here.

**AC:** We thank the reviewer for pointing towards this lack of information and added this information in the revised version (Page 1 L 19):

"N inputs via BNF (biological nitrogen fixation) were similar in both parcels in 2015, (control: $55 \pm 5$ kg N and clover parcel: $72 \pm 5$ kg N) due to similar clover proportions (control: 15% and clover parcel: 21%), whereas in 2016 N inputs via BFN were substantially higher in the clover parcel compared to the much lower control (control: $14 \pm 2$ kg N with 4% clover in DM and clover $130 \pm 8$ kg N and 44% clover)."

**R:** L. 21: Here and throughout, the authors use 'no management' which is not a very clear term. I think the authors mean during background or baseline emission periods (i.e. outside of emission events associated with management). Either the term is defined early on and used or different wording should be used.

**AC:** Thank you for the comment, it is useful to define "no management". As background or baseline emission might be understood as emissions of a field under no N inputs (neither N fertilizer, nor BFN) we did not want to use these terms. As a consequence, we defined "no management" in in the method section, (Page 10 L9): "no management (here defined as no management during the previous week)"

**R:** L. 24-25: Did the overall N input also decrease in the clover treatment? An overall decrease in N input is different than the argument of 'replacing fertilizer N with BNF' used at the start of the abstract. It is not surprising that N2O emissions are lower if N inputs decreased.

**AC:** You are making an important point in highlighting the usefulness of N balance considerations, however this is difficult to conclude without measuring $NH_3$ and $N_2$ losses. Rough budget considerations (See Table R1-1) with estimation of the overall N input - $NH_3$ and $N_2$ (which is especially uncertain), and other gaseous losses resulted in lower overall N input in the clover parcel in 2015 but similar overall N input in the clover parcel in 2016 compared to the control. This was estimated based on literature values, and we are aware of the large uncertainty in these estimates. Thus, we did not want to include uncertain estimates in our manuscript and suggest to expand the activities at the site to monitor other components of the N cycle in future projects and evaluate changes over a longer period.

Table R1-1. N Budget considerations for both experimental years at parcels. Please note that we do not see a measured N budget within the scope of our study and see this as a rough estimate, which needs to be refined in future projects.

| Flux | Uncertainty | 2015 Control parcel | 2015 Clover parcel | 2016 Control parcel | 2016 Clover parcel |
|---|---|---|---|---|---|
| total BNF (above and below ground) | ±10% | 76 | 90 | 18 | 160 |
| $N_{fert}$ | ±5% | 296 | 0 | 181 | 0 |
| N dep (dry + wet) $NO_x$ $HNO_3^-$ $NO_3^-$ $NH_3$ | ± - 10 +50% | 28 | 28 | 28 | 28 |
| **Total input** | | **400** | **117** | **226** | **188** |
| N in Biomass export - 10% | | 271 | 238 | 214 | 236 |
| $NH_3$ (30% N fert) Ammann 2009 | 20-35% | 89 | 0 | 54 | 0 |
| $NO_3^-$ Leaching 7% of $N_{fert}$ + BNF (Ledgard et al. 2009) | (2-40%) of $N_{fert}$ + BNF (was below 3.5 kg in Ammann 2009) | 26 | 6 | 14 | 11 |
| $N_2O$ | ± see text | 4.1 | 1.9 | 6.3 | 3.8 |
| $N_2$ (5.7* $N_2O$) Butterbach-Bahl et al. 2013 | 2-11* $N_2O$ | 23 | 11 | 36 | 22 |
| **Total export** | | **413** | **256** | **324** | **272** |
| **Budget*** | | **13** | **139** | **98** | **85** |
| **Budget (uncertainty range)** | | **13 (-94 - 233)** | **139 (97-228)** | **98 (14-258)** | **85 (25-210)** |

*-Input + Export, positive sign indicates soil to atmosphere

**R:** Page 2 L. 13-14: Instruments for EC measurements have been available for a least a decade.

**AC:** The sentence is referring to high frequency in-situ N2O measurements that would be continuously possible over the whole year, not basic EC measurements, we specified the sentence to make this clearer "instruments capable of high-frequency continuous N2O concentration measurements and steadily deployable in the field have only become available in recent years".

**R:** L. 17: 'abiotic factors are generally known'

**AC:** The text was corrected accordingly.

**R:** L. 18-19: N2O emissions strongly depend on management practices in most managed systems so I suggest to remove 'particularly grasslands'.

**AC:** Removed according to your suggestion.

**R:** L. 31: Here the authors refer to 'external fertilizer amendments' but slurry would not be considered an external amendment.

**AC:** We agree that there is no reason for using external without definition and deleted it in the revised text.

**R:** Page 3: L. 3-5: This statement seems a bit misleading given that this study is not dealing with synthetic fertilizer.

**AC:** We deleted this statement in order to avoid confusion.

**R:** L. 33: What is the reason for substituting the N fertilizer from organic origin that is already available on the farm?

**AC:** We would like to refer to our initial explanation made at the beginning of this response. In brief, our aim was to test a potential mitigation strategy which could lead to less mineral fertilizer being used in other places.

**R:** Page 4: L. 3: Could the second objective be made more specific? 'to identify the drivers of N2O emissions' is a bit broad.

**AC:** We rewrote this sentence "… to identify meteorological and soil chemical drivers of N2O emissions…"

**R:** L. 21: Is this 'dairy' liquid slurry? Why 'predominantly'? Table 1 refers to 'organic fertilizer': was there another source beside slurry?

**AC:** There was no other source besides liquid slurry (originally cattle slurry/manure digested in the biogas plant and then returned to the farm via pipeline) during the experiment, Table 1 specifies "slurry" in the column right to the entry of "organic fertilizer". To clarify we changed "slurry" to "liquid slurry" in Table 1. It needs to be noted, that during the past ten years there were occasional manure and synthetic fertilizer applications.

**R:** Was the aim to replace all the ~260 kg N/ha with BNF? Table 2 shows a maximum of 130 kg N/ha from BNF. What is the recommended N input to sustain the N removed?

**AC:** Recommendations for Swiss intensively managed grasslands suggest 30 kg available N ($N_{available}$) per cut (Flisch et al., 2009) (Table 27), whereby $N_{available}$ is calculated as 60% (50-70%) from the total slurry N ($N_{slurry}$) (Table 40 in Flisch et al. 2009). Thus, 50 kg $N_{slurry}$ recommended (following Flisch et al. 2009 $N_{slurry} = N_{available}$ /0.6, range of $N_{slurry}$: 43-60 kg ) per cut result in 250 (214 – 300) kg $N_{slurry}$ for five cuts (2015) and 200 (171-240) kg $N_{slurry}$ for 4 cuts (2016) for intensively managed grassland. For BFN we can estimate that the amount of fixed N total exceeds the presented above-ground BFN in shoot biomass, due to roots biomass N (additional 70% of shoot biomass (Jørgensen and Ledgard, 1997)) and N transferred to grasses (Nyfeler et al., 2011), which would be e.g. 30 kg N for 2016. The N exports in biomass were on average 180 kg ha over the last 10 years, and ~260 kg N as in the experiment (Table 2) would need to be replaced in the long term. Our aim was therefore to get as much N via BFN as possible, but it was clear that the sward would not be able to replace ~260 kg N with BFN within this short time after over-sowing.

**R:** L. 24-25: Where there any differences in texture, C content, etc. in the two fields monitored?

**AC:** Within the footprint, soil sampling in 0-10 cm depth suggested, that naturally there is some variability (see Table R1. However, this can be seen as typical within field variation and we do not expect that the small soil differences predominantly affected $N_2O$ exchange.

Table R1. Aggregated data separated into two parcels from soil mapping performed in (Roth, 2006)

| Parcel | Corg (kg ha$^{-1}$) | | Corg (kg ha$^{-1}$) | | Bulk density (kg m$^{-3}$) | | Number of samples |
|---|---|---|---|---|---|---|---|
| | Mean (±sd) | Range | Mean (±sd) | Range | Mean (±sd) | Range | |
| **Control** | 33.7 (±4.5) | 24.9–39.7 | 34.1(±2.7) | 29.3–37.3 | 1.03 (±0.07) | 0.92 1.18 | 13 |
| **Clover** | 27.4 (±2.5) | 24.4–33.7 | 30.7 (±2.3) | 27.3–35.2 | 1.13 (±0.03) | 1.08–1.19 | 16 |

**R:** Page 5: L. 3: Was the control plot also grazed? If not, what are the implications of the different grazing regimes?

**AC:** "The control parcel was mown once instead of being grazed during this time of the year (beginning of July)." Furthermore, the control was not grazed in June-July 2015 and cut instead, as this grazing was regarded part of the clover-adapted management, and thus it should be part of the management that we wanted to assess. In contrast, the winter grazing had no clover-related motivation and was just part of the farmers usual practice, and therefore was performed on both parcels (as part of business-as-usual). Overall, we do not expect large effects of grazing due to its low intensity (see also result section 3.6).

**R:** L. 16: better to say 'digested cattle and pig slurry obtained from a local biogas plant'. If this digestate is not applied to land what is its fate?

**AC:** We rephrased this phrase according to your suggestion and further refer to our explanation of the aim of this study in our first comment.

**R:** Page 6: L. 7-8: Any replicates or just one sensor per depth?

**AC:** This was only one sensor per depth due to limited resources.

**R:** L. 9: How many locations of the 2 ha plots were monitored to determine impacts on microclimate?

**AC:** We installed one representative sensor field plot per parcel, in the main footprint area.

**R:** Page 8: L. 3: How well does EddyPro work for processing N2O data? How were the lag times determined given that low N2O signal often means the cross correlation method for determining lag times will be off.

**AC:** Processing $N_2O$ is fluxes in EddyPro works well if the search window for the lag time between eddy covariance wind data and the QCL $N_2O$ mixing ratios is constrained as much as possible. For the calculation of the time lag between the wind component $w$ (after coordinate rotation) and the mixing ratio of $N_2O$, we followed the covariance maximization method in combination with a pre-determined nominal lag time. This means, we identified the peak in the cross-correlation function between $w$ and $N_2O$ in a defined time window of physically possible time lags. As pointed out by Reviewer#1, this cross-correlation function is often noisy due to low signalto-noise ratios as a consequence of low $N_2O$ fluxes. In our study, analyzing the frequency distribution of detected lag times over the course of a "low-flux" year yielded clear results that assisted our processing in constraining the lag time search to a relatively short but adequate time window. In order to accurately identify the time lag and avoid systematic bias, we followed a multi-step approach before the calculation of final $N_2O$ fluxes for each year: Step (1): We first calculated the time lag for the respective year using the covariance maximisation method for a relatively large search window of $\pm$ 10 s, based on despiked and detrended raw data. Next, we analysed the frequency distribution of all found lag times and identified a clear distribution spike between 0.95 and 1.40 s, i.e. most cross-correlation peaks where found in this range (Fig. R1-1 shows results for the year 2013).

[Figure]

Figure R1-1. Histogram of found lag times between the turbulent wind component w and N2O mixing ratio in a time window of $\pm$ 10 s in 2013. Peak distribution was found at lag time 1.20 s. Shown are only found lag times below 5 s for clarity.

Step (2): Based on the frequency distribution from (1), we defined a narrow lag search window of 0.6 – 1.8 s, i.e. the range in which we found most cross-correlation peaks in (1). In addition, we defined the lag time of peak distribution from (1) as the nominal lag time (1.20 s), that is, whenever the returned time lag was one of the limits of the window (i.e. 0.6 or 1.8 s), the time lag was set to the nominal time lag of 1.20 s. This specific nominal lag time was chosen because it was identified as the most representative lag time over the course of a year. We also analyzed the time series of found lag times to investigate potential seasonal differences. We found that lag times over the course of a year fluctuated slightly but were well within the range of the pre-defined search window of 0.6 – 1.8 s. We are therefore confident that the accurate lag time is found in the defined range.

In addition, we are confident to have adequately constrained the time window due to the used measurement setup at the site. At Chamau, we measured eddy covariance data using a fully digital and real-time logging system that was described previously (Eugster and Plüss, 2010). One major advantage of this system is that wind and scalar data at a specific moment in time are merged immediately in the same data file. As a consequence, the system is not prone to time drifts between the two instruments (e.g. sonic anemometer and QCL) and the lag time can therefore consistently be found in a relatively narrow time range.

**R:** The same applies to the corrections - if cospectra are being used for any of the corrections they may not be correct for periods with low N2O or CH4 signals.

**AC:** We applied the spectral correction following (Fratini et al., 2012), which builds upon the work by (Ibrom et al., 2007). For each instrument and gas, this method applies spectra correction factors (analytical) in combination with cut-off frequencies (*in-situ*). This method allows to define a threshold between small (in this study: below 2

nmol m$^{-2}$ s$^{-1}$) and large (above 2 nmol m$^{-2}$ s$^{-1}$) N$_2$O fluxes. For CH$_4$ during the same study period the threshold was set to 10 nmol m$^{-2}$ s$^{-1}$. Small fluxes are then corrected following an adjusted model by (Ibrom et al., 2007) (see also Section 2.3, equation 4, and Appendix A in (Fratini et al., 2012), large fluxes are corrected following the approach by (Hollinger et al., 1999) (see Section 2.2, equation 2, and Section 2.3, equation 3, in Fratini *et al.*, 2012). In addition, EddyPro allows the application of the same spectral assessment for different years to improve comparability of flux results between years. Therefore, we first calculated spectral assessments for the year 2012, the year of grassland restoration and the widest range of observed fluxes (from high fluxes during management after restoration to low fluxes later in the year). We then applied these flux assessments to all other years to achieve corrected fluxes. Therefore, to our current best knowledge, we think that the applied corrections are adequate. In addition, we want to acknowledge ongoing discussions within the Integrated Carbon Observation System (ICOS) regarding best-practice calculations of non-CO$_2$ eddy covariance fluxes, upon which our selected processing steps are based on (Nemitz et al., *in review*).

**R:** L. 16: Are there any biases associated with S or N flow being associated with different kinds of weather (rain occurrence, higher T, etc.)? Please expand on this potential impact.

**AC:** Higher temperatures are associated with higher coverage for the clover parcel fluxes which is more frequently covered during sunny days during daytime. This result in a better representation of fluxes from the clover parcel and less representation of the fluxes from the control. However, both parcels were covered well resulting in a quite low. Precipitation was not especially associated with any wind direction. For the comparison, we minimize this bias by using only days covered on both parcels. We added a sentence to make this clearer (see manuscript Page 9 L5).

"Relative flux differences between parcels were defined as the difference of daily averages between clover and control parcels with respect to the average flux from the control, calculated based on all days for which data from both parcels were available. This was done to minimize potential biases associated with periods of unequal coverage of both parcels."

**R:** L. 24: Were directions always steady over this time period or did you use high frequency data to select the 10 min periods with steady wind direction?

**AC:** On page 8 L15 we wrote "Each 10-min flux average was attributed to a parcel only if a minimum of 80% of the flux footprint was in the direction of the respective parcel (i.e. footprint weights from the direction of the respective parcel divided by the total of all flux footprint weights > 80%)." In other words, from the overall 10-min footprint, our selection criteria were that a minimum of 80% of the footprint weights had to originate from this respective parcel and thus were representing steady conditions. Wind direction changes during unsteady conditions within <10 minutes largely influence this criterion, thus no further restriction was applied.

**R:** Page 10: L. 34: Could more information be given on 'management'? Which 'Management' aspects? These are not quantitative variables or continuous so how was this handled?

**AC:** We handled them as categorical variables and refer here to page 10 L9 "For introducing management influence in the regression analysis, dates were labelled according to three *a priori* selected management categories only: post-fertilization (F), post-harvest (H) and no management (0) in combination with the treatment

clover (Clo) or control (Ctr). Thus, five management categories existed (Ctr-F, Ctr-H, Ctr-0, Clo-H, Clo-0). The control parcel without recent management activity (Ctr-0) served as the reference level in comparison to all other management categories."

R: Page 12: L. 22: Odd wording: 'later clover parcel'; please edit. Was there a trend for lower yields for the 'clover' parcel before the experiment started? Is this related to differences in soil? L. 24: Was difference consistently <0 for all years? What are the implications for previous C input to soil and N2O fluxes during measurement period?

AC: We changed "later clover parcel" to "parcel which was transformed into the experimental parcel during the years 2015 and 2016". For the 2007-2013 yields were consistently lower in the parcel which was transformed into the experimental parcel. Differences might occur for multiple reasons: (1) differences in management, i.e. more grazing on the parcel which was transformed into the experimental parcel compared to the other, (2) slight differences in soil properties (i.e. N stocks) due to differences in N inputs (3) a bias in the farmer's field book estimate.

R: Page 14: L. 15-16: This interpretation assumes that there was no interaction between factors, i.e. all effects during non-events, called 'times without management' by authors, are due to sward composition while all effects during events was due to slurry application. Please discuss potential interaction effects.

AC: This might be a misinterpretation. Effects of other environmental drivers were included. The gam model as used here takes all predictors into account (Table 4). The $N_2O$ flux in the model is predicted using the effects of management, temperature, soil water content etc. Thus, by including these "confounding variables" they are no "confounding variables", but taken into account as predictors instead. Thus, variability in soil temperature and soil water content is represented in the model and is therefore not confounding this interpretation. The management effects in the model can be interpreted as offsets (resulting in different multipliers for all management categories). What we were not able to include were carry-over effects and land use history effects, including past fertilization on the time of "no management".

R: Page 15: L. 23-24: 'In sum, our results indicate that N2O can be effectively reduced through the replacement of fertilizer N with N from BNF.' But was all slurry N replaced by BNF? Did the system have an N surplus before that could have been addressed by just matching the Slurry N input to crop needs? IF that is the case then the most achievable mitigation practice would be to adjust the slurry rates to the plant demand instead of using BNF.

AC: The N surplus in the soil during and shortly after fertilization is unavoidable and relevant in all fields. It needs to be said, that the farmers in our experiment did not excessively amend slurry, and the site was managed within the framework of regulations. We fully agree with the reviewer, that is crucial to increase NUE as much as possible by optimizing the timing of slurry application, however the steady N input achieved via BNF differs from "bulk" N input via slurry application. Timing and amount of fertilizer application are indeed mitigation practices (but not investigated in this study); As the farmer managed the field according to recommended practices which are based on N balance considerations, there would arise concerns about productivity when reducing fertilizer amounts at no further N input changes.

**R:** Page 17: L. 31: 'Thus, we observed lower N2O emissions at higher levels of photosynthesis'. Higher levels of photosynthesis also correspond to periods of higher water loss as transpiration and lower water content as seen in your dataset. How can you separate these two confounding effects?

**AC:** An effect of soil water content, may it be due to high transpiration or lack of precipitation or both, would be accounted for in the model as we put soil water content as one of the predictor variables so such and effect would be attributed to the soil water content. Adding NEE as a predictor, however, improves the model (i.e. reduces AIC) and adds explanatory power. Of course, photosynthesis is also correlated with soil water content and temperature, however as we included both in our model they cannot be confounding (i.e. low $N_2O$ emissions at low water contents are already reflected by the effects of soil water content, if the inclusion of NEE in the model lowers AIC in addition to that, it means that there is an additional effect which was not explained without NEE. Evapotranspiration is not a causal driver of $N_2O$ production.

**R:** Page 19: L. 24: This brings up an important point. 'Permanent grasslands' are temporarily restored as was done with the study area in 2012. Does this restoration involve ploughing or a similar tillage at this site? If so, then authors should discuss the impact of incorporating a larger proportion of legumes into soil on the following N2O emissions.

**AC:** Yes, the site was ploughed it 2012 and this was done at both parcels. However, we could not find carry-over effects nor would we have found differences in $N_2O$ exchange during our experiment as the parcels were not ploughed in 2015/2016. This could become a very nice experiment in a few years when each parcel will have to be restored.

We added in the discussion (Page 15 L 18): "This study covered two years and did not include potential effects of incorporation of clover into the soil during ploughing (which is every 8-10 years)."

**R:** L. 25-26: But what would farmers do with the slurry?

**AC:** Our suggestion is to use the slurry from the biogas plant in fields where currently mineral fertilizers are used. See also our initial statement on this issue.

Technical corrections

**R:** Page 4: L. 5-6: Word missing here: 'while fertilization to play the dominant role in driving N2O emissions in the control parcel'

**AC:** We separated this into two sentences in the revised manuscript: "We hypothesized considerably lower $N_2O$ emissions in the clover parcel, lower soil nutrient availability in the clover parcel and thus no effect of legume proportions on $N_2O$ emissions, and hypothesized fertilization to play the dominant role in driving $N_2O$ emissions in the control parcel" (Page 4: L. 5-6).

**R:** Page 5: L. 7: Mowing is only one of the activities used in harvesting (presumably biomass is also removed). I think it is less confusing if 'harvest' is used throughout.

**AC:** We changed this line to "Management activities comprised the regular harvest activities (mowing, swathering and subsequent biomass removal)". However, we used mowing elsewhere, as this defines clearly the start of the harvest process and gives thus an indication that we refer to the start of the event.

**R:** L. 8-9: Some of this information is already given on L 31 in previous page.

**AC:** We shortened that sentence to avoid redundancy.

**R:** L. 11: Please clarify the study years (2015- 2016, correct). Are the data for 2014 presented?

**AC:** We did not present the biomass data for 2014 and thus changed to 2015-2016.

**R:** L. 17: Check table as 'herbicide' not mentioned.

**AC:** The latest herbicide application was in 2013, while no herbicide was applied during the period displayed in the table (2015-2016).

**R:** L. 20: I think you mean with instruments mounted on a mast...

**AC:** We think that in tower is correct terminology here, also for the smaller towers as applied on cropland and grassland sites, see for example Aubinet et al. (2012).

**R:** L. 21: Instead of 'lying' use 'placed'.

**AC:** We changed to "being located"

**R:** L. 26: Instead of 'The air inlet for N2O, CH4 and H2O' use 'The air inlet for the absorption spectrometer'.

**AC:** We changed this according to the reviewer's suggestion.

**R:** Page 6: L. 1: Should refer to Fig. 1 in this section.

**AC:** We agree and added the reference Fig. 1 (Page 6 L6).

**R:** L. 13-15: Is this in addition to the 'soil sensors for microclimate measurements' mentioned in L. 9-10?

**AC:** No, with the sentence in L9-10 the soil plots are introduced, while "additional" refers to the fact that we have sensors installed directly at the tower and not in a specific parcel. In order to clarify this, we added a reference to the soil plots in Fig.1, i.e. in L10 we changed to "In addition to the sensors close to the tower, each parcel was equipped with a similar set of soil sensors in 2015 (see soil plots, Fig.1) to compare potential differences in soil microclimatic conditions and subsequent effects on GHG fluxes". Then the specification for these follows. Sensor locations are also specified in Table S1.

**R:** Are matric potential and soil heat flux data presented?

**AC:** No, we excluded both from the analysis as they were highly correlated with variables such as soil water content, soil $O_2$ concentrations, soil temperature and were suspecting problems with multicollinearity if we included them.

**R:** L. 26: Do you mean the 'average footprint'?

**AC:** Indeed, therefore, we adjusted to average footprint (L. 26).

**R:** Page 7: L. 20: Perhaps give info on clover proportion before (start of methods). The assumption is that proportion of establishment is the same as sowing composition. Did you check the final stand composition?

**AC:** Please see page 7 L4: "Vegetation was separated into legumes and non-legumes (grasses and forbs) to assess the actual legume proportion in the dry biomass" There is no assumption involved, we directly measured clover biomass weight of the harvested biomass at every harvest (separating harvested biomass in non-legume and legume species), and the resulting percentage is based on the annual biomass. The percentages of clover in dry biomass per harvest are presented in Figure 5c.

**R:** L. 23: Use 'Gauss'.

**AC:** We changed gauss to Gauss.

**R:** L. 29-30: CO2 molar density refers to the Licor measurements? Since CO2 was also measured with the QCLAS it may be helpful to inform the reader in EC section as to which data will be resented for CO2.

**AC:** $CO_2$ was not measured with the QCLAS; In the method section 2.3 we described "The flux measurement setup consisted of a 3-D sonic anemometer (Solent R3, Gill Instruments, Lymington, UK), an open-path infrared gas analyser for $CO_2$ and $H_2O$ concentrations (LI-7500, LiCor Biosciences, Lincoln, NE, USA) and a quantum cascade laser absorption spectrometer (QCLAS) capable to measure $N_2O$, $CH_4$ and $H_2O$ concentrations (mini-QCLAS, Aerodyne Research Inc., Billerica, MA, USA)".

**R:** Page 8: L. 6: 'below' instead of 'above' for N2O and CH4 values.

**AC:** Thanks, we corrected this.

**R:** L.14: fig 1 refers to Kljun et al 2004.

**AC:** The figure was indeed produced with the Kljun et al. (2004) tool for the submitted manuscript since the new 2015 footprint code made available by Kljun et al. (2015) was buggy and provided unrealistic results in our specific case (we are in discussion with N. Kljun to solve these issues). Thus, we used the cross-wind integrated footprints of her 2004 version of the footprint model for the plot, but we plan to update the Figure in the revised manuscript with the new (2015) version calculations after having solved the remaining issues.

**R:** L. 15: "Parcel" does not seem to be the correct term here. Do you mean 'field' or 'plot'. Why is footprint so different for South vs North?

**AC:** We checked the wording ("parcel") and confirm that the terminology corresponds to what we mean. The topography is not symmetric, the Alps are in the South and the Swiss Plateau is in the North. The pressure and temperature differences between the Alps and the Swiss Plateau drive the wind, and hence the wind rose and footprints are not perfectly mirrored between northerly and southerly winds.

**R:** Page 9: L. 13: What was the 'similar management practice' in 2014? Please explain earlier in text.

**AC:** We added text earlier in the revised version of the manuscript (page 5 L5):

"During our reference years 2013 and 2014, management was identical in both parcels in 2013, while in 2014 instead of mowing, cattle were grazing in the control parcel whereas the clover parcel was mown, resulting in similar reference fluxes from both parcels."

**R:** L.15: I suggest 'Three management types and one natural event type' be changed to 'Three management-derived events and one natural event'

**AC:** Changed similarly to your suggestion to "three management event types"

**R:** Page 11: L. 3: Check citation. What does 'these' refer to?

**AC:** Wilks, 2011, page 147 is correct, and this is Copernicus style. "These" refers to the effective sample sizes, we clarified this in the text. "By calculating the effective sample sizes according to (Wilks, 2011:147) and using the effective sample sizes …"

**R:** L. 23: Sentence not complete.

**AC:** We corrected this. "Volumetric soil water content (at 0.1 m depth) were similar in the control ($33 \pm 4\%$) and the clover parcel ($31 \pm 5\%$)"

**R:** Page 12: L. 20: Table has SE as 0.5 not 0.6.

**AC:** The correct version is in the Table thus we changed to 0.5.

**R:** Page 13: L. 8-9: I am not sure what is being compared here (within years or across years): biomass at the clover parcel was lower ($5.1 \pm 0.3$ t ha-1 yr-1) in 2015 and similar ($4.8 \pm 0.2$ t ha-1 yr-1) in 2016.

**AC:** Here we want to compare C in biomass within years. We made the sentence clearer by rearranging and, putting the years together. "C in annual yields at the control parcel was higher ($5.8 \pm 0.2$ t ha$^{-1}$) compared to the clover parcel ($4.7 \pm 0.3$ t ha$^{-1}$) in 2015, while C in biomass was similar for the control parcel ($5.1 \pm 0.3$ t ha$^{-1}$) and the clover parcel ($4.8 \pm 0.2$ t ha$^{-1}$ yr$^{-1}$) in 2016 (Table 2)."

**R:** L. 10-11: But values are indicated as 'ab' so not different?

**AC:** Your interpretation of the letters is correct, N exported is not significantly lower in the control. We adapted this: " N exported was similar across parcels  in the second year (control: $238 \pm 13$ kg ha-1 yr-1 ; clover: $262 \pm 8$ kg ha-1 yr-1 )  (Table 2)."

**R:** Page 14: L. 8: This statement is confusing: 'periods without management'.

**AC:** Hier war das Ziel das dazuzuschreiben, damit klar ist, dass die Referenz Ctr-0 ist. We revised: "Nitrous oxide emissions significantly increased after fertilizer application (Ctr-F compared to Ctr-0, $p < 0.05$) when compared to $N_2O$ fluxes during periods of no management on the same (control) parcel (Fig. 8a, Table 4)."

**R:** L. 10-11: 'a 2.5-fold increase in N2O emissions during the seven days following slurry amendment compared to no management (Table 4).'. It is hard to find this information in table; please help reader by giving values...

**AC:** We addressed this by an explicit explanation in brackets. "The effect size showed 2.5-fold $N_2O$ emissions during the seven days following slurry amendment compared to no management (resulting from applying the back-transformation of to the fertilization effect: $10^{0.4} = 2.5$; Table 4)". We stick with the straightforward model outputs in the table.

**R:** L. 11: 'It is important to state that the management effect exists in addition to the effect explained' is not clear, please edit.

**AC:** With this statement, we wanted to clarify what the reviewer correctly questioned before: At the days after harvest, not all explained variability is necessarily explained by harvest, and rather a combination of the effects of soil temperature and all others driver variables included in the model. We changed the sentence to: The effects of management influence $N_2O$ fluxes jointly with other measured driver variables (L. 11).

**R:** Page 16: L. 21: I do not consider soil conditions as not 'external' factors. . .

**AC:** We agree that the word "external" is misplaced here and replaced this with 'other'.

**R:** Page 17: L. 11: 'Clo-0' is not used in table, please use consistent terminology.

**AC:** In the table caption we described "The control parcel without recent management (Ctr-0) was used as the reference level for the categorical variable management, thus the constant represents predictions for Ctr-0 and the effect sizes of all other management categories depict differences compared to Ctr-0". Therefore, we decided to keep this as it is as to our understanding `intercept` (constant) means something different than the effects of the other treatments.

**R:** Page 19: L. 24: This brings up an important point. 'Permanent grasslands' are temporarily restored as was done with the study area in 2012. Does this restoration involve ploughing or a similar tillage at this site? If so, then authors should discuss the impact of incorporating a larger proportion of legumes into soil on the following N2O emissions.

**AC:** Yes, permanent grasslands are restored, here typically every eight to ten years. We clarified the importance of the ploughing

**R:** L. 25-26: But what would farmers do with the slurry?

**AC:** Please see our initial comment on this issue.

**R:** Figure 1: Add explanation of 'blue dots' into figure caption.

**AC:** We added "Blue dots represent soil sampling locations."

**R:** Why are there many more contour lines in the footprint for the N plot vs. S plot?

**AC:** The footprint is shown as a probability density function in 2-d, thus there are two aspects which lead to this: (1) northerly directions are somewhat more frequent than southerly directions; and (2) winds from south vary slightly more in exact wind direction than winds from the north, and hence the probability function is wider (and longer) than when wind is from the north, thus lowering the probability for each unit surface area to be within the

footprint. As an example: each meter squared along one contour line has the same weight in the annual flux budget. In other words: the true area covered with measurements when wind is from the north is constrained to a smaller surface than if the wind comes from the south, but since we kept homogeneous conditions in each of the two plots, both directions are representative for the respective plot that measurements belong to.

**R:** Figure 2: Please indicate if symbols are for daily values

**AC:** We corrected the caption as follows: "**Figure 2.** Meteorological conditions during 2015 and 2016. (a) Average daily air temperature (2 m), (b) average daily photosynthetically active radiation (2 m). The grey bars indicate the sub-daily variability (quartiles based on 10 min values). (c) Daily precipitation sums during 2015 and 2016 (1 m)."

**R:** Figure 6b: not referred to in text.

**AC:** We found that the Figure 6b is referred to on page 13 L23 where we wrote: "… and 39% (36–42%) lower than at the control parcel in 2015 and 2016, respectively (Fig. 6b)". Still, the Figure 5b was not referred to. We added the sentence: "The living aboveground biomass remaining on the parcel after mowing was $1.0 \pm 0.3$ t DM ha-1 on the control parcel and $0.8 \pm 0.4$ t DM ha-1 on the clover parcel (Fig. 5b)."

**R:** Figure 8: letter a, b, . . ..missing from graphs. X-axis scale for CO2 flux seems to be wider than dataset.

**AC:** We added the missing letters, which occurred due to the mix-up between earlier versions of this Figure.

**R:** Table 2: Was means test applied to treatments within a year or across two years? The latter (I think) but some of the discussion in text seems to consider ab different than b or a. Letters are missing for last row and 3rd last row year 2016 data.

**AC:** The interpretation of the reviewer is correct, tests were applied across two years and your interpretation of the letters were correct. The missing letters are intention, you can either give a letter to each group (including values that don not go together in a group with others or use only a letter in case of "not significantly different" comparisons in the latter choice. We used the latter option, which means that all values without letter are significantly different from all others. Realizing that your approach might be the more common one, we will change the revised manuscript version accordingly.

**R:** Table 4: Use 'harvest' instead of 'mowing'.

**AC:** We changed this according to your suggestion.

**Anonymous Referee #2**

**R:** This is an important paper, comparing N2O emissions from adjacent fields with different proportions of clover content. The paper is well written, and suitable for publication in BG. My main comments are as follows: 1) In the 'mitigation' treatment more clover was added, but as it took time to establish, the differences between clover % was rather small in year 2015 (15% & 21%), whereas in year 2016 the differences were large (4% and 44%). Similar differences were observed for the BNF rates (Table 2). These differences and their implications on the yield and N2O fluxes is not adequately addressed in this paper. 2) N2O was measured using eddy covariance, from 2 adjacent fields. The overall data coverage of both fields was similar (Table 3), but the authors need to demonstrate that the temporal coverage of measurements was similar for both fields. One would not want situations where the airflow is always from field 1 at dawn, for example.

**AC:** With respect to (1), we now address this aspect stating the clover proportions directly when we presented the results for $N_2O$ fluxes (Abstract and Results) and addressed the implications on our findings in the revised manuscript (see Section 4.2). With respect to point (2), we had checked the data coverage also in dependence of daytime, which covered reasonably well all times of day in both parcels. We will add this information to the Supplementary material (Figure R2-1). In fact, in a broad valley with a typical wind system with up-valley winds during peak daytime and down-valley winds during the second part of the night we get the best share of coverage during dusk and dawn (almost 1:1), whereas the extremes are in the range 1:2 to 2:1 (see Figure R2-1).

[Figure]

Figure R2-1: Proportion of flux records during the observation period origination from the clover parcel depending on the hour of the day.

**R:** Further suggestions of edits can be found below Abstract: needs to contain the grass clover proportions for the 2 fields

**AC:** We find this a useful suggestion and added the clover proportions and biologically fixed N in the abstract:

"N inputs via BFN were similar in both parcels in 2015, (control:  55 ($\pm$ 5) kg N and clover parcel: 72 ($\pm$ 5) kg N) due to similar clover proportions (control: 15% and clover parcel: 21%), whereas in 2016 the N inputs via BNF were higher in the clover parcel compared to the control (control: 14 ($\pm$ 2) kg N with 4% clover in DM and clover 130 ($\pm$ 8) kg N and 44% clover)."

**R:** Introduction: 'Apart from the environmental benefits of a reduced N surplus when mineral fertilizer is replaced by BNF, total GHG emissions from fertilizer production of 1.6– 6.4 kg CO2-eq per kg fertilizer N, could technically be avoided (Andrews et al., 2007; Brentrup 5 and Pallière, 2008).' R: In which country of climate zone can such GHG reduction rates be achieved?

**AC:** This statement was referring to mineral fertilizer production. We deleted it in the revised manuscript, as this further aspect seems to be misleading and is not directly important for the study.

**R:** Methods: ''The site has been well investigated in terms of CO2 exchange (Burri et al., 2014; Zeeman et al., 2010), as well as for N2O and CH4 exchange under management that is typical for Swiss grasslands located on the Swiss Plateau (Imer et al., 2013; Merbold et al., 2014; Wolf et al., 2015).' **R:** Add that CO2 exchange is measured by EC and N2O/CH4 by, presumably, static chambers.

**AC:** According to your suggestion, we addressed the methods specifically for each reference:  The site has been well investigated in terms of $CO_2$ exchange (Burri et al., 2014 using static chambers (SC); Zeeman et al., 2010 using EC), as well as for $N_2O$ and $CH_4$ exchange under management that is typical for Swiss grasslands located on the Swiss Plateau (Imer et al., 2013 using SC for $N_2O$ and $CH_4$ and EC for $CO_2$; Merbold et al., 2014 using EC for all three gases; Wolf et al., 2015 using EC and SC for $N_2O$).

**R:** Given that you have reduced the EC averaging time to 10 min from the usual 30 min, I assume that you must have had a relatively equal spread between coverage of both the two plots. You need to demonstrate this, for example by including a graph of N2O versus time with different colour dots for the two treatments.

**AC:** We had analyzed in more detail the coverage of both parcels. This included looking at 'time of day' versus 'number of flux measurements' per parcel binned for time classes as well as of 'time of day' versus N2O flux per parcel, especially after management events. We added a Figure in the Supplement (Figure R2 and R3).

**R:** Why did you fertilize with 296 kg N/ha/2015 and 181 in 2016?

**AC:** Our aim was to not artificially change the behavior of the farmer and let him choose, according to previous practices, which are a joint outcome of N budget considerations but also practical issues such as weather conditions and how much slurry is available at the time when suitable for field application.

Section 2.6: 'and a subsample of 5 mg was weighed into tin capsules for further analyses (n = 5 for each parcel per date).' **R:** You need to add: '. . .for further analysis of total C and N and . . .. . ..'

**AC:** We adjusted this according to the reviewers suggestion (Page 7 L8) "of total C and N, $\delta^{13}$C and $\delta^{15}$N "

**R:** Figure 1: include the prevailing wind direction, or say what it is in the legend

**AC:** We will include in the caption: "The prevailing wind direction was from the north."

**R:** Legend to Figure 2 needs to be tidied up.

**AC:** The correct caption, as in the list of figures was inserted.

**R:** Figure 5: Why do you join the dots for graphs b-d, but not for graphs e-f? Looks like there is some inconsistency here.

**AC:** We inserted the line for prediction only for the variables which improved the model fit. NO3- and DOC did not improve the fit and thus were not included in the reduced model. Consequently, we did not draw the prediction lines (please see also the caption) and only showed the data points.

**R:** Section 3.5: 'During the reference year 2013' Add the reference to these 2013 data.

**AC:** The reference data are part of this paper, we realize that we should have introduced the concept of reference years more clearly in the method section.

Therefore we added (Page 4 L27): "We use the two years 2013 and 2014 as reference years (no treatment). In order to test the $N_2O$ mitigation option, the treatment parcel was over-sown in March 2015 and April 2016 with clover."

**R:** Figure 8 'the factor management' delete 'the factor', or place it before '(a)'

**AC:** It is now deleted.

**R:** Discussion 1st Parag. 'major changes compared to the "business as usual" practice; (1) omitted fertilization and (2) over-sowing clover, leading to an increased clover proportion in the experimental sward' R: Add the % of clover to remind the reader 'to an increased clover proportion of x %'

**AC:** We added (i.e. 21% versus 15 % in 2015, 44% versus 4% in 2016) after this sentence.

**R:** Last sentence and elsewhere: change 'in sum' to 'in summary'.

**AC:** Done in both occurrences (L16 and L26)

Section 4.1: 'than our site, showed typically lower N2O emissions (0.38–2.28 kg N2O- N ha-1 yr-1), which can be explained by lower fertilizer inputs compared to our site (Hörtnagl et al. 2018). 'In sum, our year-round measurements of N2O emissions are higher than multi-site averages due to its fertilizer regime and site conditions, but within plausible ranges compared to other sites. R: Discuss the differences in fertiliser rate and the differences in site composition between the Hortnagl study and yours in greater details, so that the reader also understands why your N2O fluxes are larger. Provide more information on the differences between your site and the Hortnagl sites. And, to improve the English change 'In sum, our year-round measurements of N2O' to 'In summary, our one-year measurements of N2O'

**AC:** Page 15 L26: We will elaborate on the discussion of site conditions and fertilizer rates in more detail and refine the paragraph in the revised version. "In summary" was corrected.

**R:** Section 4.2: 'N2O emissions in the clover parcel during our two-year observation period summed up to 1.9 and 3.8 kg N2O-N ha-1 yr-1 in 2015 and 2016, respectively. These values were clearly lower than the values observed from the control parcel.' R: You need to discuss these observations and others in this section with the fact that the differences in clover proportions between the two fields in 2015 were rather small compared to 2016 (Table 2).

**AC:** We addressed this by adding the clover proportions and the implications in the revised manuscript. "$N_2O$ emissions in the clover parcel during our two-year observation period summed up to 1.9 and 3.8 kg $N_2O$-N ha$^{-1}$ yr$^{-1}$ in 2015 and 2016, respectively. These $N_2O$ emissions were clearly lower than the values observed in the control parcel during both years. In 2015, the difference can be attributed to the difference in fertilization between parcels, as the clover proportion was still similar in both parcels (control parcel: 15%; clover parcel: 21% clover). In 2016, large differences in clover proportion (control parcel: 4%; clover parcel: 44% clover) resulted in similarly lower $N_2O$ emissions on the clover parcel as in 2015."

**R:** 'Jensen et al. (2012) based on site-years.' R: based on how many site years?

**AC:** Added "Jensen et al. (2012) based on eight site-years."

**R:** 'In addition, high total N deposition (NH3-N, NO3-N, HNO3-N, NO2-N) on intensively managed Swiss grasslands (15–40 kg N ha-1 yr-1, Seitler et al., 2016)' **R:** Can you be more specific regarding the N dep rate in your study area. The range you quote is very large.

**AC:** These values were from an available report, in order to get a specific estimate for the study area we contacted B. Rihm who was able to provide spatially explicit estimates in the revised version of the manuscript. We changed to: "In addition, high total N deposition ($NH_3$-N, $NO_3$-N, $HNO_3$-N, $NO_2$-N) in the study area (in total 33.8 kg N ha$^{-1}$ yr$^{-1}$ in 2015, Rihm, personal communication 27$^{th}$ June 2018; Rihm and Achermann, 2016) might foster background $N_2O$ emissions due to increased $NH_4$-N and $NO_3$-N availability (Butterbach-Bahl et al., 2013)"

**R:** Section 4.3: 1st paragraph: you should qualify phrases such as those shown below 'N2O emissions vary widely across sites' R: add the ranges of emissions, and presumably the studies for reference are from grasslands?) 'Higher N2O fluxes following cutting were similarly observed on a pasture in Central France (Klumpp et al., 2011).' R: what is the difference relative to your study? You have done this much better in the 2nd paragraph.

**AC:** Indeed, specific values will add clarity. Therefore, we added how large the effects were in the cited articles. Furthermore "sites" was changed to "grassland sites". (Page 16: L. 17-20) "Nevertheless, effects of fertilization on $N_2O$ emissions vary widely across grassland sites and years (0.01–3.56% in Flechard et al., 2007; 0.1–8.6% in Hörtnagl et al., 2018, 1.3 and 3.5% of fertilizer N across years in this study), indicating that fertilization alone is insufficient for explaining $N_2O$ emissions and highlighting the need to take additional drivers into account."

Similarly, we added specific values following your suggestion in the sentence on page 16: L. 24: "Higher $N_2O$ fluxes following cutting were similarly observed on a pasture in Central France (up to 3.7 mg $N_2O$-N m$^{-2}$ d$^{-1}$in Klumpp et al., 2011; up to 7.0 mg $N_2O$-N m$^{-2}$ d$^{-1}$ in this study)."

**R:** 'In agreement with our result, an experiment without seasonal frozen soils at an Irish permanent ryegrass/clover mixture, annual N2O emissions between unfertilized ryegrass' R: 'change to ' In agreement with our result, measurements from permanent grasslands in Ireland, where winter freeze-thaw cycles are very rare, a comparison of a ryegrass/clover mixture, with . . .. . .'

**AC:** We changed this similar to your suggestion (Page 17: L.09): "In agreement with our result, measurements from permanent grasslands in Ireland, where winter freeze-thaw cycles are very rare, showed that annual $N_2O$ emissions in unfertilized ryegrass (2.38 ± 0.12 kg $N_2O$-N ha$^{-1}$ yr$^{-1}$) were not significantly different from an unfertilized grass–clover sward (2.45 ± 0.85 kg $N_2O$-N ha$^{-1}$ yr$^{-1}$)", …

**R:** 'The magnitude of the fertilization effect of 2.5-fold N2O emissions on average during the week after fertilization (at 43 kg N amendment per event on average) was comparable to the effect of a 14 °C soil temperature increment if further environmental variables remained constant.' R: This sentence requires an introduction and significant explanation. It is a bit out of place here.

**AC:** An explanation would be similarly misplaced in the discussion, so we deleted the sentence it in the revised manuscript to avoid confusing the reader. We added an

**R:** Section 4.4 'Additionally, high SON content due to previous year's fertilizer amendments are expected to contribute to the persistently high production levels' R: I suppose you mean 'years' and not year's

**AC:** Correct, we meant years'. Thank you for pointing this out.

**R:** 'the over-sowing was more effective and biologically fixed nitrogen found in shoot biomass in the clover parcel summed up to 130 kg N ha-1 yr-1 while only 14 kg N ha-1 yr-1 were measured in the control parcel' R: What is the reason for the legume proportion in the control to decrease between 2015 and 2016?

**AC:** We think that several factors are relevant for the explanation, firstly growth conditions and secondly conditions for germination and establishment. Warmer and dryer conditions in summer 2015 generally favored growth of these clover plants that were already present in both fields (clover proportions were previously were 5–10%, and *T. repens* and *T. pratense* have higher optimum temperatures compared to abundant grass species). Further, but less important, the earlier timing of cutting (quite early 2015-04-22 at lower height, compared to 2016-05-27), and the higher number of cuts in 2015 (5 in 2015 compared to 4 in 2016) might have contributed to promoted clover growth. Secondly, spring conditions in 2015 were not ideal for clover germination and establishment, i.e. air temperature during the week following the over-sowing in 2015 was low ($T_{mean}$ = 5.5°C), and minimum temperatures were below zero ($T_{min}$ = –4.3°C), there was no precipitation. In contrast, conditions the week after over-sowing were warmer and wetter in 2016 ($T_{mean}$ = 9.1, $T_{mean}$ = 0.03, precipitation total 19.1 mm during the week after over-sowing). This is how the relatively small difference between parcels in 2015 and the large differences in 2016 can be explained.

**R:** 'This indicates that biologically fixed nitrogen at the Chamau could reach higher amounts than observed during our experiment.' R: Can you really deduce this statement from the New Zealand study, where the climate, soil types and perhaps even the grass and clover species used may be rather different?

**AC:** We consider statement sound as the as the cited experiment (Nyfeler et al. 2011) took place in Zurich-Reckenholz (about 30 km from our study site) under similar climatic conditions using the same clover species (*Trifolium repens* and *Trifolium pratense*).

**R:** Section 4.5 'due to large springtime emissions (Virkajärvi et al., 2010) indicating that the mitigation strategy is likely to be inappropriate for sites with seasonally frozen soils.' R: Your Swiss soils also experience winter freeze-thaw cycles, but your data suggest that this mitigation strategy works in Switzerland. Please address this discrepancy.

**AC:** We clarified here (P19L17): Much higher $N_2O$ emissions from an unfertilized grass-clover mixture (92% increase) compared to $N_2O$ emissions from a grass sward fertilized with 220 kg N $ha^{-1}$ $yr^{-1}$ were observed under boreal climate conditions in eastern Finland, due to large springtime emissions associated with freeze-thaw cycles (Virkajärvi et al., 2010). Such an effect could not be found at our site, although soils also freeze occasionally during the cold season, but at most in the top few centimetres.

**R:** 'Due to this effect, temporary grasslands may not reproduce the findings from permanent grassland.' R: You need to provide evidence for this statement. Temporary grasslands are maintained for several years, so are rather different to croplands.

**AC:** We revised to make clearer what we mean: "Although our tested mitigation strategy seems to be beneficial for permanent grasslands, Basche et al. (2014) and Lugato et al., 2018) have shown that incorporation of clover into the soil may lead to increased $N_2O$ fluxes and thus may not be the best mitigation strategy for croplands and temporary grasslands, where ploughing is done much more frequently."

**Anonymous Referee #3**

GENERAL QUESTIONS AND COMMENTS

**R:** The paper is well structured and written, it presents a comparison between two differently managed grasslands, to evaluate the impact of the addition of clover as a mitigation strategy for N2O emissions. In my opinion, it is suitable for publication in Biogeosciences. See below for some minor comments to the paper.

**R:** The focus of the paper is the evaluation of the integration of legumes as a mitigation strategy, but that does not appear explicitly in the title: I think it should be part of it.

**AC**: We agree and changed the title "Management matters: Testing a mitigation strategy for nitrous oxide emissions using legumes …."

**R:** To what extent do you think soil properties are needed (frequency of sampling, etc) to interpret flux data? Why daily or bi-weekly? Why many 20cm samples?

**AC**: We assumed large changes in soil nutrients directly after management events, while variation in soil nutrients in times without management were considered much lower following Wolf et al. (2015). We used sampling depth 0-20 cm for soil sampling to replicate the same sampling strategy as applied in Wolf et al. 2015 to assure comparability. Further, the 0-20 cm depth is also represented in biogeochemical models (e.g. DayCent SOC pool, Mineral N) which is relevant as we aim at using our measurements for model validation.

**R:** Would you have any specific suggestions for long term measurements? This is not necessary, of course, but I think it would add value to the discussion to know what you found out to be the most useful variable for your parameterisation, beside the interpretation of your results, of course.

**AC**: Concluding from our data analysis, soil temperature and moisture were crucial and the most important variables. Measurements in 0.1 m depth represented well the microclimatic conditions, while deeper sensors were redundant as these were highly correlated with 0.1 m but less powerful in explaining $N_2O$ emissions. At sites experiencing regular to freeze-thaw events, in addition to 0.1 m, the 0.05 m soil sensors would be important. $O_2$ concentrations in 0.1 m improved the prediction in addition to soil moisture, however the effort was quite high as standard sensors were not available and they correlate with soil moisture. The measured matrix potential was not used as it correlated with soil moisture and $O_2$, and we therefore consider them less important for long-term measurements. While NH4 measurements improved the predictions, NO3 and TOC did not improve predicted N2O emissions on a daily resolution, thus less frequent sampling for the latter would be sufficient.

**R:** Could you also report the GWP of the N2O measurements to express the mitigation induced by the grassland composition change?

**AC**: We added the difference between both parcels expressed as CO2-eq in the abstract of the revised version. "The mitigation management effectively reduced $N_2O$ emissions by 54% and 39% in 2015 and 2016, respectively, corresponding to 1.0 and 1.6 t $CO_2$-equivalents." We further added in the method section 2.8 the following sentence: "For the calculation of $CO_2$ equivalents ($CO_2$-eq) used factor 298, which is the current IPCC global warming potential including climate-carbon feedbacks on a 100 year basis (IPCC, 2013)."

**R:** The eddy covariance tower in this experiment is placed at the edge between two differently treated fields. Could you explain a bit better the way you tackled the advection issues between the two treatments/crops? Two years is an impressive duration for such dataset, and I would guess all conditions (time of the day, stability, etc.) have been met in such long time for both fields, but it would be good if it were expressed more clearly through the results. Also, for what concerns the special events, do you have a suitable coverage for both fields in terms of footprint?

**AC**: Quality control (see section 2.7 Eddy covariance flux post-processing) assures that stability criteria were met for all final fluxes, while fluxes were excluded if this was not the case. Furthermore, we analyzed "time of the day" versus $N_2O$ flux per event, and overall, we had both parcels well represented. Still, there were events at

which the coverage from the control parcel was less covered. In our results we compared only cases where both parcels were adequately covered in order to avoid a bias and argue that this direct comparison of both parcels provides a reasonable estimate. We will address this in a supplementary figure.

BY SECTIONS: MATERIAL AND METHODS

**R:** P5 L1> Replace "fertilised" with "added".

**AC**: Changed.

**R:** Section 2.2: What is the reason to use two different fertilisation rates in the 2 years? Is it for simulating the business as usual behaviour of the farmer, or did you increase the amount to enhance the effects of the contrast? I see that the clover abundance difference between the two years is quite relevant: is it solely due to the additional grazing? Was the field over sown at the same rates? In connection to the conclusion that up to 44% of clover addition does not lead to further N2O emissions, it could be useful to suggest how to achieve such abundances. Perhaps you could expand on this.

AC: This was done to interfere as little as possible with the farmers business as usual activities. Thus, we let the farmer chose the amount and timing consistent with how he made his choices previous to the study. We did not want to artificially increase the fertilizer rates, nor artificially change the type of fertilizer, average annual org. fertilizer amounts in previous years (2003-2015) were 260 kg N on average with a range of 184–430 kg. Logically, the 296 kg N in the first year and 181 kg N in the second year are within this range. Sowing rates are given in Table 1, we added a comment on Page 2 L6 " (see Table 1 for specific management dates, slurry composition and sowing rates)".

**R:** Section 2.7: could you specify and motivate what method you used to calculate the time lag for the different GHG species?

For the calculation of the time lag between the wind component $w$ (after coordinate rotation) and the mixing ratio of $N_2O$ ($CH_4$), we followed the covariance maximization method in combination with a pre-determined nominal lag time. This means, we identified the peak in the cross-correlation function between $w$ and $N_2O$ ($CH_4$) in a defined time window of physically possible time lags. This cross-correlation function is often noisy due to low signal-to-noise ratios as a consequence of low $N_2O$ ($CH_4$) fluxes. In our study, analyzing the frequency distribution of detected lag times over the course of a "low-flux" year yielded clear results that assisted our processing in constraining the lag time search to a relatively short but adequate time window. In order to accurately identify the time lag and avoid systematic bias, we followed a multi-step approach before the calculation of final $N_2O$ ($CH_4$) fluxes for each year:

Step (1): We first calculated the time lag for $N_2O$ ($CH_4$) for the respective year using the covariance maximization method for a relatively large search window of $\pm 10$ s, based on despiked and detrended raw data. Next, we analyzed the frequency distribution of all found lag times and identified a clear distribution spike between 0.95 and 1.40 s for $N_2O$ (Figure R3-1 shows results for the year 2013) and between 1.00 and 1.40 s for $CH_4$ (Figure R3-2), i.e. most cross-correlation peaks where found in these ranges.

[Figure]

Figure R3-1. Histogram of found lag times between the turbulent wind component w and $N_2O$ mixing ratios in a time window of $\pm\,10$ s in 2013. Peak distribution was found at lag time 1.20 s. Shown are only found lag times below 5 s for clarity.

[Figure]

Figure R3-2. Histogram of found lag times between the turbulent wind component w and $CH_4$ mixing ratios in a time window of $\pm\,10$ s in 2013. Peak distribution was found at lag time 1.10 s. Shown are only found lag times below 5 s for clarity.

Step (2): Based on the frequency distribution from (1), we defined a narrow lag search window of $0.6 - 1.8$ s, i.e. the range in which most cross-correlation peaks were found in (1). In addition, we defined the lag times of peak distribution from (1) as the nominal lag times ($N_2O$: 1.20 s; $CH_4$: 1.10 s), that is, whenever the returned time lag was one of the limits of the window (i.e. 0.6 or 1.8 s), the time lag was set to the respective nominal lag time. These specific nominal lag times were chosen because they were identified as the most representative lag times over the course of a year. We also analyzed the time series of found lag times to investigate potential seasonal differences. We found that lag times over the course of a year fluctuated slightly but were well within the range of the pre-defined search window of $0.6 - 1.8$ s. We are therefore confident that the accurate lag time is found in the defined range.

In addition, we are confident to have adequately constrained the time window due to the used measurement setup at the site. At Chamau, we measured eddy covariance data using a fully digital and real-time logging system that was described previously (Eugster and Plüss, 2010). One major advantage of this system is that wind and scalar data at a specific moment in time are merged immediately in the same data file. As a consequence, the system is not prone to time drifts between two instruments (e.g. sonic anemometer and QCL) and the lag time can therefore consistently be found in a relatively narrow time range (with small fluctuations).

RESULTS

**R:** P12 L6-7. From figure 4, it almost looks like this is not the case in the last 2 events of 2016, "41,33". The value of NH4+ after these events seems to be almost double compared to the period before. Could you comment on this, especially addressing the N deposition issue?

**AC:** You refer to the effect of the events on 2016-08-17 and 2016-09-30 taking place in the control affecting soil mineral N in the clover parcel. We agree on your observation for the event on 2016-08-17 and find it worthwhile a closer look. We always took a set of soil samples directly before the respective event in order to get the direct reference. Before the slurry application 2.5 mg $NH_4^+$-N $kg^{-1}$ increases to 5.8 mg $NH_4^+$-N $kg^{-1}$ and stays above the reference for seven days. On the last event, the story is not that clear. The reference was 5.6 mg $NH_4^+$-N $kg^{-1}$, then we measured lower (4.9 mg $NH_4^+$-N $kg^{-1}$) at the application date but higher (7.2 mg $NH_4^+$-N $kg^{-1}$) the day after and lower data compared to the reference afterwards (explicitly 2.5, 5.2, 2.9, 2.7, 2.9 and 3 mg $NH_4^+$-N $kg^{-1}$), thus only the $NH_4^+$-N measurement of the day following the application date was higher compared to the reference. The overall picture across events was rather diverse.

We therefore changed "no distinct patterns" to "no consistent patterns. We further addressed it in the discussion of the revised manuscript: "Additionally, $NH_3$ deposition on the clover parcel originating from $NH_3$ emissions from the adjacent control parcel is likely to be the cause of increased soil $NH_4^{-+}$ concentrations after the event on 17th August 2016."

**R:** P13,L2: it could be helpful to quantify this similarity, e.g. providing a ratio of C content in biomass between the 2 different treatments directly in the text.

**AC**: We addressed this by directly referring to the values. "Average C concentrations in the biomass of all harvests were similar across parcels and plant functional types (legumes 42.9– 45.6%, non-legumes 43.0–45.2% C in biomass across parcels and years, Tab. 2; Fig. 5e)."

DISCUSSION

**R:** P17,L3-4. "Grazing had only a minor influence on the overall N2O budget of the Chamau site and data analysis showed that N2O fluxes did not significantly respond to the presence of animals (Fig.7c)". I think that a quantification would be better here than referring to the plot only, i.e. the relative % in contribution on the total N2O budget, for example.

**AC**: According to your suggestion we will add the relative contribution of fluxes during grazing on the total N budget in the revised version of the manuscript (P17, L3-4).

TABLES AND FIGURES

**R:** Figures 1 and 6 seem to have the appropriate resolution, the others tend to be a bit blurry: would it be possible to increase the image resolution?

**AC:** This effect arises due to the insertion in MS WORD, the original graphics are pdfs and will be submitted as such, in high quality.

**R:** Figure 2. The caption under the image needs correction. "(b) Footprint climatology of the year 2016 with footprint contour lines of 10% to 90% in 10% steps using the Kljun et al. (2004) footprint model" belongs to previous figure; explain panels a, b, and c.

**AC:** The correct caption was added: **Figure 2.** Meteorological conditions during 2015 and 2016. (a) Average daily air temperature (2 m), (b) average daily photosynthetically active radiation (2 m). The grey bars indicate the sub-daily variability (quartiles based on 10 min values). (c) Daily precipitation sums during 2015 and 2016 (1 m).

**R:** Figure 4. Albeit the treatments with slurry were not applied on the clover field, I think it would be useful to introduce the days of treatment also on the North field charts (perhaps the same dotted lines, no arrows). If no slurry was directly applied, the amount of N in the air during the fertilisation events has certainly changed, and potentially increased the amount of BNF on the clover field.

**AC:** Fertilizer applications are followed by $NH_3$ emissions (not measured), which can subsequently increase $NH_3$ deposition on the clover parcel and therefore fertilize the clover parcel. However, this should not affect BNF as BNF is not expected to be limited by N availability.

**References**

Aubinet, M., Vesala, T. and Papale, D.: Eddy Covariance - A practical guide to measurement and data, Springer. [online] Available from: //www.springer.com/de/book/9789400723504 (Accessed 6 December 2017), 2012.

Brentrup, F. and Pallière, C.: GHG emissions and energy efficiency in European nitrogen fertiliser production and use., Proceedings - International Fertiliser Society, (No.639), 1–25, 2008.

Eugster, W. and Plüss, P.: A fault-tolerant eddy covariance system for measuring CH4 fluxes, Agricultural and Forest Meteorology, 150(6), 841–851, doi:10.1016/j.agrformet.2009.12.008, 2010.

Flechard, C. R., Ambus, P., Skiba, U., Rees, R. M., Hensen, A., van Amstel, A., van den Pol-van Dasselaar, A., Soussana, J.-F., Jones, M., Clifton-Brown, J., Raschi, A., Horvath, L., Neftel, A., Jocher, M., Ammann, C., Leifeld, J., Fuhrer, J., Calanca, P., Thalman, E., Pilegaard, K., Di Marco, C., Campbell, C., Nemitz, E., Hargreaves, K. J., Levy, P. E., Ball, B. C., Jones, S. K., van de Bulk, W. C. M., Groot, T., Blom, M., Domingues, R., Kasper, G., Allard, V., Ceschia, E., Cellier, P., Laville, P., Henault, C., Bizouard, F., Abdalla, M., Williams, M., Baronti, S., Berretti, F. and Grosz, B.: Effects of climate and management intensity on nitrous oxide emissions in grassland systems across Europe, Agric. Ecosyst. Environ., 121(1–2), 135–152, doi:10.1016/j.agee.2006.12.024, 2007.

Flisch, R., Sinaj, S., Charles, R. and Richner, W.: Grundlagen für die Düngung im Acker- und Futterbau (GRUDAF), edited by Agroscope, 2009.

Foken, T.: Micrometeorology, 2008th ed., Springer, Heidelberg., 2008.

Fratini, G., Ibrom, A., Arriga, N., Burba, G. and Papale, D.: Relative humidity effects on water vapour fluxes measured with closed-path eddy-covariance systems with short sampling lines, Agricultural and Forest Meteorology, 165, 53–63, doi:10.1016/j.agrformet.2012.05.018, 2012.

Hollinger, D. Y., Goltz, S. M., Davidson, E. A., Lee, J. T., Tu, K. and Valentine, H. T.: Seasonal patterns and environmental control of carbon dioxide and water vapour exchange in an ecotonal boreal forest, Global Change Biology, 5(8), 891–902, doi:10.1046/j.1365-2486.1999.00281.x, 1999.

Hörtnagl, L., Barthel, M., Buchmann, N., Eugster, W., Butterbach-Bahl, K., Díaz-Pinés, E., Zeeman, M., Klumpp, K., Kiese, R., Bahn, M., Hammerle, A., Lu, H., Ladreiter-Knauss, T., Burri, S. and Merbold, L.: Greenhouse gas fluxes over managed grasslands in Central Europe, Global Change Biology, doi:10.1111/gcb.14079, 2018.

Ibrom, A., Dellwik, E., Flyvbjerg, H., Jensen, N. O. and Pilegaard, K.: Strong low-pass filtering effects on water vapour flux measurements with closed-path eddy correlation systems, Agricultural and Forest Meteorology, 147(3–4), 140–156, doi:10.1016/j.agrformet.2007.07.007, 2007.

Jensen, E. S., Peoples, M. B., Boddey, R. M., Gresshoff, P. M., Hauggaard-Nielsen, H., Alves, B. J. R. and Morrison, M. J.: Legumes for mitigation of climate change and the provision of feedstock for biofuels and biorefineries. A review, Agron. Sustain. Dev., 32(2), 329–364, doi:10.1007/s13593-011-0056-7, 2012.

Jørgensen, F. V. and Ledgard, S. F.: Contribution from stolons and roots to estimates of the total amount of $N_2$ fixed by white clover (*Trifolium repens* L.), Ann Bot, 80(5), 641–648, doi:10.1006/anbo.1997.0501, 1997.

Klumpp, K., Bloor, J. M. G., Ambus, P. and Soussana, J.-F.: Effects of clover density on $N_2O$ emissions and plant-soil N transfers in a fertilised upland pasture, Plant Soil, 343(1–2), 97–107, doi:10.1007/s11104-010-0526-8, 2011.

Nemitz et al. (in review) Standardisation of eddy-covariance flux measurements of methane and nitrous oxide, International Agrophysics

Nyfeler, D., Huguenin-Elie, O., Suter, M., Frossard, E. and Lüscher, A.: Grass–legume mixtures can yield more nitrogen than legume pure stands due to mutual stimulation of nitrogen uptake from symbiotic and non-symbiotic sources, Agriculture, Ecosystems & Environment, 140(1–2), 155–163, doi:10.1016/j.agee.2010.11.022, 2011.

Roth, K.: Bodenkartierung und GIS-basierte Kohlenstoffinventur von Graslandböden, Diploma Thesis, University of Zurich., 2006.

Wilks, D. S.: Statistical Methods in the Atmospheric Sciences, Volume 100, Third Edition, 3 edition., Academic Press, Amsterdam ; Boston., 2011.

---

## Referee Report (RR1)

Dear authors,

I have edited the below few sentences, and deleted the last one, which is a bit confusing and does not really add to the story.

Page 3: Our mitigation approach investigated the potential for reductions in slurry application accompanied with increased clover proportion in the pasture to reduce $N_2O$ emissions at the field-scale. Farmers currently use a combination of home-produced slurries and **bought in** mineral fertilizer. Our suggestion is to apply the slurry **to the** fields which are amended with mineral fertilizers. This would have an additional benefit of reducing the indirect greenhouse gas emissions i.e. those during the manufacture of mineral fertilizers.

reductions in GHG emissions, which are beyond the field-scale, as well as the full farm nitrogen and GHG budget are well¶
beyond the focus of this study would need further investigation.